# Carbene-stabilized enantiopure hetero-metallic clusters featuring EQE of 20.8% in circularly-polarized OLED

Xiao-Hong Ma[1,4], Jing Li[1,4], Peng Luo[2], Jia-Hua Hu[1], Zhen Han [1],
Xi-Yan Dong [1,2] ✉, Guohua Xie [3] ✉ & Shuang-Quan Zang [1] ✉

Bright and efficient chiral coinage metal clusters show promise for use in emerging circularly polarized light-emitting materials and diodes. To date, highly efficient circularly polarized organic light-emitting diodes (CP-OLEDs) with enantiopure metal clusters have not been reported. Herein, through rational design of a multidentate chiral N-heterocyclic carbene (NHC) ligand and a modular building strategy, we synthesize a series of enantiopure Au(I)-Cu(I) clusters with exceptional stability. Modulation of the ligands stabilize the chiral excited states of clusters to allow thermally activated delayed fluorescence, resulting in the highest orange-red photoluminescence quantum yields over 93.0% in the solid state, which is accompanied by circularly polarized luminescence. Based on the solution process, a prototypical orange-red CP-OLED with a considerably high external quantum efficiency of 20.8% is prepared. These results demonstrate the extensive designability of chiral NHC ligands to stabilize polymetallic clusters for high performance in chiroptical applications.

Organic light-emitting diodes (OLEDs)[1–3] that emit circularly polarized luminescence (CPL)[4–6] are promising candidates for next-generation solid-state display and sensing applications. Compared with the widely studied expensive heavy metal-containing compounds, such as Ir(III) and Pt(II) complexes[7,8], abundant coinage metal-based emitters are more suitable for mass production. However, enantiopure emitters that contain coinage metals have not been developed for circularly polarized OLEDs (CP-OLEDs). Cyclometalated Au(III)[9] that emits phosphorescence and cyclic carbene-coordinated Cu(I)[10–12] that emits thermally activated delayed fluorescence (TADF)[13] are capable of utilizing triplet excitons for light generation in OLEDs. Unfortunately, the complicated synthesis of chiral ligands makes the development of new emitters for highly efficient CP-OLEDs challenging.

Metal-based clusters serving as a bridge between single atoms and metal particles have attracted widespread attention[14–21]. Luminescent coinage metal-based clusters[22–26] have been applied in OLEDs[27–30] with circularly polarized electroluminescence[31,32] because of their tunable colors, high photoluminescence quantum yields (PLQYs), solution processability, and easy synthesis. Recently, a bidentate phosphine ligand-stabilized $Cu_4(I)$ cluster with a high PLQY of 93% and an external quantum efficiency (EQE) of 11%[28], a bidentate phosphine-chelated $Cu_4(I)$-$I_4$ cluster with a high PLQY of 65% and an EQE of 8%[29], and a pair of chiral bidentate thiazolidine-2-thione-based ligand-stabilized $Ag_6(I)$ clusters with PLQYs over 95%[33] were reported. Nevertheless, no circularly polarized electroluminescence was observed from these clusters. Typically, these types of clusters are either achiral or unsuitable for use in OLEDs. In addition, the small spin-orbit coupling (SOC) parameters and the high reorganization energies are inherent shortcomings of pure Cu-based chiral emitters that prevent their application in CP-OLEDs[31]. The efficient integration of the high PLQY,

[1]College of Chemistry, Zhengzhou University, 450001 Zhengzhou, China. [2]College of Chemistry and Chemical Engineering Henan Polytechnic University, 454000 Jiaozuo, China. [3]Sauvage Center for Molecular Sciences, Hubei Key Lab on Organic and Polymeric Optoelectronic Materials, Department of Chemistry, Wuhan University, 430072 Wuhan, China. [4]These authors contributed equally: Xiao-Hong Ma, Jing Li. ✉e-mail: dongxiyan0720@hpu.edu.cn; guohua.xie@whu.edu.cn; zangsqzg@zzu.edu.cn

enantiopurity, stability, and compatibility of chiral clusters to prepare CP-OLEDs remains challenging.

The N-heterocyclic carbene (NHC) ligand, which is both neutral and electron-rich, forms strong coordination bond with metal atoms[34–37]. Its unique electronic and steric properties can be easily modified by functionalization of the imidazolium ring, principally at the nitrogen position, with different organic functional moieties[38]. Cathleen M. Crudden and collaborators reported the first examples of NHC-containing $Au_{11}$ and $Au_{13}$ clusters[39], which inspired the development of NHC-based Au and Cu clusters[40–46]. To date, only three chiral NHC ligand-based metal clusters, in which only C atoms coordinate to the metal and chiral functionalization occurs at the N position of the benzimidazole or imidazole ring, have been reported[47,48]. A pair of $Au_{10}$ clusters containing (S/R) binaphthyl-NHC was reported without any CPL[47]. Two pairs of $CAu_6^I$ using the enantiomer chiral NHC ligands were synthesized and exhibited a low PLQY of 30%[48]. Inspired by these findings, we utilized NHC ligands with more than one functional position on the imidazole ring to prepare stable metal clusters for use in high-performance CP-OLEDs by rational design.

Herein, we report a functional modular design strategy for generating chiral NHC-based Au(I)-Cu(I) clusters with high stability, PLQYs and excellent CPL. Considering TADF emitters through harvesting singlet excitons showed superiority in OLED applications, for high-efficiency TADF, metal clusters having the appropriate lowest unoccupied molecular orbital (LUMO) to avoid cluster-center emission is necessary[29,33]. Conceiving above strategy, we embedded pyridine/quinolone ring with adjustable π* orbitals in ligand, so that we achieve spatial separation between donor (Cu with d electrons) and acceptor (pyridine/quinolone with π*) moieties in metal cluster. In addition, chiral functionalization occurred at two C positions on the imidazole ring of the NHC ligands, and side arms containing pyridine/quinoline rings was introduced at two N atoms to increase the denticity of the ligands. These side-arms support the heterometal atoms within the cluster according to the distinct affinity of the coordinating atoms; for example, the C atom in NHC prefers to bind to Au atoms rather than to Cu atoms, and the N atom in the pyridine/quinoline ring prefers to ligate to Cu atoms. Increased denticity is expected to effectively circumvent the nonradiative decay of metal clusters in the excited state. Two pairs of enantiomers of tridentate NHC ligands were designed and synthesized (Fig. 1a): (4R, 5R)-/(4S, 5S)-N, N'-di(chloromethyl)pyridine)-4, 5-diphenyl-4, 5-dihydro-imidazolinium hexafluorophosphate (denoted as R/S-NHCpy-H·PF6) and (4R, 5R)-/(4S, 5S)-N, N'-2-(chloromethyl)quinoline-4, 5-diphenyl-4, 5-dihydro-imidazolinium hexafluorophosphate (denoted as R/S-NHCql-H·PF6) ligands. Five pairs of enantiopure Au(I)-Cu(I) clusters were obtained based on the above chiral ligands by modulating the auxiliary halide ions. They all displayed mirror-image circular dichroism (CD) and CPL responses, good thermal stability and excellent solubility. The efficient TADF mechanism favors a maximum PLQY of 93.0%. A prototypical CP-OLED fabricated with the chiral NHC-based cluster exhibited a reasonably high EQE of 20.8%, outperforming all other reported cluster complexes.

## Result
### Synthesis and structures
Enantiopure R/S-NHCpy-H·PF6 and R/S-NHCql-H·PF6 were prepared from (1R, 2R)-/(1S, 2S)-(±)-1,2-diphenyl-1,2-ethanediamine according to literature procedures[49,50] (Fig. 1a and Supplementary Fig. 1). The mononuclear NHC-gold(I) precursors R/S-[Au(NHCpy)2]PF6 and R/S-[Au(NHCql)2]PF6 were synthesized (Supplementary Fig. 2). Details of the synthesis and characterizations can be found in the Supplementary Methods.

Five enantiomeric pairs of Au(I)-Cu(I) clusters containing chiral NHC ligands were prepared by the reaction of R/S-[Au(NHCpy)2]PF6 or R/S-[Au(NHCql)2]PF6 precursors and CuX (X = Cl, Br, and I) in $CH_2Cl_2$ at room temperature. Typically, R/S-[Au(NHCql)2]PF6 was first dissolved in

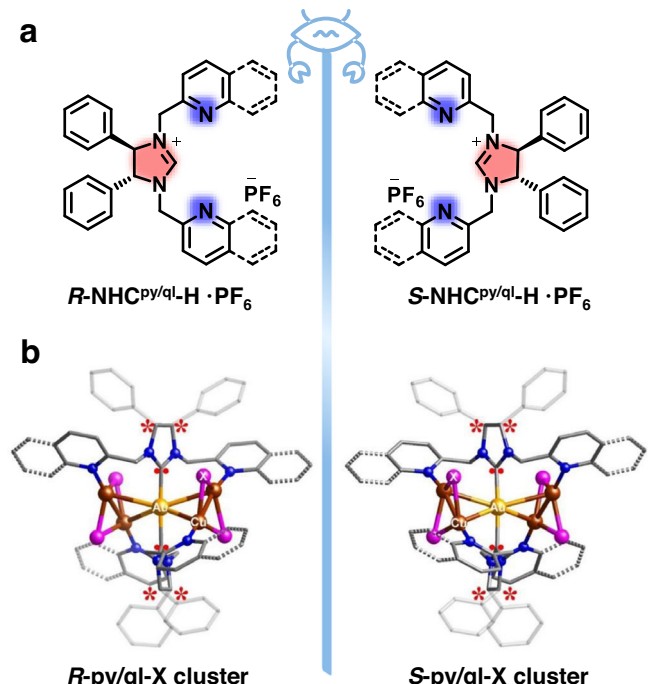

**Fig. 1 | Ligand design and crystal structures. a** Ligand structures of R/S-NHCpy/ql-H·PF6. **b** Structure of the enantiomers of R/S-py/ql-X (X = Cl, Br, and I). Au, yellow; Cu, brown; N, dark blue; C, gray; X (Cl, Br, and I), purple. The H atoms were omitted for clarity.

$CH_2Cl_2$ before a CuI (4 equiv) suspension of $CH_2Cl_2$ was added, and the mixture was subsequently stirred for an additional 2 h, resulting in a yellow solution. The solution was filtered and then concentrated by rotary evaporation, and then a yellow powder was produced after the addition of diethyl ether. The yellow powder was recrystallized from $CH_2Cl_2$ and diethyl ether to produce yellow block crystals that were identified as R/S-AuCu4I4(NHCql)2PF6·($C_2H_5$)2O (denoted as R/S-ql-I) as determined by single-crystal X-ray diffraction (SCXRD) at 200 K (Supplementary Table 1). Similarly, four additional enantiomeric pairs of clusters were synthesized (Supplementary Tables 2–5): R/S-AuCu4Br4(NHCpy)2PF6·(($C_2H_5$)2O)·($CH_2Cl_2$)0.5 (denoted as R/S-py-Br), R/S-AuCu4I4(NHCpy)2PF6·(($C_2H_5$)2O) (denoted as R/S-py-I), R/S-AuCu4Cl4(NHCql)2PF6·(($C_2H_5$)2O)0.5 (denoted as R/S-ql-Cl) and R/S-AuCu4Br4(NHCql)2PF6·(($C_2H_5$)2O)1.5·($CH_2Cl_2$) (denoted as R/S-ql-Br) (see the Supplementary Information for synthesis details).

SCXRD analysis revealed that R/S-py-Br belonged to the orthorhombic Sohncke space group $P2_12_12_1$ (Supplementary Table 2). R/S-py-I crystallized in the tetragonal Sohncke space group $I4_1$ (Supplementary Table 3). The R/S-ql-X (X = Cl, Br, and I) nanoclusters crystallized in the orthorhombic Sohncke space groups $C222_1$, $P2_12_12_1$ and $P2_12_12$, respectively (Supplementary Tables 1, 4, and 5). The flack parameters of all these clusters are near zero, suggesting that their crystal structures have inherent homochirality. Crystallographically, each pair of clusters exhibited a perfect mirror symmetry structure (Fig. 1b and Supplementary Figs. 3 and 4). Five pairs, including R/S-py-X (X = Br and I) and R/S-ql-X (X = Cl, Br, and I), exhibited similar structures and ligand coordination modes. R-ql-I, as a representative, was primarily discussed in the structure analysis. R-ql-I contains four copper(I) atoms and one gold(I) atom bridged by two NHC ligands and stabilized by four I atoms, with one PF6- counterion. The motif of AuCu4 can be represented as two triangles of Cu-Au-Cu sharing a gold atom, which is nearly orthometric with a dihedral angle of 88.27° (Supplementary Fig. 5). In addition, two Cu-Au-Cu triangular planes and four I-Cu-Cu triangular planes have different bond lengths and bond angles, suggesting structural distortion of inorganic $AuCu_4I_4$

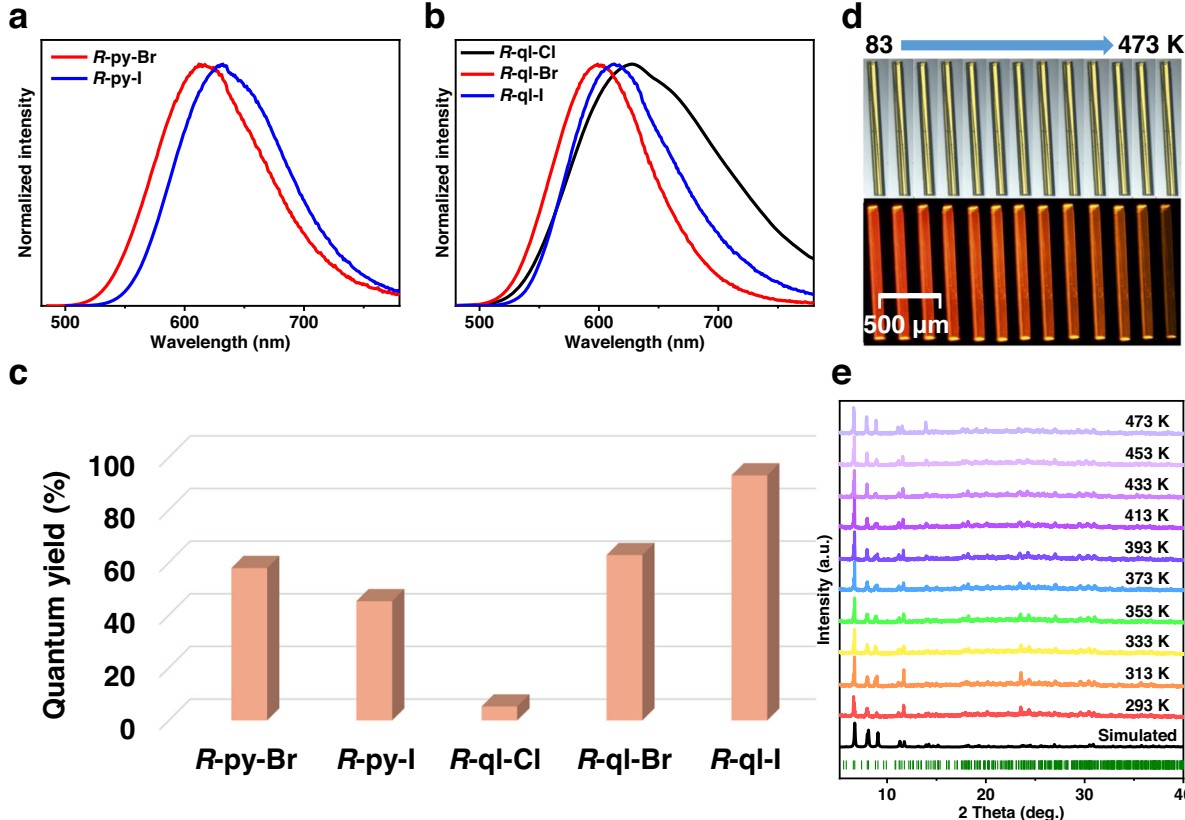

**Fig. 2 | Photoluminescence properties and thermal stability. a, b** Solid-state photoluminescence spectra of complexes *R*-py-X (X = Br and I) and *R*-ql-X (X = Cl, Br, and I) excited at 400 nm. **c** PLQYs of five *R*-type clusters in the solid state. **d** Images of a single crystal of *R*-ql-I in the range of 83–473 K under ambient and UV light. **e** PXRD patterns of *R*-NHC$^{ql}$-AuCu$_4$-I in the range of 293–473 K.

moieties induced by chiral ligands. Moreover, each center Au(I) bridges two NHC moieties via strong Au–C bonds. Each Cu(I) atom is coordinated by two iodine atoms with ∠ICuI ranging from 119.29–120.45° and one N atom of the pyridyl group, where the Cu–Cu distances are remarkably short, with a range from 2.429 to 2.431 Å, indicating the presence of ultrastrong cuprophilic interactions[51,52] (Supplementary Tables 6–10). Additionally, hydrogen atoms of the two methylene groups in all clusters are depicted interacting with gold atoms, with a minimum C–H⋯Au distance of 2.759 Å, which is shorter than the sum of Van Der Waals radii of hydrogen and gold (2.860 Å) (Supplementary Fig. 6)[53,54].

The formula and crystalline-phase purity of these five pairs of chiral clusters were further confirmed by elemental analysis, Fourier transform infrared (FT-IR) spectroscopy, thermogravimetric analysis (TGA), and powder X-ray diffraction (PXRD) (Supplementary Figs. 7–9). *R/S*-py-X (X = Br and I) and *R/S*-ql-X (X = Cl, Br, and I) were further characterized by solution-phase $^1$H-nuclear magnetic resonance ($^1$H-NMR) spectroscopy (Supplementary Figs. 10–14). When heated at 150 °C, it was found a slight shift in PXRD patterns of *R*-ql-Br, suggesting that the stacking was slightly compressed, yet the cluster molecules were not destroyed, as evidenced by the $^1$H-NMR spectroscopy of the redissolved samples (Supplementary Fig. 15) and the refinement of the PXRD patterns using the reflex module combined in the Materials Studio program with ultra-fine convergence quality (Supplementary Fig. 16). The other four pairs of crystals, including *R/S*-py-X (Br and I) and *R/S*-ql-X (Cl and I), also maintained identical crystalline behaviors after high temperatures in the air, which was rare among metal clusters[33]. We further carefully examined the temperature-dependent PXRD of *R*-ql-I until 200 °C, and the good crystalline phase was retained (Fig. 2 and Supplementary Fig. 9).

## Photoluminescence

We then examined the photoluminescence properties (Fig. 2 and Table 1). These NHC-stabilized chiral Au(I)-Cu(I) nanoclusters showed intense PL in the solid and solution states, and their colors differed depending on the arms of the pyridine/quinoline and halides (X = Cl, Br, and I), with emission peaks ranging from 600 nm to 653 nm (Fig. 2a and Supplementary Figs. 17–21). Interestingly, we found that for pyridine-containing clusters, the emission wavelengths for the solid samples were longer than those in dichloromethane solutions. In terms of the quinoline-containing clusters, the emission peak of the *R*-ql-Cl cluster was blue-shifted from 628 nm in the solid state to 608 nm in solution ($10^{-5}$ M). In contrast, it was slightly redshifted from 600 nm to 608 nm for the *R*-ql-Br cluster. Nevertheless, the peak positions remained nearly identical at 613/612 nm for the *R*-ql-I cluster (Fig. 2b and Supplementary Figs. 20 and 21). Moreover, the emission intensity at the maximum of solid-state *R*-py-X (X = Br and I) and *R*-ql-X (X = Cl, Br, and I) barely changed after irradiation for 3 h with a 400 nm xenon lamp, indicating the excellent photostability of these clusters

**Table 1 | Room-temperature luminescence parameters of five *R*-type clusters in the solid state and CH$_2$Cl$_2$ solution (1.0 × 10$^{-5}$ mol/L)**

| Cluster | Solid | | | Solution | | |
|---|---|---|---|---|---|---|
| | λ$_{em}$ (nm) | τ (μs) | PLQY (%) | λ$_{em}$ (nm) | τ (μs) | PLQY (%) |
| *R*-py-Br | 616 | 8.50 | 57.9 | 628 | 5.10 | 20.1 |
| *R*-py-I | 631 | 1.51 | 45.3 | 653 | 0.64 | 27.1 |
| *R*-ql-Cl | 628 | 1.63 | 5.4 | 608 | 0.30 | 0.1 |
| *R*-ql-Br | 600 | 7.75 | 62.8 | 607 | 4.42 | 12.8 |
| *R*-ql-I | 613 | 2.02 | 93.0 | 612 | 1.75 | 49.4 |

(Supplementary Fig. 22). In terms of PLQY, both *R*-py-Br and *R*-py-I clusters had moderate values of 57.9% and 45.3% in the solid state, respectively. For the set of *R*-ql-X (X = Cl, Br, and I), the PLQY increased abruptly from 5.4% to 62.8% to 93.0% in the order of Cl→Br→I (Fig. 2c and Table 1). Although the PLQY decreased in solution due to non-radiative energy loss, all of the samples exhibited an identical trend in the solid state. Furthermore, these clusters all displayed microsecond radiative lifetimes determined by time-resolved decay measurements (Table 1 and Supplementary Figs. 23 and 24). Notably, *R*-ql-I had a high PLQY of over 93.0% and a delayed lifetime of 2.02 µs, which are strongly reminiscent of the properties of carbene-Cu/Au complexes with TADF properties[12]. Therefore, the effects of the temperature on the PL intensity and the radiative lifetime of solid-state *R*-ql-I were investigated.

## Temperature-dependent photoluminescence

To investigate the TADF properties, we first tracked the low-temperature PL properties. To exclude the possible structural variation due to cooling, we performed SCXRD on the *R*-ql-I in the range of 100–300 K, which showed an identical crystalline structure, with a slight expansion of the unit-cell size, and slight elongation of Cu-Cu separation (average from 2.419 to 2.437 Å) (Supplementary Tables 11 and 12).

Considering the good stability at high temperatures, PL spectra of *R*-ql-I in the temperature range of 93 to 383 K were recorded. Solid-state *R*-ql-I cluster demonstrated blueshifts of emission peaks with the increase of temperature (Fig. 3a). Interestingly, two peaks at low temperatures were observed at 625 (high-energy peak, HE) and 700 nm (low-energy peak, LE) (Fig. 3b and Supplementary Fig. 25). As the temperature increased from 93 to 183 K, the HE peak intensity gradually increased, while the LE peak intensity gradually decreased until it disappeared at -183 K (Fig. 3c and Supplementary Fig. 26). With further warming from 183 K to 273 K, the HE peak intensity peak further increased, with a continuous blueshift from 620 nm to 615 nm.

From 273 to 383 K, the position and intensity of the HE peak remained nearly consistent. These observations indicated that the thermal activation of the HE peak was continuous from 93 to 273 K. The room-temperature PLQY was determined to be up to 93.0%. Nevertheless, room-temperature emission intensity was slightly lower than that at 273 K (Fig. 3c), probably because of the competition between the thermal-induced nonradiative effects and thermal-activated emission enhancement[33].

The decay lifetimes (τ) of the HE and LE peaks of solid state *R*-ql-I at different temperatures were characterized by microseconds of 2.5–4.5. Below 183 K, the τ of the LE peak was slightly longer than that of the HE peak, and they both gradually decreased (Fig. 3d). The curve of temperature versus HE peak lifetime was fitted according to the modified Boltzmann equation[55], giving a radiative rate of $k(S_1)$ of $2.68 \times 10^6$ s$^{-1}$, a $k(T_1)$ equal to $2.61 \times 10^5$ s$^{-1}$ and a $\Delta E(S_1-T_1)$ of 0.052 eV. The fast radiative rate and sufficiently small energy gap between the emissive $S_1$ and $T_1$ states support the possible TADF emission[56] in the range of measured temperatures. We tentatively ascribed the HE peak to the $S_1$ state and the LE peak to the $T_1$ state. However, the emission spectra of *R*-ql-Cl consisted of two peaks (Supplementary Fig. 27): the resolved HE peak was slightly blueshifted, its intensity remained nearly constant over the measured temperature range (93–303 K), and it was assigned to fluorescence/TADF emission; the resolved LE peak was largely blueshifted, with intensity decreased quickly, which was assigned to phosphorescence. For *R*-ql-Br, there was only one emission peak in the range of 93 to 303 K, and the emission intensity increased gradually as the temperature decreased (Supplementary Fig. 28). For the *R*-py-X (X = Br and I), although the emission blue-shifted with increasing temperature, in the range of 93 to 303 K (Supplementary Figs. 29 and 30), no increase in emission was observed. Based on these comparisons of variable-temperature emission behaviors, we speculated that the luminescence mechanism was distinct among *R*-ql-X (X = Cl, Br, and I). Subsequently, we applied density functional theory (DFT) and time-dependent DFT (TD-DFT) calculations to gain insights into the PL origin.

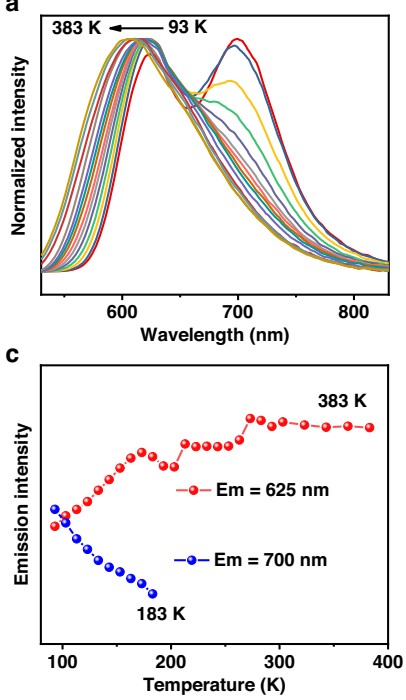

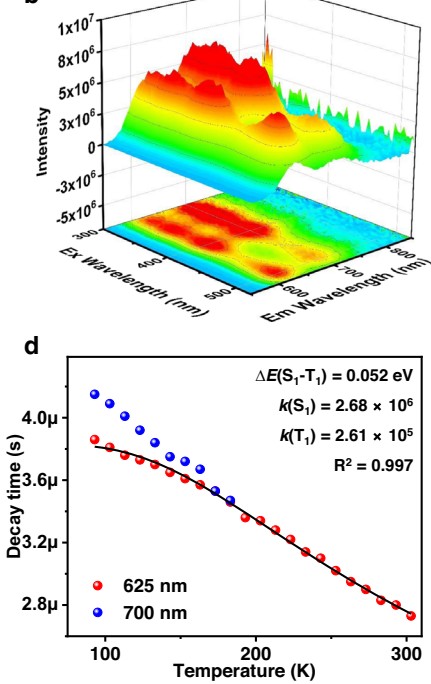

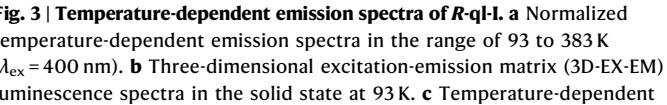

**Fig. 3 | Temperature-dependent emission spectra of *R*-ql-I. a** Normalized temperature-dependent emission spectra in the range of 93 to 383 K ($\lambda_{ex}$ = 400 nm). **b** Three-dimensional excitation-emission matrix (3D-EX-EM) luminescence spectra in the solid state at 93 K. **c** Temperature-dependent low-energy and the high-energy emission peak intensities in the range of 93 to 383 K. **d** Plot of transient decay lifetimes against temperature (93 to 303 K); the black line represents the fit according to the equation accounting for TADF.

## Theoretical calculations

By comparing the LUMO energy level between ligands $R$-NHC$^{py}$-H and $R$-NHC$^{ql}$-H, we found that they are localized on the N-containing aromatic arms. Yet the LUMO of $R$-NHC$^{ql}$-H was 4.08 eV lower than that of $R$-NHC$^{py}$-H (3.81 eV), and the energy gap was smaller than the latter by 0.85 eV (Supplementary Fig. 31). The important distinction directly led to the different electronic structures and electron transitions (Supplementary Figs. 32–39 and Supplementary Tables 13–17) for emission among $R$-py-X (X = Br and I) and $R$-ql-X (X = Cl, Br, and I) of the metal cluster. For these five pairs of Au(I)-Cu(I) clusters, the highest occupied molecular orbital (HOMO) was principally derived from Cu(d) and X(sp). The LUMO density of $R$-ql-X (X = Cl, Br, and I) was mainly distributed on the quinoline moieties, while the LUMO of $R$-py-X (X = Br and I) was localized on Au-Cu motifs, with much less distribution on the pyridine moieties (Supplementary Fig. 32). Thus, for the $R$-py-X (X = Br and I), the electronic transitions for efficient emission mainly originated from the intracluster ligand-to-metal-metal charge transfer (LMMCT; X → Cu-Au) hybridizing metal-centered (ds/dp) transitions[33]. The $R$-py-X (X = Br and I) had a similar PLQY of 40–60%, probably due to the inherent intracluster transitions[57]. For the $R$-ql-X (X = Cl, Br, and I), we assumed that the low-energy transition arose from the inter-ligand trans-metallic charge-transfer transition (ITCT, X/Au-Cu → π* of quinoline), which is usually observable in closed-shell metal coordination complexes such as those of Ag(I) and Pt(II) containing N-heterocyclic ligands[58,59].

To visually analyze the transition characteristics of the excited states of these clusters, we delineated the natural transition orbital (NTO) hole-electron pairs in the $R$-ql-X (X = Cl, Br, and I) and $R$-py-I using the overlap integral ($Sr$) and centroid distance ($D$) of the norm of each NTO pair of $S_1$, $T_1/T_2$ states (Fig. 4, Supplementary Fig. 40 and Supplementary Tables 18–20), and we analyzed the SOC strength between these two states. For the $R$-ql-X (X = Cl, Br, and I) in $S_1$ states, they all had similar charge-transfer (CT) excited states[59] due to larger $D$ values (>4.0 Å) and smaller $Sr$ values (<0.21). While in the $T_1$ states of $R$-ql-X (X = Cl, Br, and I) with the gradually increased heavy-atom effect from Cl to I, the $D$ values gradually decreased (1.93 Å, 1.72 Å, 0.77 Å)

and the $Sr$ values gradually increased (0.71, 075, 0.86). Yet, the optimized $T_2$ states of $R$-ql-X (X = Cl and Br) showed more local excitation (³LE) character ($D$ < 0.60 Å) than that of $T_1$ ($D$ > 1.70 Å), which also had the higher <$S_1$|$H_{SOC}$|$T_2$> than <$S_1$|$H_{SOC}$|$T_1$> values. In contrast, the $T_1$ states in $R$-ql-I showed a more ³LE character, which also had bigger <$S_1$|$H_{SOC}$|$T_1$> values than that of $R$-ql-X (X = Cl and Br). The values of spin-orbit coupling matrix elements (SOCME) can also be qualitatively estimated through El-Sayed's rules[60], namely, the SOCME is relatively large if the transition between singlet and triplet states involves a change of orbital type. The respectively calculated SOCME based on the geometries of $S_1$, $T_1/T_2$ (Supplementary Table 21) and the structure of the minimum energy crossing point (MECP) (Supplementary Fig. 41 and Supplementary Tables 22–24) had a consistent trend, which both conformed to the El-Sayed's rule. Therefore, in $R$-ql-I, the LE characteristics in the triplet state ($T_1$) in synergy with the CT characteristics for the singlet state ($S_1$) could be in favor of activating TADF emission. The calculations of $R$-py-I was given in Supplementary Fig. 40 and Supplementary Table 25.

For gaining more insight into the heavy-atom effect contributions to SOC, we analyzed the atom (Au, Cu, Cl, Br, I) contributions to holes and electrons in $S_1$, $T_1/T_2$ states and calculated contributions of basis functions to hole and electron (Supplementary Tables 26–28). Cu atoms (bonded $Cu_2$ dimer)[51] and halide (Cl, Br, and I) ions mainly contributed to the hole of $R$-ql-X (X = Cl, Br, and I) (Fig. 4) and R-py-I (Supplementary Fig. 40). For $R$-ql-X (X = Cl, Br, and I), the Au atom contribution to holes and electrons was very small (<1%); while in $R$-py-I the Au atom contribution to holes was larger than 27%. The results demonstrated that the functional module of pyridine/quinoline with different π* orbitals in ligands modulated the electronic structures and further affected the contribution of Au to SOC.

The TD-DFT calculations showed that the $\Delta E$($S_1$-$T_1$) energy gap decreased in the $R$-ql-Cl cluster (0.294 eV), the $R$-ql-Br cluster (0.193 eV), and the $R$-ql-I cluster (0.116 eV), which could partially be ascribed to the increasing HOMO energy level along the direction in $R$-ql-X (X = Cl, Br, and I) cluster (Fig. 5). The decreased $\Delta E$($S_1$-$T_1$) supported the feasibility of TADF in the heaviest $R$-ql-I cluster[38]. The $T_2$

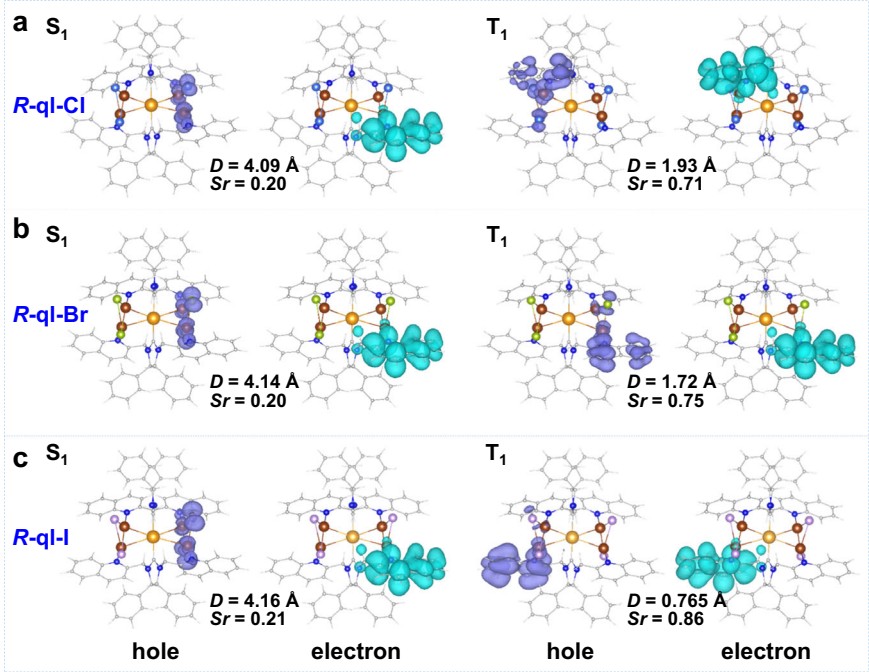

**Fig. 4 | Natural transition orbital analysis. a–c** The hole and electron pairs for $S_1$/$S_0$ and $T_1$/$S_0$ transitions were obtained by natural transition orbital analysis at optimized $S_1$ and $T_1$ geometries of $R$-ql-Cl (**a**), $R$-ql-Br (**b**), and $R$-ql-I (**c**) (isovalue of 0.02). Au, yellow; Cu, brown; N, dark blue; C, gray; Cl, blue; Br, green; I, light, purple; H, white.

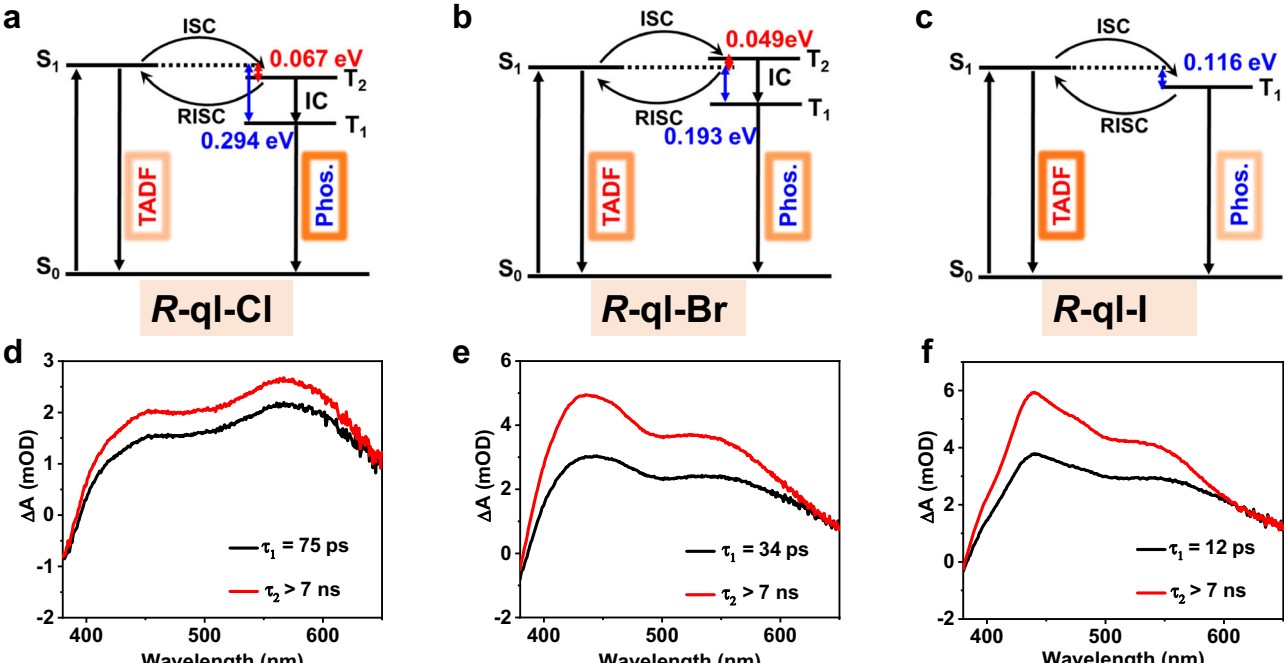

**Fig. 5 | TADF and phosphorescence emission processes and transient absorption.** **a–c** Energy diagram of *R*-ql-Cl (**a**), *R*-ql-Br (**b**), *R*-ql-I (**c**) indicating TADF and phosphorescence emission processes. **d–f** Global fitting results for the femtosecond probe TA spectra of *R*-ql-Cl (**d**), *R*-ql-Br (**e**), *R*-ql-I (**f**).

state was 0.067 eV lower than the $S_1$ state in the *R*-ql-Cl cluster, while the *R*-ql-Br cluster $T_2$ state was 0.049 eV higher than the $S_1$ state (Fig. 5). The calculated intersystem crossing rate ($k_{ISC}$) from $S_1$ to $T_1$ of *R*-ql-Br ($1.04 \times 10^{11} s^{-1}$) and *R*-ql-I ($2.60 \times 10^{12} s^{-1}$) was much larger than that of *R*-ql-Cl ($0 s^{-1}$) (Supplementary Tables 29–30). The fast calculated rate of reverse intersystem crossing ($k_{RISC}$) also followed the order of *R*-ql-I cluster ($7.05 \times 10^{10} s^{-1}$) > *R*-ql-Br cluster ($4.22 \times 10^{9} s^{-1}$) > *R*-ql-Cl cluster ($5.49 \times 10^{8} s^{-1}$) (Supplementary Table 29). Resultantly, the faster $k_{ISC}$ and $k_{RISC}$ supported the higher PLQY in the order of 5.4% (*R*-ql-Cl) < 62.8% (*R*-ql-Br) < 93.0% (*R*-ql-I). In addition, the calculated emission energies of the excited state $S_1$ (2.100 eV) and $T_1$ (1.713 eV) were close to the experimental HE (2.024 eV at room temperature) and LE (1.771 eV) at 93 K, indicating the reliability of these calculations (Supplementary Table 31).

Based on the above calculations and aforementioned temperature-dependent PL experimental results, we proposed the possible emission mechanism for the *R*-ql-X (X = Cl, Br, and I) cluster (Fig. 5a–c). The phosphorescence emission was dominated by a small section of TADF in the *R*-ql-Cl cluster (Fig. 5a). Therefore, *R*-ql-Cl, we gave a reasonable explanation: the $S_1$ to $T_2$ ISC followed by the rapid IC to $T_1$ occurred but the RISC to $S_1$ was rather prohibited, resulting in the large contribution of phosphorescence with low efficiency (5.4%). For *R*-ql-Br, the two-channel spin-flip processes $RISC_{T2-S1}$ and $RISC_{T1-S1}$ could support the TADF process. Thus, TADF and phosphorescence might contribute to the observed emission spectra, resulting in an integrated single peak of *R*-ql-Br and a moderate efficiency (62.8%) (Fig. 5b). For *R*-ql-I, we assigned the LE emission to phosphorescence, which was merely observed below 183 K. Such phosphorescence disappeared above ~183 K due to continually increasing TADF. And the HE emission was assigned to the TADF emission, which was dominant above 200 K (Fig. 5c), leading to high PLQY (93.0%) at room temperature.

For the *R*-py-X (X = Br and I), the $S_1$ states exhibited similar electron and hole distributions, whose separations were less significant than those of the *R*-ql-X (X = Cl, Br, and I). Although the emission blueshifted with increasing temperature, in the range of 83 to 303 K, no increase in emission was observed. The small blueshift was tentatively

ascribed to the expansion of metal-metal bonds in clusters, leading to a reduction in the $d\pi$ orbital of metal atoms in the HOMO and hence an increased energy gap[21]. In addition, the efficient SOC induced by the central Au atom (Supplementary Table 26) supports the high phosphorescence QY of 40–60% in the solid state at room temperature.

Therefore, we gave an overview of the functional modules in these clusters (Fig. 6). The coordinated halide (Cl, Br, and I) ions served as electron donors, contributing to the HOMOs, and optimized the ISC process through the heavy-atom effect (Br and I). With the aid of the extending arms of the pyridine/quinoline groups, the additional chelating atoms aimed to catch the N-philic Cu(I) atoms, where the $\pi$ and $\pi^*$ orbitals of the side-arms not only tuned the LUMOs but also the contribution of heavy atoms to SOC of other atoms. The carbene C atom was coordinated to Au atoms, as in the precursor, where Au promoted SOC and increased ISC because of the heavy atom effect in *R*-py-X (X = Br and I). Thus, the energy gap between $S_1$ and $T_1/T_2$, SOC, ISC and RISC was modulated by the carbene ligand and halide, leading to different emission performance and an ultrahigh PLQY of *R*-ql-I through the TADF mechanism.

## Transient absorption spectroscopy

We performed transient absorption (TA) spectroscopy to further examine the differences in the ultrafast electron dynamics *R*-ql-X (X = Cl, Br, and I) clusters (Fig. 5d–f). Under pumped laser excitation at 360 nm, *R*-ql-X (X = Cl, Br, and I) clusters displayed net ground-state bleaching (GSB) at about 380–385 nm and two photoinduced absorptions (PA) signals (centered at ~450 nm and ~570 nm, respectively). Note that spectra below 380 nm were not detected because of the photo leak of the pump laser, but we could still speculate that it was also GSB signal according to the results of UV-visible absorption of *R*-ql-X (X = Cl, Br, and I) clusters. The broad PA signal peaks were attributed to the excited state absorption. However, the two absorbed signal intensities of *R*-ql-Cl were slightly different from those of *R*-ql-Br and *R*-ql-I, which may be due to the larger absorption cross sections of *R*-ql-Cl at ~570 nm, while *R*-ql-X (X = Br and I) had larger absorption cross-sections at ~450 nm. Following the initial signal decay, the PA signal featured a signal build-up process, suggesting that an excited

triplet state appeared. So we could safely attribute the signal build-up process to the ISC process from the singlet to the triplet. According to the global fit, we got the rising ISC rate of *R*-ql-Cl (75 ps), *R*-ql-Br (34 ps), to *R*-ql-I (12 ps) (Fig. 5d–f), which were consistent with the trend of the DFT calculations (Supplementary Table 29).

## Chirality and circularly polarized luminescence

The CD and CPL spectra were used to characterize the chiroptical properties of these Au(I)-Cu(I) clusters in the ground state and the excited state. In CH₂Cl₂ solution ($1 \times 10^{-5}$ mol/L) at room temperature, these enantiomeric pairs of clusters all gave rise to a new CD band in the range of 340–400 nm, related to those of the precursor *R/S*-[Au(NHC$^{py/ql}$)₂]PF₆ with CD signal peaks below 350 nm (Supplementary Figs. 42 and 43). CD responses were basically consistent with the UV–vis absorption spectra, and those above 340 nm generally involved the electronic transitions between orbitals of cluster molecules (Supplementary Figs. 33 and 34, 42–47 and Supplementary Tables 13–17). For

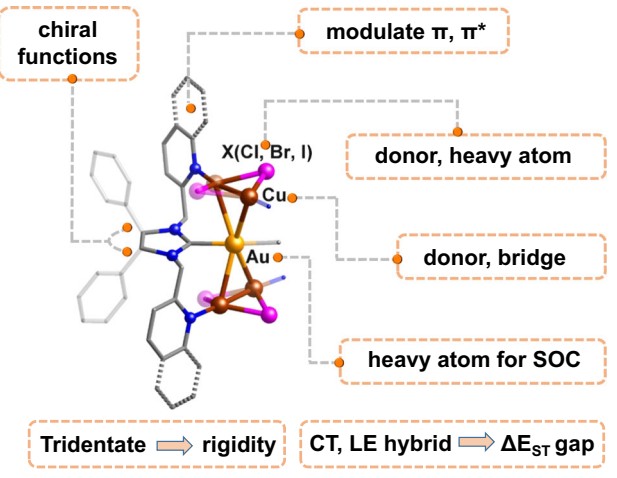

**Fig. 6 | Functional modules in the metal cluster.** Color codes: Au, yellow; Cu, brown; N, dark blue; C, gray; X (Cl, Br, and I), purple. One ligand and the H atoms were omitted for clarity.

example, *R*-ql-I cluster had CD band with peaks at 377 nm, which mainly originated from transitions from HOMO-4 (Cu₄-dominated) to LUMO+3 (quinoline dominated) and LUMO+4 (Cu₄-dominated). For those CD signals below 340 nm, the chiral electronic transitions partially overlapped with ones in the chiral precursor (Supplementary Figs. 43, 45, and 47). The mirror-image CPL peaks of five enantiomeric pairs cluster were nearly consistent with their respective PL emission peaks (Fig. 7), with $g_{lum}$ values at the level of $10^{-4} - 10^{-3}$ similar to the calculated results (Supplementary Figs. 48–50 and Supplementary Table 32). Given the above analysis of the emissive excited states of the cluster, CPL in the *R*-ql-X (X = Cl, Br, and I) should be TADF-characterized and phosphorescence-characterized in the *R*-py-X (X = Br and I), which were all contributed by the combined the ligands and Cu₄Au cluster skeleton. These findings suggested the chiroptical activities of the whole cluster molecule.

## Circularly polarized organic light-emitting diodes

The integrated advantages, including the TADF-characterized CPL with high PLQYs and the microsecond lifetime at room temperature, as well as their good solubility and resistance to radiation and high temperature, inspired us to test the CP-OLEDs with *R/S*-ql-I (defined as *R*-OLED/*S*-OLED) using the solution-process technique. We conducted a complete characterization of the electroluminescence (EL) performance of *S*-ql-I. The device structure is shown in Fig. 8a, i.e., ITO/PEDOT:PSS/mCP:*S*-ql-I/DPEPO/TmPyPB/Liq/Al, where PEDOT:PSS is poly(3,4-ethylene-dioxythiophene):poly(styrenesulfonate) and Liq is lithium 8-qui-nolinolate, which served as the hole and electron injection layers, respectively. DPEPO (bis[2-(diphenylphosphino)phenyl] ether oxide) and TmPyPB (1,3,5-tri[(3-pyridyl)-phen-3-yl]benzene) were the hole-blocking layer and the electron transporting layer, respectively. The emitting layer was composed of the host:guest structure of mCP (9,9'-(1,3-phenylene)bis-9H-carbazole):*S*-ql-I (98:2, wt./wt.%). Details of the device fabrication and characterization can be found in the Supplementary Information. The current density–voltage–luminance (*J*-*V*-*L*), EQE, power efficiency (PE) and current efficiency (CE) versus the current density of the devices are displayed in Fig. 8c. Bright orange–red emission with a peak wavelength of 586 nm was observed (Fig. 8b), corresponding to the CIE 1931 color coordinates of (0.51, 0.47) (Supplementary Fig. 51). The EL spectra of the devices were

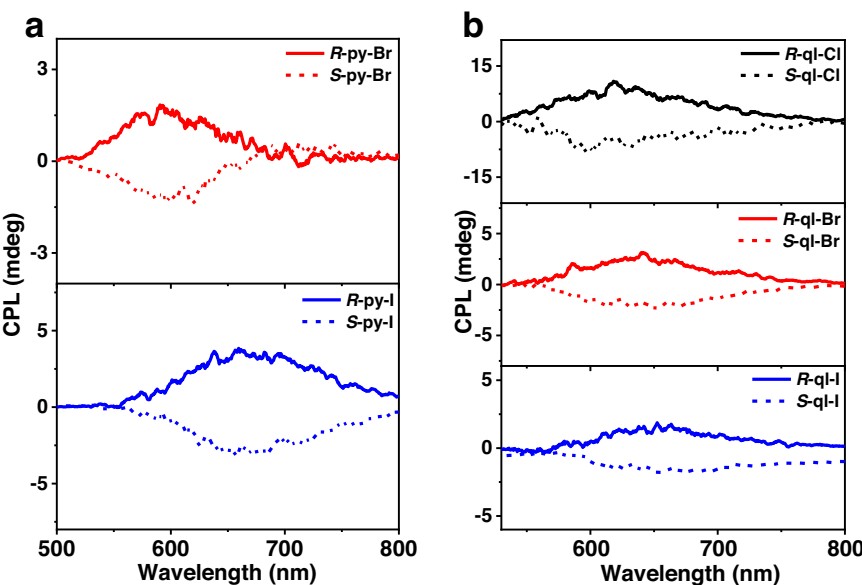

**Fig. 7 | Chiroptical properties.** CPL spectra of *R/S*-py-X (X = Br and I) (**a**) and *R/S*-ql-X (X = Cl, Br, and I) (**b**) in CH₂Cl₂ ($1 \times 10^{-5}$ mol/L) under ambient conditions.

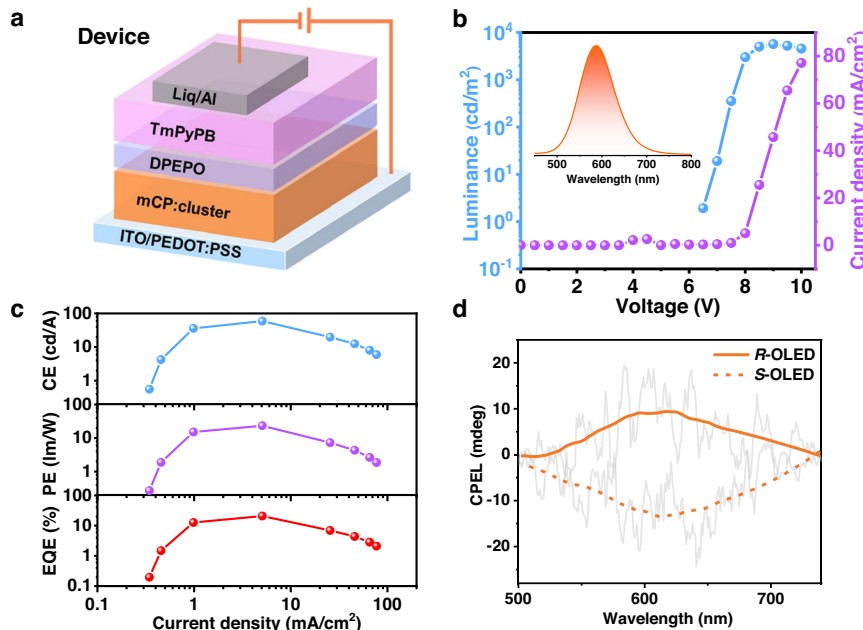

**Fig. 8 | CP-OLED performance. a** Device architecture of CP-OLED. **b** Current density–voltage–luminance characteristics of *S*-OLEDs. Inset: EL spectra. **c** CE, PE, and EQE vs. current density curves of *S*-OLEDs. **d** CPEL spectra of the devices.

blueshifted by 27 nm compared with their PL spectra in the solid state, which may be due to the electronic structure and charge transfer process of the cluster being sensitive to the polarity of the surrounding environment[61] (Fig. 8b). Unprecedentedly, maximum EQE, CE, and PE values of 20.8%, 59.2 cd/A, and 23.3 lm/W, respectively, were detected (Fig. 8c), accompanied by a high brightness over 5600 cd/m². Such values have not previously been reported in the literature for NHC-protected metal cluster-based OLEDs. The EQE of 20.8% at 586 nm of *S*-ql-I cluster was smaller than the reported linear Au(I) complex with NHC ligands (CMA1, 26.3% at 550 nm)[11], probably because the longer lifetime of *S*-ql-I (~2 μs) than CMA1 (~350 ns) was not a beneficial factor in OLED performance. Furthermore, the CP-OLEDs exhibited obvious circularly polarized EL (CPEL) signals. Reasonable dissymmetry factors ($g_{EL}$) of $5.0 \times 10^{-4}$ for the *R*-OLED and $-7.7 \times 10^{-4}$ for the *S*-OLED (Fig. 8d and Supplementary Fig. 52) were basically consistent with the photoluminescence asymmetry factor ($|g_{PL}| = 9.0 \times 10^{-4}$), which showed that the chirality of cluster in OLED devices did not change a lot. Our emitter resulted in the first bright and efficient CP-OLED based on assembled chiral NHC-stabilized coinage metal clusters.

## Discussion

In summary, a functional modular strategy has been used for judiciously designing and synthesizing Au(I)-Cu(I) clusters for high-efficiency orange–red CPL. The key feature of the TADF mechanism promotes triplet exciton utilization and makes them among the most efficient CPEL materials. We engineered NHC ligands by harmoniously incorporating chirality, multiple-denticity for rigidity, and modifiable π* orbital of arms for TADF and engineer inorganic moieties by embedding heavy atoms of a single central Au and halogen, which markedly improved TADF efficiency. The CPL of the solid-state clusters and solution-processed CP-OLEDs present perfect mirror-image symmetry with orange–red EQEs of up to 20.8%. This work holds great promise for the rational functionalization of NHC ligands in developing high-efficiency CPL cluster emitters and opens a new avenue for the development of new cluster-based LEDs and CP-OLEDs that can be solution-processable.

## Methods

### Materials and reagents

All the chemicals for synthesis were obtained from commercial sources and used without any further purification. The organic chiral ligands *R/S*-NHC^py-H PF₆ and *R/S*-NHC^ql-H PF₆ used were synthesized with modification according to the literature[49,50].

Synthesis of *R/S*-[Au(NHC^py)₂]PF₆ and *R/S*-[Au(NHC^ql)₂]PF₆. *R/S*-NHC^ql-H PF₆ (0.65 g, 1 mmol) or *R/S*-NHC^py-H PF₆ (0.54 g, 1 mmol), Ag₂O (66 mg, 0.28 mmol), and about 40 mg of ⁿBu₄PF₆ in 40 mL of CH₂Cl₂ was added. The mixture was protected from light and stirred for 10 min at room temperature. NaOH (1 M, 3 mL) was then added, and stirring was continued for 4 h. The mixture was filtered through Celite, and the clear filtrate was reduced to the minimum volume under vacuum. *R/S*-[Ag(NHC^ql)₂]PF₆ or *R/S*-[Ag(NHC^py)₂]PF₆ was precipitated as a white powder by the addition of diethyl ether. Next, a 25 mL round-bottom flask was charged with *R/S*-[Ag(NHC^ql)₂]PF₆ (0.38 mg, 0.3 mmol) or *R/S*-[Ag(NHC^py)₂]PF₆ (0.32 mg, 0.3 mmol) in 30 mL of CH₂Cl₂. Me₂SAuCl (0.096 g, 0.3 mmol) in 10 mL of CH₂Cl₂ was added dropwise. The mixture was protected from light and stirred for 30 min during which time a precipitate formed. The solution was filtered through Celite removing the precipitated AgCl. The clear filtrate was subsequently reduced to 2 mL, and a white powder was precipitated with diethyl ether. ESI-MS: 1005.3745 for *R*-[Au(NHC^py)₂]PF₆, 1205.4320 for *R*-[Au(NHC^ql)₂]PF₆.

### Synthesis of *R/S*-ql-X (X = Cl, Br, and I), *R/S*-py-X (X = Br and I)

*R/S*-[Au(NHC^ql)₂]PF₆ or *R/S*-[Au(NHC^py)₂]PF₆ (0.05 mmol) was dissolved in 6 mL of CH₂Cl₂. To this was added CuX (0.2 mmol) (X = Cl, Br, and I) suspended in 6 mL of CH₂Cl₂. The solution was stirred for an additional 3 h during which a yellow solution formed. The mixture was filtered through Celite, and the clear filtrate was reduced to 2 mL, then a yellow powder was precipitated with diethyl ether. The yellow powder was collected by filtration yielding. The yellow powder was recrystallized from CH₂Cl₂ and diethyl ether to produce yellow crystals.

### Crystallographic data collection and refinement of the structure

*R/S*-py-X (X = Br and I) and *R/S*-ql-X (X = Cl, Br, and I) were measured by single-crystal X-ray diffraction (SCXRD) with a Bruker diffractometer at

200 K, using Mo·Kα radiation (λ = 0.71073 Å). *R*-ql-I was also measured at different temperatures (100, 150, 180, 200, 250, and 300 K). The intensities were corrected for absorption using the empirical method implemented in SCALE3 ABSPACK scaling algorithm. The structures were solved with intrinsic phasing methods (SHELXT-2015), and refined by full-matrix least-squares on $F^2$ using *OLEX2*, which utilizes the SHELXL-2015 module. The least-squares refinement of the structural model was performed under hard geometry restraints and displacement parameter restraints due to the weak diffraction and serious disorder of $PF_6^-$, $Et_2O$ and $CH_2Cl_2$ molecules in the lattice, such as ISOR, SADI, SIMU, and DFIX. Solvent molecules of all clusters have been identified and further confirmed by thermogravimetric and elemental analysis. All host molecular atoms were refined anisotropically, and the hydrogen atoms were included in idealized positions. The crystallographic data were listed in Supplementary Tables 1–6 and 11.

## Quantum chemical calculations

In this study, the PL origin of a series of ligand-protected Au(I)-Cu(I) alloy clusters (*R*-py-X (X = Br and I) and *R*-ql-X (X = Cl, Br, and I)) has been studied by the DFT and TD-DFT calculations. The DFT and TD-DFT calculations were carried out using the Gaussian 16 program[62]. The hybrid PBE0 functional in conjunction with the Def2-SVP basis set was used for geometric optimization of the ground state and excited state configuration of ligand-protected Au(I)-Cu(I) alloy clusters. Hole and electron pair distribution analyses were performed using the Multiwfn code[63,64]. Kohn–Sham (KS) orbital energy levels were calculated by the Amsterdam density functional (ADF 2016) software package[65]. SOCME were calculated by ORCA 5.0.0 software package[66] based on the PBE0 functional and the DKH-def2-TZVP(-f) basis set (SARC-DKH-TZVP for Au and I atoms). The radiative rate of fluorescence ($k_f$) and phosphorescence ($k_p$) were evaluated by Einstein spontaneous emission relationship. The ISC rate constant ($k_{ISC}$), RISC rate constant ($k_{RISC}$), and the internal conversion rate constant ($k_{IC}$) were obtained by the semiclassical Marcus theory expression. The MECP structure has been calculated by TD-DFT using sobMECP Program[67] and Gaussian 16 program. The specific calculation details are present in Supplementary Information.

# Data availability

The data that support the findings of this study are available from the corresponding author upon request. Source data for Figs. 2, 3, 5, 7, and 8 are provided in the figshare (https://doi.org/10.6084/m9.figshare.23559900). The X-ray crystallographic coordinates for the structures reported in this article have been deposited at the Cambridge Crystallographic Data Centre (CCDC) under deposition number CCDC *S*-ql-Cl (2225237), *R*-ql-Cl (2225238), *R*-py-Br (2225239), *S*-py-Br (2225240), *S*-ql-Br (2225243), *R*-ql-I-100 K (2225244), *R*-ql-I-180 K (2225245), *R*-ql-I-150 K (2225246), *R*-ql-I-200 K (2225247), *R*-ql-I-250 K (2225248), *R*-ql-I-300 K (2225249), *R*-ql-Br (2225250), *S*-py-I (2225251), *R*-py-I (2225252), *S*-ql-I-200 K (2225253). These data can be obtained free of charge from the CCDC via www.ccdc.cam.ac.uk/data_request/cif.

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

## Acknowledgements

This work was supported by the National Natural Science Foundation of China (Nos. 92061201, S.-Q.Z.; U21A20277, X.-Y.D.; 21825106, S.-Q.Z.; 21975065, X.-Y.D.; and 62175189, G.X.) and Zhengzhou University. This work was also supported by the Zhongyuan Thousand Talents (Zhongyuan Scholars) Program of Henan Province (234000510007, S.-Q.Z.) and The Excellent Youth Foundation of Henan Scientific Committee (No.232300421022, X.-Y.D.). The authors also appreciate the assistance of Professor Yong Pei from Xiangtan University in the calculation.

## Author contributions

S.-Q.Z. conceived and designed the experiments. X.-H.M. conducted the synthesis and characterization. J.L. and P.L. performed the calculations. J.-H.H. and Z.H. drew figures in the main text. G.X. is in charge of the fabrication of OLED devices. X.-Y.D. and S.-Q.Z. analyzed the experimental results. X.-H.M., X.-Y.D., and S.-Q.Z. co-wrote the manuscript.

## Competing interests

The authors declare no competing interests.
