## [Peer Review File · Nature Communications]

Carbene-Stabilized Enantiopure Heterometallic Clusters
Featuring EQE of 20.8% in Circularly-Polarized OLEDReviewers' Comments:

Reviewer #1:

Remarks to the Author:

In this work, the authors developed the Au(I)-Cu(I) clusters with N-heterocyclic carbene (NHC) ligands for the efficient orange-red circular polarized luminescence (CPL). The clusters show high external quantum efficiency (EQE) up to 20.8% in the thermally activated delayed fluorescence (TADF) mechanism. The DFT and TDDFT calculations were performed to analyze the TADF mechanism and the photoluminescence as well as absorption spectra. Although this work is of interest, there are still many errors and ambiguous discussion points in the theoretical analysis part and therefore, I cannot recommend the publication of this paper in the present form. The authors should consider the following points before the publication.

(1) In principle, the spin-orbit coupling (SOC) for the intersystem crossing should be calculated at the minimum energy crossing points or the potential seam between S1 and T1 in the present system. However, the authors did not consider this point in the calculations; the geometries of S1 and T1 were used.

(2) In p. 8, the authors discussed the origin of the SOC values by "heavy atom effect" of Au, Br, and I, which is not clear. Which atom is relevant for the spin-orbit interaction in the transition between S1 and T1 in fact? Please go into deeper analysis and make the origin clear. Also, one can discuss the SOC values using the El-Sayed rule by analyzing the orbitals.

(3) There are still many errors and defects in the present manuscript. For example, p. 13, "0.95 eV" must be "0.85 eV" (6.03-5.18) from the values in Figure 1. Also, the HOMO level seems to be drawn in the same level for NHC^{py} and NHC^{ql}. All these levels should be shown with the absolute values. "Fig. 1B" should be "Fig. 1b". The word "concentrated" should be "localized" in the standard expression.

(4) The calculated absorption spectra in Supplementary Figure 24 are largely deviated from the experimental UV-Vis spectra. This difference should be clearly explained with reasons. The agreement of the emission energies from S1 and T1 seems to be accidental.

(5) The authors discussed the intracluster charge transfer. The orbital composition analysis should make this much clearer.

(6) In p. 14, the number of 2.59×10^{-144} seems to be strange when compared to the other values. Please check the value and this value may have cancellation of significant digits.

(7) I do not understand the values of 5.4% and 62.8% to 93% given in p. 14. Please make this clear.

(8) In p. 15 and Table S18, the value of (23.4.26) is wrong and not clear. Because R-NHC^{pl}-AuCu₄-I lacks T2, I do not understand this discussion. Also, the values of T1(1.713 eV) and LE(1.771 eV) cannot be found in any table, please make this part clearer.

(9) It is very difficult to follow the discussion in particular, the theoretical part. The discussion is sometimes for R-NHC-AuCu₄-Cl, -Br, -I and sometimes only for R-NHC-AuCu₄ -I. Please write down the discussion much clearer.

(10) The explanation of D and Sr values in p. 15 will help the understanding the general readers, although the authors cited ref. 53.

(11) I do not understand the word "triplet effects" in p. 16. The authors should explain what this means in more detail.

(12) In p. 19, Supplementary Figs. 24-25 are absorption spectra of these clusters and not related to the chiral electronic transitions.

(13) The observed rotatory strengths of CPL were very small in Figure 6 and the g_{lum} values were 10^{-4} - 10^{-3} . The g -value and rotatory strength of these clusters can be easily calculated. I encourage the authors to show the calculated values and compare with the experimental values.

Reviewer #2:

Remarks to the Author:

Please see the attached file for my review comments.

In this manuscript, Zang et al. reported five enantiopure pairs of Au^ICu^I₄ clusters possessing interesting chiroptic properties for Circularly-Polarized OLED (CP-OLED) applications. The synthesis of the Au^ICu^I₄ clusters was carried out in the same way as the previous literature (ref 44 of this study), using CuX (X = Cl, Br, I) instead of using [Cu(MeCN)₄]PF₆. The authors address the important issue of studying in detail, both experimentally and theoretically, the effects of ligands (both NHC and secondary ligand halides) on the luminosity, chirality, and electronic structure of these clusters. An interesting aspect of this research is its application to CP-OLEDs based on strong photoluminescence with a quantum yield of up to 93% in the solid state and an external quantum efficacy as high as 20.8%. In order to finally be published in Nature Communications, the following issues should be properly resolved.

Abstract

1. On page 2, lines 20,21. "clusters without heavier metals such as Ru, Ir, Os and Pt...". Au is "heavier" than these metals, so please reconsider this expression.

Introduction

2. On page 5, lines 80, 84: "pyridine" => "pyridine/quinoline".
3. On page 5, line 87: "cationic R/S-NHC^{py} and R/S-NHC^{ql} ligands". First, the ligands are neutral, as noted in line 62. Second, this is the first time the authors mention the abbreviation of the ligand, it is necessary to give full names or formulas.

Synthesis and structures

4. On page 5, line 96: "R/S-NHC^{py}-PF₆ and R/S-NHC^{ql}-PF₆". Since the pre-ligands are protonated, it is more appropriate to name them "R/S-NHC^{py}-H·PF₆ and R/S-NHC^{ql}-H·PF₆".
5. In the Supplementary Information (page 5, line 98). The "yield: 72%" is misleading, and therefore the respective yields of **R/S-[Au(NHC^{py})₂]PF₆** and **R/S-[Au(NHC^{ql})₂]PF₆** should be addressed. Also, NMR data for the new clusters including **R/S-[Au(NHC^{py})₂]PF₆**, **R/S-[Au(NHC^{ql})₂]PF₆**, **R/S-NHC^{ql}-AuCu₄-X** (X = Cl, Br and I), and **R/S-NHC^{py}-AuCu₄-X** (X = Br and I) are missing.
6. On page 7, line 140. For Cu(I)...Cu(I) interactions, rather than using "copperphilic interactions", the term of "cuprophilic interactions" is commonly used in the reported literature. See references, for example, Chem. Commun., 2016, 52, 2932-2935; J. Am. Chem. Soc. 2021, 143, 3808-3816.
7. On page 8, line 154, "... , yet the cluster molecules were not destroyed, as evidenced by the photoluminescence of the redissolved samples". I understand what the author is trying to say, but if decomposition has occurred, it would be

- better to confirm the chemical species by NMR, since the decomposition components that do not emit light cannot be determined by photoluminescence.
8. The possible existence of the C-H...M interactions between the hydrogen atoms of the methylene groups in the wingtip and gold (or copper) in the cluster structure should be commented on. The following references may be consulted: H. Schmidbaur, *Angew. Chem. Int. Ed.* 2019, 58, 5806; M. Shionoya et al., *Bull. Chem. Soc. Jpn.* 2021, 94, 1324.
 9. On page 8, the authors first explain **Fig. 2** in the paragraph of "lines 147-160", then go back to **Fig.1d** in the paragraph of "lines 161-173". For example, it would be better to explain **Fig. 1d** first, then **Fig. 2**, and so on, explaining each figure in turn. In fact, it was only upon reviewing the theoretical calculations section (pp. 13-18) that I finally realized that the paragraph "lines 161-173 on page 8" may be the conclusion the authors reached after conducting a theoretical review. (For example, lines 164-165, "...where Au promoted SOC and increased intersystem crossing ISC by the heavy atom effect". Line 170, "optimized the ISC process through the heavy-atom effect (Br and I)." Page 9, line 172, "...HOMOs and LUMOs to promote the TADF mechanism (?)") Thus, the conclusion after the experiment seems strange and abrupt in **Fig. 1d**. Such conclusions based on theoretical calculations would be better properly moved to the theoretical calculation section to avoid confusion.

Photoluminescence

10. On page 9, lines 186-189, the sentence "In addition,....**R-NHC^{qL}-AuCu₄-I** barely changed after irradiation of the sample with 400 nm light for 3 h, indicating the excellent photostability of this material." seems unclear. For instance, is this result in solution or in the solid state? What is the light source? Is it strong light (for example strong UV light (300 W Hg lamp)) or just normal light of the HORIBA FluoroLog-3 fluorescence spectrometer used in this work? Besides the stability of **R-NHC^{qL}-AuCu₄-I**, what about photostability studies of other clusters involved in this study?
11. In the Supplementary Information (page 27, line 388), "Supplementary Figure 17.... (c) ... in the solution state". What solvent was used? Under argon or ambient condition? Please show the solvent in solution state in each related Supplementary Figure.

Temperature-dependent Photoluminescence

12. On page 11, lines 217-218, "... variable-temperature SCXRD on the solvent-free **R-NHC^{qL}-AuCu₄-I** in the range of 100-300 K, which yielded an identical

crystalline structure". What is the Cu(I)...Cu(I) distance, because the Cu(I)...Cu(I) distance was considered important for tuning the photoluminescence?

13. On page 11, line 221, "... measurements were performed from 93 to 383 K". This shows the temperature for Fig. 3b, but is not consistent with the "83 K to 303 K in Supplementary Figures 19-20". And line 231, "...was continues from 83 to 273 K" is also the same. It would be better to unify the temperature ranges to avoid confusion.
14. On pages 11-12, lines 231-234, "The experimental QY was determined to be up to 93%, lower than that at 273 K because of competition between the thermal-induced nonradiative effects and thermal-activated emission enhancement". This sentence is difficult to understand. Does the sentence "The experimental QY (at what temperature?) was determined to be up to 93%, lower than that at 273 K..." mean that the QY at 273 K is the highest QY among the QY values measured at 83-383 K?

Theoretical calculations

15. On page 13, lines 255-257, "...yet is 0.95 eV lower than that of R-NHCpy (Fig 1B)". The author here returned to Fig. 1B for an explanation. It would be clearer to move the theoretical part in Fig 1b to this theoretical section. The sentences of lines 163-172 seem to be the conclusion provided by the authors in the theoretical part, so it is a bit strange when it appeared in the part of synthesis and structure.
16. On page 8, lines 164-165, "...where Au promoted SOC and increased intersystem crossing ISC by the heavy atom effect". As claimed by the authors, Au atom is an important heavy atom for SOC (spin-orbit coupling) as shown in Fig. 1d. However, no clear evidence is provided in the theoretical calculations. For example, have the authors studied the heavy atom effect of this Au atom in comparison to its homometallic Cu analog (such as a similar cluster in which the central Au atom in the Au^ICu^I₄ cluster is replaced by a Cu atom)? Moreover, the reference (Chem. Commun., 2016, 52, 2932-2935) mentioned that dicopper NHC-picoyl complexes show that coprophilic interactions can ensure strong SOC. Therefore, it seems important to identify the real factors driving SOC.

Chirality and circularly polarized luminescence

17. On page 18, lines 368-369. To understand the chiroptical signals contributed by the asymmetrically arranged Cu(I)-halogen moieties in these novel Au^ICu^I₄ clusters, the authors may compare the enantiopure pairs of ***R/S***-[Au(NHCpy)₂]PF₆ and ***R/S***-[Au(NHCql)₂]PF₆ with the Au^ICu^I₄ clusters by CD measurements. What is the

possible reason for "transferring the chirality at the two C atoms of NHC ligands to the whole cluster molecule (page 19, lined 378-379)"?

18. On page 18. The conditions for CD spectra (Supplementary Figures 35-36) and CPL spectra (Figure 6) in solution should be described.
19. In the corresponding Supplementary information, pages 46-47, lines 548-554, Supplementary Figs. 35-36, the CD signals for the clusters in the solid state and in solution are different, for example, the *q*l-based cluster (***R/S*-NHC^q*-AuCu₄-I***) in the range of 300-200 nm. Why does this make a difference? Does cluster aggregation affect the chiroptical signals?

Circularly polarized organic light-emitting diodes

20. With regard to the result that the AuI₄ cluster in this study shows a high external quantum efficiency, it is also interesting to evaluate the comparison with a linear Au(I) complex with NHC ligands (external quantum efficiency = 26.3%, ref. 11 in this study).

General issues:

21. Regarding cluster names: it is confusing that different cluster names are used in parts of the cluster. For example, *R/S*-AuCu₄I₄(NHC^q)₂PF₆ is denoted *R/S*-NHC^q-AuCu₄-I in the text (lines 110-114) and Figure 6, and the same compound is abbreviated "V" in Table 1 and Figures 2, 4 and 5. In addition, it was sometimes referred to as a "*q*l-I cluster" (line 186), "*q*l set cluster" (line 221), "*q*l base set" (line 259), or "*q*l-I base cluster" (line 278). In one paper, systematically name all compounds and use only one abbreviation for each compound. For example, I-V or 1-5 would be acceptable.
22. It is better to indicate the solvent used, especially in Table 1 and in the captions of the supplementary figures.
23. Regarding significant digits: the authors sometimes use "93%" (e.g., line 28) and sometimes "93.0%" (e.g., line 91). All values of quantum yield, efficiency, etc. in the text should be consistent.

Reviewer #3:

Remarks to the Author:

This review is of crystallographic work presented in the manuscript.

A total of 17 single crystal structures are presented in the manuscript along with figures of PXRD data. There are significant problems with the refinement and reporting of many of the structures which need to be addressed before the work can be recommended for publication. Additional points on the presentation of the SCXRD and PXRD work in the manuscript should be addressed.

Comments on main manuscript crystallographic content:

125 Did you mean "chiral space groups" or Sohnke space groups?

135 Excessive quoted precision on discussion of a dihedral angle (88.873°).

150 Did the authors mean dissolved or desolvated?

152 Inference of stacking change made from 0.02 shift in 2theta. A fitting refinement of the data (such as the Le Bail method) should be performed to confirm that the shift is attributable to a significant change in unit cell geometry.

SI Fig 8 (PXRD) make plots bigger – cannot be seen in enough detail. Suggest each plot needs to be full page.

159 reproduce fig 2 at a larger size in SI. I agree that it shows PXRD structure generally retained, but many possibly relevant small details in patterns cannot be discerned with image so small.

General SCXRD:

For all level A and B alerts in the checkCIF report a validation response form field should be added to the CIF giving details of the likely origin of the alert and what steps were taken to mitigate any potential problems.

Use the refine_special_details field in each CIF to describe special measures taken during refinements to overcome the issues noted for all structures (e.g. reason for use of restraints, reason for omitting data, check on suggested twinning, estimates of solvent screened content). Contents of these sections should be reproduced in SI.

Where the reported Flack parameters of structure deviate from zero by $>3 \times \text{esd}$ there should be a comment or explanation about what this means for the assignment of absolute structure.

Wherever the DELU restraint has been used please explain why it has been selected rather than the updated version of this restraint: RIGU (or re-refine with RIGU instead). Use of RIGU might mitigate the need for widespread unexplained used of ISOR in many structures.

All structures should employ Gaussian absorption correction (Numerical absorption correction based on Gaussian integration over a multifaceted crystal model) (current is multiscan) as this is essential for heavy atom structures and will alleviate bad reflection horror show. If not possible – recollect.

Where a solvent mask has been applied to the data (R_{py}_I, S_{py}_I, R_{ql}_Br) and estimation of the omitted solvent content should be made and added to the unit cell contents and moiety entries.

Solvent contents are used to calculate several reported parameters in the CIF and should reflect the best estimate of the complete contents of the crystal.

R_{py}_Br

Explain the strategy for use of DFIX on PF6

Why DELU, not RIGU?

Explain why ISOR employed for DCM and PF6 (obsv prob disordered)

Solvent mask used but no estimation of contents made or included in unit cell contents (suggest

diethyl ether and a DCM)

S_py_Br

Excessive (55) omits

R_py_I

Solvent mask used but no estimation of contents made or included in unit cell contents (suggest around two DCM)

Excessive omit list of 282 reflections omitted. This suggests an underlying problem with the data, such as untreated twinning, which should be addressed or explained. The assignment of absolute structure, as indicated by the Flack parameter, cannot be taken at face value with such a large portion of data omitted. What is the outcome of refinement and absolute structure determination without omits?

S_py_I

The checkCIF report indicates that the Flack parameter is inconclusive and that a BASF/TWIN refinement should be attempted. This refinement should be attempted and the outcome and conclusions recorded in a validation response form (VRF) in the cif.

R_ql_Br

Used DELU not RIGU

Wide spread use of ISOR – why?

269 Reflections omitted – see R_py_I comments.

BASF/TWIN refinement suggested – see comment for S_py_I

S_ql_Br

Used DELU not RIGU

Wide spread use of ISOR – why?

Suggested BASF/TWIN refinement.

R_ql_Cl

Use of DELU instead of RIGU

Include solvent mask contents

S_ql_Cl

Use of DELU instead of RIGU

R_ql_I_CH2Cl2

Use of DELU instead of RIGU

Include solvent mask contents

S_ql_I_CH2Cl2

Use of DELU instead of RIGU

Include solvent mask contents

R_ql_i_100K

Include solvent mask contents

R_ql_i_150K

Suggested BASF/TWIN refinement.

Include solvent mask contents

R_ql_i_180K

Include solvent mask contents

R_qI_i_200K

Include solvent mask contents

R_qI_i_250K

Include solvent mask contents

R_qI_i_300K

Include solvent mask contents

Use of DELU instead of RIGU

R_qI_I_300K

Include solvent mask contents

Reviewer #4:

Remarks to the Author:

The work performed by Zang and coworkers demonstrated the synthesis of asymmetric Au-Cu₄ clusters with multidentate chiral NHC ligands exhibiting both the chemical stability and bright photoluminescence with CPL activity. The chiral heterometallic clusters were further applied to a CP-OLED device with a high external quantum efficiency. Some of the obtained clusters are considered to possess TADF property contributing to the extremely high PLQY over 93% in the solid state. The significance of this study may include such the remarkable records (PLQY and EQE), TADF, thermal stability, and CPL and CP-OLED activity. With all those performances comprehensively considered, I think the present results may attract the readership of Nat. Commun. The following questions maybe needed to be addressed in the revision cycles.

1. In the introduction section, the ligand design was a little introduced. How is the present ligand design rational from the viewpoint of OLED application?
2. The chiral centers of ligand are a little bit apart from the actual coordination sites. Could the authors discuss the introduction of distortion in the AuCu₄(X₄) core motifs with the result of SCXRD?
3. Page 8, from line 161: Is it possible to discuss the perspective of electronic structure of whole cluster with individual consideration of each given part in this stage? After all, the electronic structure of clusters is discussed based on the DFT calculation later.
4. Page 14, line 287: "from 5.4% and 62.8% to 93.0%." Maybe "of PLQY" is missing. This sentence says that the more efficient ISC leads to the higher PLQY value. Is this right? If my understanding is correct, the final emitting state is the S₁ for TADF. For the cluster-III, I guess the S₁ to T₂ ISC followed by the rapid IC to T₁ occurred but the RISC to S₁ was rather prohibited, resulting in the large contribution of phosphorescence with low efficiency. The similar explanation appears later (in the end of Page 16).
5. Could the effect of Au atom on the SOC be discussed quantitatively with theoretical calculations?
6. Fig. 5d-f: More detailed explanation should be provided for the result of transient absorption measurement. How is the shape of TA spectrum of III different from those of IV and V.
7. Supplementary Figure 39b: Spectra in the top of (b) should be those for R/S-NHC(qI)-AuCu₄-"Cl".
8. A comment on the comparison of gEL values with gPL in the solid state could be given.

Point-by-point response to the reviewers.

Reviewer #1

In this work, the authors developed the Au(I)-Cu(I) clusters with N-heterocyclic carbene (NHC) ligands for the efficient orange-red circular polarized luminescence (CPL). The clusters show high external quantum efficiency (EQE) up to 20.8% in the thermally activated delayed fluorescence (TADF) mechanism. The DFT and TDDFT calculations were performed to analyze the TADF mechanism and the photoluminescence as well as absorption spectra. Although *this work is of interest*, there are still many errors and ambiguous discussion points in the theoretical analysis part and therefore, I cannot recommend the publication of this paper in the present form. The authors should consider the following points before the publication.

Response: We are very thankful to **Reviewer 1** for recognition of the novelty and significance of the work and for professional suggestions, which helped to improve the rigor and reliability of our work. We addressed all the concerns and made a detailed revision of our manuscript.

(1) In principle, the spin-orbit coupling (SOC) for the intersystem crossing should be calculated at the minimum energy crossing points or the potential seam between S_1 and T_1 in the present system. However, the authors did not consider this point in the calculations; the geometries of S_1 and T_1 were used.

Response: Thanks **Reviewer 1** for the professional suggestion. As suggested, the minimum energy crossing point (MECP) structures have been calculated by TD-DFT using sobMECP Program (Lu, T. (2020) sobMECP Program. <http://sobereva.com/286> (Accessed Nov 6, 2020)) and Gaussian 16 program. The MECP was recognized when the energy gap was smaller than 0.0001 Hartree and the gradient criteria are met. The hybrid PBE0 functional in conjunction with def2-SVP basis set was used for MECP geometric optimization of **R-ql-X** ($X = \text{Cl, Br and I}$) between S_1 and T_1/T_2 . Based on the MECP structures, the coupling matrix elements (SOCME) were calculated by ORCA 5.0.0 software package based on the PBE0 functional and the DKH-def2-TZVP(-f) basis set (SARC-DKH-TZVP for Au and I atoms).

As shown in **Supplementary Fig. 41**, for **R-ql-X** ($X = \text{Cl, Br}$) system, the $\langle S_1 | \text{HSOC} | T_2 \rangle$ of S_1/T_2 MECP structure was much larger than the $\langle S_1 | \text{HSOC} | T_1 \rangle$ of S_1/T_1 MECP structure. Moreover, the energy gap between S_1/T_2 MECP and S_1 geometries was very small, which was beneficial to the inter-system crossing (ISC) process from S_1 to T_2 . This result was consistent with the conclusion of the original calculations (K_{ISC} of $S_1 \rightarrow T_2$ is faster than K_{ISC} of $S_1 \rightarrow T_1$).

Compared with **R-ql-X** ($X = \text{Cl and Br}$), **R-ql-I** exhibited a large spin-orbit coupling effect in MECP between S_1 and T_1 (110.67 cm^{-1}), and the energy gap between S_1/T_1 MECP and S_1 geometries was smaller (0.67 eV). Therefore, **R-ql-I** was more favorable in ISC and RISC between S_1 and T_1 , which facilitated the occurrence of TADF.

Therefore, the SOCME calculated from the structure of MECP was basically consistent with previous conclusions.

Supplementary Fig. 41. Calculated spin-orbit coupling matrix elements (SOCME) values for the geometries of S_1 , T_1/T_2 and minimum energy crossing point (MECP) between S_1 and T_1/T_2 for **R-ql-X** ($X = \text{Cl}$, Br and I).

(2) In p. 8, the authors discussed the origin of the SOC values by “heavy atom effect” of Au, Br, and I, which is not clear. Which atom is relevant for the spin-orbit interaction in the transition between S_1 and T_1 in fact? Please go into deeper analysis and make the origin clear. Also, one can discuss the SOC values using the El-Sayed rule by analyzing the orbitals.

Response: As suggested by **Reviewer 1**, we have calculated the basis function contribution to the hole and electron of the MECP structures and the geometries of S_1 and T_1/T_2 . The hole and electron pair distribution analyses were performed using the Multiwfn code. The input wave function for the analysis of hole and electron pair distribution was calculated using the Gaussian 16 program (**Supplementary Tables 18-28**).

Overall, Cu atoms and halide X (Cl , Br , I) ions mainly contributed to the hole of **R-ql-X** ($X = \text{Cl}$, Br , I) and **R-py-I**. For **R-ql-X** ($X = \text{Cl}$, Br , I), the Au atom contribution to holes and electrons was very small ($< 1\%$); while in **R-py-I** the Au atom contribution to holes was larger than 27% (**Supplementary Tables 26-28**). The results demonstrated

that the functional module of pyridine/quinoline with different π^* orbitals in ligands modulated the electronic structures further affecting the contribution of Au to SOC. These descriptions have been added and highlighted in the revised manuscript (Page 15).

As shown in **Supplementary Tables 18-25**, the main basis function contribution to holes and electrons was similar to the MECP structure and $S_1/T_1/T_2$ structure. In ***R-ql-X*** ($X = \text{Cl, Br, I}$) and ***R-py-I*** systems, the main contribution to holes were the D-basis functions of Cu and P-basis functions of X ($X = \text{Cl, Br, I}$) (**Supplementary Tables 18-24**). While the main contribution electron was the P-basis functions of C and N in ***R-ql-X*** ($X = \text{Cl, Br, I}$) systems, P-basis functions of Au ($> 13\%$) were the main contribution electron in ***R-py-I***. So, the excited electron of $S_1/T_1/T_2$ in ***R-ql-X*** ($X = \text{Cl, Br, I}$) came from the d atomic orbitals of Cu and p atomic orbitals of X to p atomic orbitals of the NHC^{ql} ligands ($d_{\text{Cu}} \rightarrow p_{\text{NHC}^{\text{ql}}}$, $p_{\text{Cl, Br, I}} \rightarrow p_{\text{NHC}^{\text{ql}}}$). While the excited electron of S_1/T_1 in ***R-py-I*** came from the d atomic orbitals of Cu and p atomic orbitals of I to p atomic orbitals of the Au ($d_{\text{Cu}} \rightarrow p_{\text{Au}}$, $p_{\text{I}} \rightarrow p_{\text{Au}}$) (**Supplementary Table 25**).

At the same time, we can combine the characteristics electronic structure of the excited state and El-Sayed rules to analyze the SOC strength between these two states. For the ***R-ql-X*** ($X = \text{Cl, Br and I}$) in S_1 states, they all had similar charge-transfer (CT) excited states⁵⁹ due to larger D values ($> 4.0 \text{ \AA}$) and smaller Sr values (< 0.21). While in the T_1 states of ***R-ql-X*** ($X = \text{Cl, Br and I}$) with the gradually increased heavy-atom effect from Cl to Br to I, the D values gradually decreased (1.93 \AA , 1.72 \AA , 0.77 \AA) and the Sr values gradually increased (0.71 , 0.75 , 0.86). Yet, the optimized T_2 states of ***R-ql-X*** ($X = \text{Cl, Br}$) showed more local excitation (³LE) character ($D < 0.60 \text{ \AA}$) than that of T_1 ($D > 1.70 \text{ \AA}$), which also had the higher $\langle S_1 | \text{HSOC} | T_2 \rangle$ than $\langle S_1 | \text{HSOC} | T_1 \rangle$ values. In contrast, the T_1 states in ***R-ql-I*** showed a more ³LE character, which also had bigger $\langle S_1 | \text{HSOC} | T_1 \rangle$ values than that of ***R-ql-X*** ($X = \text{Cl, Br}$). The values of SOCME can also be qualitatively estimated through El-Sayed's rules⁶⁰, namely, the SOCME was relatively large if the transition between singlet and triplet states involves a change of orbital type. The respectively calculated SOCME based on the geometries of S_1 , T_1/T_2 (**Supplementary Table 21**) and the structure of the minimum energy crossing point (MECP) (**Supplementary Fig. 41 and Tables 22-24**) had a consistent trend, which both conformed to the El-Sayed's rule. For ***R-py-I***, the NHC^{py} ligands rendered the resulting ***R-py-I*** complexes had CT excited states T_1 , and more local excitation (¹LE) character in S_1 . At the same time, the atomic orbitals contribution of S_1 and T_1 had big differences (**Supplementary Table 25**). According to El-Sayed rules, ***R-py-I*** would have a large $\langle S_1 | \text{HSOC} | T_1 \rangle$ value.

Supplementary Table 18. Calculated basis function contribution to hole and electron of S₁ and T₂ for the S₁, T₂ geometries of **R-ql-Cl**, respectively.

	Atom	Shell	Type	Hole	Atom	Shell	Type	Electron
S ₁	2(Cu)	21	d _{xz}	34.69%	28(C)	186	p _x	9.36%
	46(Cu)	299	d _{xz}	22.37%	9(N)	72	p _x	7.71%
	2(Cu)	22	d _{xz}	4.80%	28(C)	187	p _x	7.27%
	47(Cl)	308	p _z	3.08%	16(C)	114	p _x	6.70%
	5(Cl)	49	p _x	2.98%	9(N)	73	p _x	4.36%
	5(Cl)	49	p _z	2.63%	28(C)	186	p _z	3.97%
	2(Cu)	21	d _{z²}	2.62%	9(N)	72	p _z	3.94%
	46(Cu)	300	d _{xz}	2.57%	16(C)	115	p _x	3.65%
	47(Cl)	308	p _x	2.52%	43(C)	276	p _x	3.20%
	47(Cl)	307	p _z	2.23%	22(C)	150	p _x	3.15%
T ₂	3(Cu)	32	d _{z²}	16.28%	79(C)	506	p _x	4.78%
	48(Cu)	318	d _{z²}	16.17%	42(C)	270	p _x	4.74%
	3(Cu)	32	d _{x²-y²}	3.74%	6(N)	54	p _x	4.06%
	48(Cu)	318	d _{x²-y²}	3.72%	53(N)	350	p _x	4.05%
	4(Cl)	41	p _x	2.88%	79(C)	507	p _x	3.71%
	49(Cl)	327	p _x	2.87%	42(C)	271	p _x	3.67%
	3(Cu)	33	d _{z²}	2.54%	26(C)	174	p _x	3.32%
	48(Cu)	319	d _{z²}	2.52%	61(C)	398	p _x	3.28%
	4(Cl)	40	p _x	2.09%	6(N)	55	p _x	2.26%
	49(Cl)	326	p _x	2.09%	53(N)	351	p _x	2.26%

Supplementary Table 19. Calculated basis function contribution to hole and electron of S₁ and T₁ for the S₁, T₁ geometries of **R-ql-Br**.

	Atom	Shell	Type	Hole	Atom	Shell	Type	Electron
S ₁	4(Cu)	45	d _{xz}	32.99%	55(C)	299	p _x	9.49%
	75(Cu)	402	d _{xz}	21.00%	6(N)	62	p _x	7.72%
	4(Cu)	46	d _{xz}	4.32%	55(C)	300	p _x	7.36%
	76(Br)	413	p _x	3.80%	29(C)	176	p _x	6.78%
	3(Br)	33	p _z	3.48%	6(N)	63	p _x	4.38%
	3(Br)	33	p _x	3.41%	55(C)	299	p _z	4.06%
	76(Br)	412	p _x	3.19%	6(N)	62	p _z	4.01%
	3(Br)	32	p _x	3.01%	29(C)	177	p _x	3.69%
	3(Br)	32	p _z	2.94%	63(C)	335	p _x	3.20%
	76(Br)	413	p _z	2.93%	40(C)	230	p _x	3.14%
T ₁	4(Cu)	45	d _{xy}	26.36%	55(C)	299	p _y	3.71%
	75(Cu)	402	d _{xy}	26.36%	113(C)	618	p _y	3.71%
	3(Br)	33	p _y	5.56%	55(C)	299	p _x	3.15%
	76(Br)	413	p _y	5.56%	113(C)	618	p _x	3.15%
	3(Br)	32	p _y	4.79%	6(N)	62	p _y	3.08%
	76(Br)	412	p _y	4.79%	81(N)	450	p _y	3.08%
	4(Cu)	46	d _{xy}	3.19%	55(C)	300	p _y	2.92%
	75(Cu)	403	d _{xy}	3.19%	113(C)	619	p _y	2.92%
	6(N)	62	p _y	1.39%	6(N)	62	p _x	2.91%
	81(N)	450	p _y	1.39%	81(N)	450	p _x	2.91%

Supplementary Table 20. Calculated basis function contribution to hole and electron of S₁ and T₁ for the S₁, T₁ geometries of **R-ql-I**.

	Atom	Shell	Type	Hole	Atom	Shell	Type	Electron
S ₁	9(Cu)	94	d _{xz}	28.22%	63(C)	358	p _x	9.40%
	8(Cu)	83	d _{xz}	17.98%	14(N)	124	p _x	7.48%
	5(I)	50	p _x	5.20%	63(C)	359	p _x	7.31%
	4(I)	39	p _x	5.14%	80(C)	439	p _x	6.62%
	5(I)	49	p _x	5.06%	14(N)	125	p _x	4.26%
	4(I)	40	p _x	5.02%	63(C)	358	p _z	4.18%
	4(I)	40	p _z	4.02%	14(N)	124	p _z	4.06%
	4(I)	39	p _z	3.94%	80(C)	440	p _x	3.58%
	5(I)	49	p _z	3.42%	102(C)	544	p _x	3.21%
	9(Cu)	95	d _{xz}	3.42%	78(C)	430	p _x	3.16%
T ₁	7(Cu)	72	d _{z²}	9.22%	134(C)	691	p _x	10.81%
	38(C)	235	p _x	7.59%	134(C)	692	p _x	8.22%
	95(C)	511	p _x	6.81%	16(N)	136	p _x	6.94%
	16(N)	136	p _x	6.50%	72(C)	400	p _x	5.98%
	134(C)	691	p _x	6.03%	95(C)	511	p _x	5.61%
	38(C)	236	p _x	5.41%	38(C)	235	p _x	5.00%
	95(C)	512	p _x	4.54%	95(C)	512	p _x	4.05%
	134(C)	692	p _x	3.91%	38(C)	236	p _x	4.00%
	16(N)	137	p _x	3.74%	16(N)	137	p _x	3.92%
	84(C)	460	p _x	2.99%	89(C)	484	p _x	3.20%

Supplementary Table 21. Computed SOCME values of **R-ql-X** (X = Cl, Br and I) nanoclusters at their S₁ optimized structures.

	$\langle S_1 HSO T_1 \rangle$	$\langle S_1 HSO T_2 \rangle$	$\langle S_1 HSO T_3 \rangle$	$\langle S_1 HSO T_4 \rangle$	$\langle S_1 HSO T_5 \rangle$
R-ql-Cl	45.25	271.34	3.2101	0.7559	0.4555
R-ql-Br	14.05	95.92	19.20	1.57	1.59
R-ql-I	70.07	234.26	37.88	5.64	1.35
R-py-I	96.03	505.05	870.85	827.03	198.15

Supplementary Table 22. Calculated basis function contribution to hole and electron of S₁ and T₂ for the **MECP** between S₁ and T₁ geometries of **R-ql-Cl**.

	Atom	Shell	Type	Hole	Atom	Shell	Type	Electron
S ₁	2(Cu)	21	d _{xz}	35.08%	28(C)	186	p _x	9.41%
	46(Cu)	299	d _{xz}	22.07%	9(N)	72	p _x	7.72%
	2(Cu)	22	d _{xz}	4.87%	28(C)	187	p _x	7.32%
	47(Cl)	308	p _z	3.06%	16(C)	114	p _x	6.67%
	5(Cl)	49	p _x	2.98%	9(N)	73	p _x	4.36%
	2(Cu)	21	d _{z²}	2.69%	28(C)	186	p _z	4.00%
	5(Cl)	49	p _z	2.57%	9(N)	72	p _z	3.97%
	47(Cl)	308	p _x	2.55%	16(C)	115	p _x	3.63%
	46(Cu)	300	d _{xz}	2.53%	43(C)	276	p _x	3.19%
	47(Cl)	307	p _z	2.22%	22(C)	150	p _x	3.12%
T ₂	2(Cu)	21	d _{z²}	18.76%	28(C)	186	p _x	9.45%
	2(Cu)	21	d _{x²-y²}	6.11%	28(C)	187	p _x	7.32%
	28(C)	186	p _x	4.50%	9(N)	72	p _x	7.30%
	22(C)	150	p _x	3.31%	16(C)	114	p _x	6.13%
	43(C)	276	p _x	3.21%	9(N)	73	p _x	4.09%
	9(N)	72	p _x	3.12%	28(C)	186	p _z	4.01%
	2(Cu)	22	d _{z²}	3.09%	9(N)	72	p _z	3.69%
	28(C)	187	p _x	3.02%	43(C)	276	p _x	3.67%
	2(Cu)	21	d _{xz}	2.78%	22(C)	150	p _x	3.37%
	22(C)	151	p _x	2.61%	16(C)	115	p _x	3.29%

Supplementary Table 23. Calculated basis function contribution to hole and electron of S_1 and T_2 for the **MECP** between S_1 and T_1 of ***R-ql-Br*** geometries.

	Atom	Shell	Type	Hole	Atom	Shell	Type	Electron
S_1	4(Cu)	45	d_{xy}	26.81%	55(C)	299	p_y	4.52%
	75(Cu)	402	d_{xy}	26.81%	113(C)	618	p_y	4.52%
	3(Br)	33	p_x	4.91%	29(C)	176	p_y	4.14%
	76(Br)	413	p_x	4.91%	89(C)	498	p_y	4.14%
	3(Br)	32	p_x	4.16%	6(N)	62	p_y	3.83%
	76(Br)	412	p_x	4.16%	81(N)	450	p_y	3.83%
	4(Cu)	46	d_{xy}	3.20%	55(C)	300	p_y	3.53%
	75(Cu)	403	d_{xy}	3.20%	113(C)	619	p_y	3.53%
	3(Br)	33	p_y	2.15%	1(Au)	9	p_x	2.49%
	76(Br)	413	p_y	2.15%	29(C)	177	p_y	2.49%
T_2	4(Cu)	45	d_{xy}	26.20%	55(C)	299	p_y	4.92%
	75(Cu)	402	d_{xy}	26.20%	113(C)	618	p_y	4.92%
	3(Br)	33	p_x	4.68%	29(C)	176	p_y	4.30%
	76(Br)	413	p_x	4.68%	89(C)	498	p_y	4.30%
	3(Br)	32	p_x	3.99%	6(N)	62	p_y	4.16%
	76(Br)	412	p_x	3.99%	81(N)	450	p_y	4.16%
	4(Cu)	46	d_{xy}	3.09%	55(C)	300	p_y	3.82%
	75(Cu)	403	d_{xy}	3.09%	113(C)	619	p_y	3.82%
	3(Br)	33	p_y	2.17%	29(C)	177	p_y	2.46%
	76(Br)	413	p_y	2.17%	89(C)	499	p_y	2.46%

Supplementary Table 24. Calculated basis function contribution to hole and electron of S_1 and T_1 for the **MECP** between S_1 and T_1 of ***R-ql-I*** geometries.

	Atom	Shell	Type	Hole	Atom	Shell	Type	Electron
S_1	8(Cu)	83	d_{xz}	22.39%	63(C)	358	p_x	8.97%
	9(Cu)	94	d_{xz}	17.12%	63(C)	359	p_x	6.77%
	5(I)	50	p_x	6.40%	14(N)	124	p_x	5.69%
	5(I)	49	p_x	6.02%	80(C)	439	p_x	4.74%
	4(I)	40	p_z	5.72%	63(C)	358	p_z	3.79%
	4(I)	39	p_z	5.49%	78(C)	430	p_x	3.65%
	4(I)	39	p_x	3.43%	102(C)	544	p_x	3.63%
	4(I)	40	p_x	3.16%	14(N)	124	p_z	3.59%
	9(Cu)	95	d_{xz}	2.25%	14(N)	125	p_x	3.29%
	8(Cu)	84	d_{xz}	2.24%	80(C)	439	p_z	2.87%
T_1	8(Cu)	83	d_{xz}	21.63%	63(C)	358	p_x	8.97%
	9(Cu)	94	d_{xz}	18.14%	63(C)	359	p_x	6.75%
	5(I)	50	p_x	6.27%	14(N)	124	p_x	5.74%
	5(I)	49	p_x	5.90%	80(C)	439	p_x	4.81%
	4(I)	40	p_z	5.62%	63(C)	358	p_z	3.80%
	4(I)	39	p_z	5.40%	14(N)	124	p_z	3.65%
	4(I)	39	p_x	3.47%	78(C)	430	p_x	3.58%
	4(I)	40	p_x	3.17%	102(C)	544	p_x	3.58%
	9(Cu)	95	d_{xz}	2.41%	14(N)	125	p_x	3.34%
	8(Cu)	84	d_{xz}	2.14%	80(C)	439	p_z	2.90%

Supplementary Table 25. Calculated basis function contribution to hole and electron of S₁ and T₁ for the S₁, T₁ geometries of **R-py-I**.

	Atom	Shell	Type	Hole	Atom	Shell	Type	Electron
S ₁	6(Cu)	61	d _{xz}	16.58%	1(Au)	9	p _y	26.44%
	7(Cu)	72	d _{xz}	16.58%	24(C)	185	p _y	4.12%
	2(I)	20	p _x	8.00%	26(C)	197	p _y	4.12%
	3(I)	30	p _x	8.00%	24(C)	184	p _y	3.73%
	2(I)	19	p _x	7.92%	26(C)	196	p _y	3.73%
	3(I)	29	p _x	7.92%	7(Cu)	71	p _y	2.54%
	6(Cu)	61	d _{z²}	5.69%	6(Cu)	60	p _y	2.53%
	7(Cu)	72	d _{z²}	5.69%	15(N)	130	p _y	2.27%
	6(Cu)	62	d _{xz}	2.09%	17(N)	142	p _y	2.27%
	7(Cu)	73	d _{xz}	2.09%	12(N)	112	p _y	2.26%
T ₁	2(I)	19	p _x	10.65%	1(Au)	9	p _y	25.62%
	3(I)	29	p _x	10.64%	24(C)	185	p _y	3.91%
	2(I)	20	p _x	10.19%	26(C)	197	p _y	3.91%
	3(I)	30	p _x	10.18%	26(C)	196	p _y	3.37%
	6(Cu)	61	d _{z²}	6.20%	24(C)	184	p _y	3.36%
	7(Cu)	72	d _{z²}	6.19%	6(Cu)	60	p _y	2.49%
	3(I)	29	p _z	4.12%	7(Cu)	71	p _y	2.49%
	2(I)	19	p _z	4.11%	7(Cu)	67	s	2.27%
	3(I)	30	p _z	3.92%	6(Cu)	56	s	2.26%
	2(I)	20	p _z	3.91%	12(N)	112	p _y	2.26%

Supplementary Table 26. Au atom contribution to holes and electrons of S₁, T₁, T₂ states of *R-ql-Cl*, *R-ql-Br*, *R-ql-I* and *R-py-I*.

geometries	Au atom	Hole	Electron	Overlap	Diff.
R-ql-Cl	S ₁	0.27 %	0.27 %	0.27 %	0.39 %
	T ₁	0.03 %	0.29 %	0.10 %	0.26 %
	T ₂	0.05 %	0.26 %	0.26 %	0.21%
R-ql-Br	S ₁	0.13 %	0.60 %	0.28 %	0.46 %
	T ₁	0.07 %	0.27 %	0.14 %	0.20 %
	T ₂	0.08 %	0.18 %	0.12 %	0.10 %
R-ql-I	S ₁	0.05 %	0.52 %	0.15 %	0.47 %
	T ₁	0.16 %	0.11 %	0.13 %	-0.05 %
R-py-I	S ₁	0.64 %	28.38 %	4.28 %	27.74 %
	T ₁	0.37 %	27.42 %	3.17 %	27.05 %

Supplementary Table 27. Cu atom contribution to holes and electron of S₁, T₁, T₂ states of *R-ql-Cl*, *R-ql-Br*, *R-ql-I*, and *R-py-I*.

geometries	Cu atom	Hole	Electron	Overlap	Diff.
R-ql-Cl	S ₁	71.02%	2.42%	11.45%	-68.61%
	T ₁	47.03%	3.98%	13.66%	-43.05%
	T ₂	52.85%	2.44%	11.39%	-50.40%
R-ql-Br	S ₁	65.01%	2.19%	10.34%	-62.82%
	T ₁	37.42%	3.06%	10.58%	-34.35%
	T ₂	62.98%	1.82%	10.64%	-61.20%
R-ql-I	S ₁	54.78%	1.90%	8.75%	-52.87%
	T ₁	14.04%	1.65%	4.81%	-12.38%
R-py-I	S ₁	57.11%	18.66%	27.48%	-38.45%
	T ₁	36.19%	20.48%	21.87%	-15.71%

Supplementary Table 28. Halid X (Cl, Br, I) ions contribution to holes and electron of S₁, T₁, T₂ states of *R-ql-Cl*, *R-ql-Br*, *R-ql-I*, and *R-py-I*.

geometries	X (Cl, Br, I)	Hole	Electron	Overlap	Diff.
R-ql-Cl	S ₁	19.89%	0.53%	3.17%	-19.37%
	T ₁	12.40%	0.92%	3.33%	-11.49%
	T ₂	15.37%	0.49%	2.70%	-14.86%
R-ql-Br	S ₁	25.89%	0.61%	3.83%	-25.28%
	T ₁	12.49%	0.84%	3.20%	-11.65%
	T ₂	24.58%	0.38%	2.94%	-24.22%
R-ql-I	S ₁	36.28%	0.73%	4.92%	-35.55%
	T ₁	5.81%	0.59%	1.82%	-5.22%
R-py-I	S ₁	34.82%	17.71%	19.70%	-17.12%
	T ₁	60.10%	18.04%	24.71%	-42.06%

(3) There are still many errors and defects in the present manuscript. For example, p. 13, “0.95 eV” must be “0.85 eV” (6.03-5.18) from the values in Figure 1. Also, the HOMO level seems to be drawn in the same level for NHC^{py} and NHC^{ql}. All these levels should be shown with the absolute values. “Fig. 1B” should be “Fig. 1b”. The word “concentrated” should be “localized” in the standard expression.

Response: As suggested by **Reviewer 1**, HOMO and LUMO levels for NHC^{py}-H and NHC^{ql}-H are replotted and marked with absolute values in **Supplementary Fig. 31**.

The correct “0.85 eV” number has been added in the revised manuscript.

The word “concentrated” has been corrected to “localized”. We double-checked the overall manuscript thoroughly and corrected these mistakes in the revised manuscript.

Supplementary Figure 31. HOMO-LUMO gap of *R/S*-NHC^{py}-H and *R/S*-NHC^{ql}-H.

(4) The calculated absorption spectra in Supplementary Figure 24 are largely deviated from the experimental UV-Vis spectra. This difference should be clearly explained with reasons. The agreement of the emission energies from S₁ and T₁ seems to be accidental.

Response: The original deviations were caused by the absence of a solvent model for TD-DFT calculations, and were corrected by incorporating SMD solvent model (dichloromethane), and increasing the number of excited states from 200 to 300.

As suggested by **Reviewer 1**, the theoretical UV-Vis spectra of ***R*-py-X** (X = Br and I) and ***R*-ql-X** (X = Cl, Br and I) were re-calculated under PBE0/def2SVP level, and the results showed that they were basically consistent with the experimental UV-vis absorption spectra, as shown in the **Supplementary Figs. 33-34**.

Supplementary Figure 33. Experimental electronic absorption spectra of ***R-py-Br*** (green) and ***R-py-I*** (red) in CH_2Cl_2 (1×10^{-5} mol/L) compared to the calculated spectra (black). Light blue bars show the individual transitions (delta function-like peaks showing the relative oscillator strengths).

Supplementary Figure 34. Experimental electronic absorption spectra of ***R-ql-CI*** (blue), ***R-ql-Br*** (green), and ***R-ql-I*** (red) in CH_2Cl_2 (1×10^{-5} mol/L) compared to the calculated spectra (black). Light blue bars show the individual transitions (delta function-like peaks showing the relative oscillator strengths).

(5) The authors discussed the intracuster charge transfer. The orbital composition analysis should make this much clearer.

Response: As suggested by **Reviewer 1**, the atom orbital composition analysis has been added in **Supplementary Table 18-25**, with more details in the comment (2).

(6) In p. 14, the number of 2.59×10^{-144} seems to be strange when compared to the other values. Please check the value and this value may have cancellation of significant digits.

Response: As suggested by **Reviewer 1**, we double-checked the value of the calculated data and determined that it was 2.59×10^{-144} . To avoid confusion for readers, the value should be "0" after considering the significant digit, and corrected in the revised **Supplementary Table 29**.

(7) I do not understand the values of 5.4% and 62.8% to 93% given in p. 14. Please make this clear.

Response: Following this suggestion, we have corrected “causing a dramatic increase from 5.4% and 62.8% to 93.0%” to “the faster k_{ISC} and k_{RISC} supported the higher PLQY in the order of 5.4% (**R-ql-Cl**) < 62.8% (**R-ql-Br**) < 93.0% (**R-ql-I**)” (**Page 17**).

(8) In p. 15 and Table S18, the value of (23.4.26) is wrong and not clear. Because R-NHC^{ql}-AuCu₄-I lacks T₂, I do not understand this discussion. Also, the values of T₁(1.713 eV) and LE(1.771 eV) cannot be found in any table, please make this part clearer.

Response: As suggested by **Reviewer 1**, the “23.4.26” should be “234.26”, and the $\langle S_1 | HSO | T_2 \rangle$ of **R-ql-I** was calculated based on the optimized S₁ structure. The calculated SOCME values of **R-ql-X** (X = Cl, Br and I) nanoclusters at their optimized S₁ structures have been added in **Supplementary Table 21**.

The **reviewer's** concern is justified. Since T₂ was lacking **R-ql-I** and its significance for the subsequent discussion is not obvious, the “234.26” has been removed from the revised manuscript.

Supplementary Table 21. Computed SOCME values of **R-ql-X** (X = Cl, Br and I) and **R-py-I** nanoclusters at their optimized S₁ structures.

	$\langle S_1 HSO T_1 \rangle$	$\langle S_1 HSO T_2 \rangle$	$\langle S_1 HSO T_3 \rangle$	$\langle S_1 HSO T_4 \rangle$	$\langle S_1 HSO T_5 \rangle$
R-ql-Cl	45.25	271.34	3.21	0.76	0.46
R-ql-Br	14.05	95.92	19.20	1.57	1.59
R-ql-I	70.07	234.26	37.88	5.64	1.35
R-py-I	96.03	505.05	870.85	827.03	198.15

(9) It is very difficult to follow the discussion in particular, the theoretical part. The discussion is sometimes for R-NHC-AuCu₄-Cl, -Br, -I and sometimes only for R-NHC-AuCu₄-I. Please write down the discussion much clearer.

Response: As suggested by **Reviewer 1**, with the new calculation results, we reorganized and rewrote the discussion of the theoretical part in the revised manuscript (**Pages 13-19**).

(10) The explanation of D and S_r values in p. 15 will help the understanding the general readers, although the authors cited ref. 53.

Response: As suggested by **Reviewer 1**, the explanations of D and S_r values have been added in supporting information (Page S10).

The S_r index was defined as the full space integration of a function ($S_r(r)$) describing the overlap between electron and hole distributions, formulated as $S_r(r) \text{ index} = \int \sqrt{\rho^{hole}(r)\rho^{electron}(r)}dr$, where $\rho^{hole}(r)$ and $\rho^{electron}(r)$ are the hole and electron distribution. D index is the distance between a hole and an electron center of mass, formulated as $D \text{ index} = \sqrt{(D_x)^2 + (D_y)^2 + (D_z)^2}$.

(11) I do not understand the word “triplet effects” in p. 16. The authors should explain what this means in more detail.

Response: As suggested by **Reviewer 1**, this was the wrong description. We have rewritten this sentence in the revised manuscript (Page 15).

“the LE characteristics in the triplet state (T_1) in synergy with the CT characteristics for the singlet state (S_1) could be in favor of activating TADF emission”.

(12) In p. 19, Supplementary Figs. 24-25 are absorption spectra of these clusters and not related to the chiral electronic transitions.

Response: As suggested by **Reviewer 1**, absorption spectra have been related to the chiral electronic transitions (Supplementary Figs. 42-45). The responding explanations have been added to the revised manuscript (Pages 20-21) and the revised supporting information (Pages S65-S67).

In CH_2Cl_2 solution (1×10^{-5} mol/L) at room temperature, these enantiomeric pairs of clusters all gave rise to new CD bands in the range of 340–400 nm, relative to those of the precursor **R/S-[Au(NHC^{py/ql})₂PF₆** with CD signal peaks below 350 nm (Supplementary Figs. 42-43). CD responses were basically consistent with the UV–vis absorption spectra, and those above 340 nm generally involved the electronic transitions of orbitals of cluster molecules (Supplementary Figs. 33-34, 42-47 and Tables 13-17). For example, **R-ql-I** cluster had CD band with peaks at 377 nm, which mainly originated from transitions from HOMO-4 (Cu₄-dominated) to LUMO+3 (quinoline-dominated) and LUMO+4 (Cu₄-dominated). For those CD signals below 340 nm, the chiral electronic transitions partially overlapped with ones in the chiral precursor (Supplementary Figs. 43, 45, 47). These findings suggested the chiroptical activities of the whole cluster molecule.

For **R-py-Br**, the signals in the low-energy regions of 340–400 nm are related to the absorption peak at 363 nm corresponding to the CD transition at 357 nm (peak a), which was ascribed to the transitions from HOMO-4 to LUMO (Supplementary Fig. 44a). The UV absorption band at 313 nm corresponded to the CD transition at 311 nm (peak b), which was ascribed to the transitions from HOMO-4 to LUMO+1, and HOMO-3 to LUMO+2. And the UV absorption band centered at 271 nm was

completely consistent with the CD transition at 265 nm (peak c), which was derived from HOMO-4 to LUMO+8. The UV band at 249 nm corresponded to the CD transition at 251 nm (peak d) and was mainly attributed to the HOMO-21 to LUMO transitions. The UV absorption band at 227 nm corresponded to the CD transition at 228 nm (peak e), which was ascribed to the transitions from HOMO-22 to LUMO.

For **R-py-I**, the signals in the low-energy regions of 340–400 nm are related to the absorption peak at 370 nm corresponding to the CD transition at 359 nm (peak a), which was ascribed to the transitions from HOMO-4 to LUMO (Supplementary Fig. 44b). The UV absorption band at 317 nm corresponded to the CD transition at 326 nm (peak b), which was ascribed to the transitions from HOMO-3 to LUMO+2, and HOMO-2 to LUMO+4. And the UV absorption band centered at 271 nm ascribed to the CD transition at 269 nm (peak c), which was derived from HOMO-3 to LUMO+7, and HOMO-2 to LUMO+8. The UV band at 250 nm corresponded to the CD transition at 255 nm (peak d) and was mainly attributed to the HOMO-1 to LUMO+14, LUMO+19, and HOMO to LUMO+15 transitions. The UV absorption band at 229 nm corresponded to the CD transition at 234 nm (peak e), which was ascribed to the transitions from HOMO-4 to LUMO+14 and LUMO+19.

For **R-ql-Cl**, the signals in the low-energy regions of 320–400 nm are related to the absorption peak at 362 nm corresponding to the CD transition at 366 nm (peak a), which was ascribed to the transitions from HOMO-4 to LUMO+4 (Supplementary Fig. 45a). The UV–Vis absorption band at 270 nm corresponded to the CD transition at 275 nm (peak b), which was ascribed to the transitions from HOMO-20 to LUMO+3, and HOMO-19 to LUMO+1. The UV band at 235 nm corresponded to the CD transition at 242 nm (peak c) and was mainly attributed to the HOMO-29 to LUMO+4, and HOMO-23 to LUMO+4 transitions.

For **R-ql-Br**, the signals in the low-energy regions of 340–400 nm are related to the absorption peak at 357 nm corresponding to the CD transition at 371 nm (peak a), which was ascribed to the transitions from HOMO-4 to LUMO +4 (Supplementary Fig. 45b). The UV absorption band at 323 nm corresponded to the CD transition at 332 nm (peak b), which was ascribed to the transitions from HOMO-6 to LUMO, and HOMO-5 to LUMO+2. And the UV absorption band centered at 270 nm ascribed to the CD transition at 280 nm (peak c), which was derived from HOMO-22 to LUMO, HOMO-21 to LUMO+3, and HOMO-19 to LUMO. The UV band at 235 nm corresponded to the CD transition at 260 nm (peak d) and was mainly attributed to the HOMO-24, HOMO-14 to LUMO+4 transitions.

Supplementary Figure 42. CD (left) and UV (purple line; right) spectra of *R/S*-[Au(NHC^{py})₂]PF₆ and *R/S*-py-X (X = Br and I) in CH₂Cl₂ under ambient conditions.

Supplementary Figure 43. CD (left) and UV (purple line; right) spectra of *R/S*-[Au(NHC^{ql})₂]PF₆ and *R/S*-ql-X (X = Cl, Br and I) in CH₂Cl₂ under ambient conditions.

Supplementary Figure 44. Experimental CD spectra of *R-py-Br* (a) and *R-py-I* (b) in CH_2Cl_2 compared to the calculated spectra.

Supplementary Figure 45. Experimental CD spectra of *R-ql-Cl* (a), *R-ql-Br* (b), and *R-ql-I* (c) in CH_2Cl_2 compared to the calculated spectra.

(13) The observed rotary strengths of CPL were very small in Figure 6 and the g_{lum} values were 10^{-4} - 10^{-3} . The g -value and rotary strength of these clusters can be easily calculated. I encourage the authors to show the calculated values and compare with the experimental values.

Response: As suggested by **Reviewer 1**, we calculated the rotation intensity of *R-ql-X* ($X = \text{Cl}, \text{Br}$ and I) and *R-py-X* ($X = \text{Br}$ and I), and made a comparison between calculated values and experimental values (**Supplementary Table 32**). Although the calculated values were larger than the experimental values, they still were at the same magnitude, probably because of the inevitable scattering effects in the solid state and the solvation effect in the solution during experiments.

Supplementary Table 32. Experimental g_{lum} values of ***R-ql-X*** ($X = Cl, Br$ and I) and ***R-py-X*** ($X = Br$ and I) in the solid and solution states; Theoretical g_{lum} values and rotatory strength of ***R-ql-X*** ($X = Cl, Br$ and I) and ***R-py-X*** ($X = Br$ and I).

Cluster	Exp-Solid ($\times 10^{-3}$)	Exp-Solution ($\times 10^{-3}$)	Calculated ($\times 10^{-3}$)	Rotatory strength (10^{-40} esu cm erg G^{-1})
R-py-Br	1.5	0.4	7.3	0.87
R-py-I	0.6	0.6	1.4	8.36
R-ql-Cl	0.7	0.9	8.3	1.02
R-ql-Br	1.5	0.5	2.8	0.49
R-ql-I	0.9	0.4	3.4	0.41

Reviewer #2

In this manuscript, Zang et al. reported five enantiopure pairs of AuICu₄ clusters possessing interesting chiroptic properties for Circularly-Polarized OLED (CP-OLED) applications. The synthesis of the AuICu₄ clusters was carried out in the same way as the previous literature (ref 44 of this study), using CuX (X = Cl, Br, I) instead of using [Cu(MeCN)₄]PF₆. *The authors address the important issue of studying in detail, both experimentally and theoretically, the effects of ligands (both NHC and secondary ligand halides) on the luminosity, chirality, and electronic structure of these clusters. An interesting aspect of this research is its application to CP-OLEDs based on strong photoluminescence with a quantum yield of up to 93% in the solid state and an external quantum efficacy as high as 20.8%.* In order to finally be published in Nature Communications, the following issues should be properly resolved.

Response: We appreciate **Reviewer 2's** positive comments very much. We have addressed all concerns.

Abstract

1. On page 2, lines 20,21. "clusters without heavier metals such as Ru, Ir, Os and Pt...". Au is "heavier" than these metals, so please reconsider this expression.

Response: As suggested by **Reviewer 2**, we have revised the expression in the revised manuscript.

“Bright and efficient chiral coinage metal clusters show promise....”

Introduction

2. On page 5, lines 80, 84: "pyridine" => "pyridine/quinoline".

Response: Following this suggestion, we have corrected these mistakes (Page 5).

3. On page 5, line 87: "cationic *R/S*-NHC^{py} and *R/S*-NHC^{ql} ligands". First, the ligands are neutral, as noted in line 62. Second, this is the first time the authors mention the abbreviation of the ligand, it is necessary to give full names or formulas.

Response: As suggested by **Reviewer 2**, the first appearance of the ligands in the revised manuscript has been corrected and supplemented with the full molecular formula.

We have corrected “cationic *R/S*-NHC^{py}-PF₆ and *R/S*-NHC^{ql}-PF₆” to “Two pairs of enantiomers of tridentate NHC ligands were designed and synthesized (Fig. 1a): (4*R*/4*S*, 5*R*/5*S*)-N, N'-di(chloromethyl)pyridine)-4,5-diphenyl-4,5-dihydro-imidazolium hexafluorophosphate (denoted as *R/S*-NHC^{py}-H·PF₆) and (4*R*/4*S*, 5*R*/5*S*)-N, N'-2-(chloromethyl)quinoline-4,5-diphenyl-4,5-dihydro-imidazolium hexafluorophosphate (denoted as *R/S*-NHC^{ql}-H·PF₆) ligands” (Page 5).

Synthesis and structures

4. On page 5, line 96: "*R/S*-NHC^{py}-PF₆ and *R/S*-NHC^{ql}-PF₆". Since the pre-ligands are protonated, it is more appropriate to name them “*R/S*-NHC^{py}-H·PF₆ and *R/S*-NHC^{ql}-H·PF₆”.

Response: As suggested by **Reviewer 2**, we have corrected “*R/S*-NHC^{py}-PF₆ and *R/S*-

NHC^{q1}-PF₆” to “*R/S*-NHC^{py}-H·PF₆ and *R/S*-NHC^{q1}-H·PF₆” in the revised manuscript (Page 5).

5. In the Supplementary Information (page 5, line 98). The "yield: 72%" is misleading, and therefore the respective yields of *R/S*-[Au(NHC^{py})₂]PF₆ and *R/S*-[Au(NHC^{q1})₂]PF₆ should be addressed. Also, NMR data for the new clusters including *R/S*-[Au(NHC^{py})₂]PF₆, *R/S*-[Au(NHC^{q1})₂]PF₆, *R/S*-NHC^{q1}-AuCu₄-X (X = Cl, Br and I), and *R/S*-NHC^{py}-AuCu₄-X (X = Br and I) are missing.

Response: As suggested by **Reviewer 2**, we have conducted yield calculations for each precursor, respectively, and the results have been added to the supporting information (Page 5).

In addition, ¹H-NMR data for all clusters were tested and have been included in the revised manuscript, and the spectra have been added in **Supplementary Figs. 10-14**.

Supplementary Figure 10. ¹H-NMR spectra of (a) *R*-py-Br and (b) *S*-py-Br (600 MHz, CD₂Cl₂).

Supplementary Figure 11. $^1\text{H-NMR}$ spectra of (a) *R*-py-I and (b) *S*-py-I (600 MHz, CD_2Cl_2).

Supplementary Figure 12. $^1\text{H-NMR}$ spectra of (a) *R*-ql-Cl and (b) *S*-ql-Cl (600 MHz, CD_2Cl_2).

Supplementary Figure 13. $^1\text{H-NMR}$ spectra of (a) *R*-ql-Br and (b) *S*-ql-Br (600 MHz, CD_2Cl_2).

Supplementary Figure 14. $^1\text{H-NMR}$ spectra of (a) *R*-ql-I and (b) *S*-ql-I (600 MHz, CD_2Cl_2).

6. On page 7, line 140. For Cu(I)...Cu(I) interactions, rather than using "copperphilic interactions", the term of "cuprophilic interactions" is commonly used in the reported literature. See references, for example, Chem. Commun., 2016, 52, 2932-2935; J. Am. Chem. Soc. 2021, 143, 3808-3816.

Response: As suggested by **Reviewer 2**, we have corrected "copperphilic interactions" to "cuprophilic interactions" and the relevant references have been included in the reference list (Ref.51-52).

7. On page 8, line 154, "... , yet the cluster molecules were not destroyed, as evidenced by the photoluminescence of the redissolved samples". I understand what the author is trying to say, but if decomposition has occurred, it would be better to confirm the chemical species by NMR, since the decomposition components that do not emit light cannot be determined by photoluminescence.

Response: As suggested by **Reviewer 2**, ¹H-NMR spectrum of the **R-ql-Br** cluster after heat treatment at 150 °C was tested, which was basically consistent with the ¹H-NMR spectra before heating treatment except that the peaks of solvent molecules decreased, showing that the structure was not destroyed (Supplementary Fig. 15). We have modified the corresponding description in the revised manuscript (Page 8).

In addition, the PXRD data of **R-ql-Br** after heat treatment at 150 °C were determined through the refinement of the PXRD patterns using the reflex module combined in the Materials Studio program with ultra-fine convergence quality. The pseudo-Voigt function and Berar-Baldinozzi correction were applied to fit the peak profile and asymmetry, respectively. The refined parameters included FWHM (U, V, W), profile parameters (NA, NB), asymmetry (P1–P4), background coefficients, lattice (*a*, *b*, *c*, *V*), and lattice strain (A, B, C). As shown in Supplementary Fig. 16, the data fitting results showed that compared with the as-synthesized, the volume of the **R-ql-Br** after heating at 150 °C was significantly reduced and the *a*, *b*, *c* parameter also had a slight change, indicating that the slight shift of PXRD pattern may be due to the change of cluster stacking in the unit cell caused by the escape of solvent molecules.

Supplementary Figure 15. (a) $^1\text{H-NMR}$ spectrum of the *R-ql-Br* cluster. (b) $^1\text{H-NMR}$ spectrum of the *R-ql-Br* cluster after heat treatment at $150\text{ }^\circ\text{C}$ (600 MHz, CD_2Cl_2).

Supplementary Figure 16. Pawley refinement of the PXRD pattern for *R-ql-Br* cluster after heat treatment at $150\text{ }^\circ\text{C}$.

8. The possible existence of the C-H...M interactions between the hydrogen atoms of the methylene groups in the wingtip and gold (or copper) in the cluster structure should be commented on. The following references may be consulted: H. Schmidbaur, *Angew. Chem. Int. Ed.* 2019, 58, 5806; M. Shionoya et al., *Bull. Chem. Soc. Jpn.* 2021, 94, 1324.

Response: As suggested by **Reviewer 2**, hydrogen atoms of the two methylene groups in all clusters are depicted interacting with gold atoms, with a minimum C-H...Au distance of 2.759 Å, which is shorter than the sum of van der Waals radii of hydrogen and gold (2.860 Å). The related discussion and reference have been included in the revised manuscript (Ref.53-54), and the spectra have been added in **Supplementary Fig. 6**.

Supplementary Figure 6. Structure motifs and the corresponding schemes showing intramolecular C-H...Au interactions in (a) *R*-py-Br, (b) *R*-py-I, (c) *R*-ql-Cl, (d) *R*-ql-Br, and (e) *R*-ql-I.

9. On page 8, the authors first explain **Fig. 2** in the paragraph of "lines 147-160", then go back to **Fig. 1d** in the paragraph of "lines 161-173". For example, it would be better to explain **Fig. 1d** first, then **Fig. 2**, and so on, explaining each figure in turn. In fact, it was only upon reviewing the theoretical calculations section (pp. 13-18) that I finally realized that the paragraph "lines 161-173 on page 8" may be the conclusion the authors reached after conducting a theoretical review. (For example, lines 164-165, "...where Au promoted SOC and increased intersystem crossing ISC by the heavy atom effect". Line 170, "optimized the ISC process through the heavy-atom effect (Br and I)." Page 9, line 172, "...HOMOs and LUMOs to promote the TADF mechanism "?) Thus, the conclusion after the experiment seems strange and abrupt in **Fig. 1d**. Such conclusions based on theoretical calculations would be better properly moved to the theoretical calculation section to avoid confusion.

Response: As suggested by **Reviewer 2**, we have moved the conclusions of **Fig. 1d** to the theoretical calculation section, which was presented as **Fig. 6** in the revised manuscript.

Photoluminescence

10. On page 9, lines 186-189, the sentence "In addition, **R-ql-I** barely changed after irradiation of the sample with 400 nm light for 3 h, indicating the excellent photostability of this material." seems unclear. For instance, is this result in solution or in the solid state? What is the light source? Is it strong light (for example strong UV light (300 W Hg lamp)) or just normal light of the HORIBA FluoroLog-3 fluorescence spectrometer used in this work? Besides the stability of **R-ql-I**, what about photostability studies of other clusters involved in this study?

Response: As suggested by **Reviewer 2**, the photostability of **R-ql-I** in the solid state was tested under xenon lamp (the HORIBA FluoroLog-3 fluorescence spectrometer) irradiation. Besides the stability of **R-ql-I**, the photostability studies of other clusters involved in this study were also investigated and spectra were added to **Supplementary Fig. 22**.

Supplementary Figure 22. Time-dependent fluorescence spectra of (a) **R-py-Br**, (b) **R-py-I**, (c) **R-ql-Cl**, (d) **R-ql-Br**, (e) **R-ql-I** exposed to 400 nm UV light for 3 h in the solid state.

11. In the Supplementary Information (page 27, line 388), "Supplementary Figure 17...(c) ... in the solution state". What solvent was used? Under argon or ambient condition? Please show the solvent in solution state in each related Supplementary Figure.

Response: As suggested by **Reviewer 2**, the used solvent dichloromethane and the ambient conditions have been indicated in the revised supporting information. In addition, we also provided more experimental details.

Temperature-dependent Photoluminescence

12. On page 11, lines 217-218, "... variable-temperature SCXRD on the solventfree **R-NHCql-AuCu4-I** in the range of 100-300 K, which yielded an identical crystalline structure". What is the Cu(I)...Cu(I) distance, because the Cu(I)...Cu(I) distance was considered important for tuning the photoluminescence?

Response: As suggested by **Reviewer 2**, we explained the Cu(I)...Cu(I) distance in the revised manuscript.

As shown in **Supplementary Table 12**, there was a slight shortage of the Cu(I)···Cu(I) distance with decreasing temperature (average bond length from 2.437 (300 K) to 2.419 Å (100 K)). Such slight changes in Cu(I)···Cu(I) bond variation were not enough cause the dramatic temperature-dependent dual emission behavior of solid state **R-ql-I**. The corresponding results have been added to the revised manuscript and **Supplementary Table 12**.

Supplementary Table 12. Cu-Cu bond lengths for **R-ql-I** at different temperatures.

	100 K	150 K	180 K	200 K	250 K	300 K
Cu1-Cu2	2.4236(9)	2.4264(9)	2.4265(9)	2.4305(9)	2.4327(12)	2.4356(14)
Cu3-Cu4	2.4152(9)	2.4214(9)	2.4281(9)	2.4289(10)	2.4325(11)	2.4363(16)

13. On page 11, line 221, "... measurements were performed from 93 to 383 K". This shows the temperature for Fig. 3b, but is not consistent with the "83 K to 303 K in Supplementary Figures 19-20". And line 231, "...was continues from 83 to 273K" is also the same. It would be better to unify the temperature ranges to avoid confusion.

Response: As suggested by **Reviewer 2**, we unified the temperature ranges in the revised manuscript.

14. On pages 11-12, lines 231-234, "The experimental QY was determined to be up to 93%, lower than that at 273 K because of competition between the thermal-induced nonradiative effects and thermal-activated emission enhancement". This sentence is difficult to understand. Does the sentence "The experimental QY (at what temperature?) was determined to be up to 93%, lower than that at 273 K..." mean that the QY at 273 K is the highest QY among the QY values measured at 83-383 K?

Response: As suggested by **Reviewer 2**, we clarified this sentence in the revised manuscript (**Page 11**).

"The room-temperature PLQY was determined to be up to 93.0%. Nevertheless, room-temperature emission intensity was slightly lower than that at 273 K..."

Theoretical calculations

15. On page 13, lines 255-257, "...yet is 0.95 eV lower than that of R-NHC^{py} (Fig 1B)". The author here returned to Fig. 1B for an explanation. It would be clearer to move the theoretical part in Fig 1b to this theoretical section. The sentences of lines 163-172 seem to be the conclusion provided by the authors in the theoretical part, so it is a bit strange when it appeared in the part of synthesis and structure.

Response: As suggested by **Reviewer 2**, we have moved **Fig. 1b** to the theoretical calculation section, which was presented as **Supplementary Fig. 31** in the revised

manuscript. We have moved the conclusions sentences of lines 163-172 to the theoretical calculation section (Pages 18-19), which was presented as Fig. 6 in the revised manuscript.

16. On page 8, lines 164-165, "...where Au promoted SOC and increased intersystem crossing ISC by the heavy atom effect". As claimed by the authors, Au atom is an important heavy atom for SOC (spin-orbit coupling) as shown in Fig. 1d. However, no clear evidence is provided in the theoretical calculations. For example, have the authors studied the heavy atom effect of this Au atom in comparison to its homometallic Cu analog (such as a similar cluster in which the central Au atom in the AuICu₄ cluster is replaced by a Cu atom)? Moreover, the reference (Chem. Commun., 2016, 52, 2932-2935) mentioned that dicopper NHC-picoly l complexes show that coprophilic interactions can ensure strong SOC. Therefore, it seems important to identify the real factors driving SOC.

Response: Just as **Reviewer 2** commented, a homometallic Cu analog was indeed an ideal model for comparison. Nevertheless, we failed to prepare an isostructural Cu₅ cluster with a substituted Cu-center.

As suggested by **Reviewer 2**, we provided a detailed explanation of the contribution SOC by the new calculation results (Pages 15-16), which showed that Au, Cu (two bonded Cu₂ dimer), halide (Cl, Br, I) differently contributed to SOC in different degree in each cluster.

"Chem. Commun., 2016, 52, 2932-2935" have been cited in **ref. 51**.

"For gaining more insight into the heavy-atom effect contributions to SOC, we analyzed the atom (Au, Cu, Cl, Br, I) contributions to holes and electrons in S₁, T₁/T₂ states and calculated contributions of basis functions to hole and electron (Supplementary Tables 26-28). Cu atoms (bonded Cu₂ dimer)⁵¹ and halide (Cl, Br, I) ions mainly contributed to the hole of **R-ql-X** (X = Cl, Br, I) (Fig. 4) and **R-py-I** (Supplementary Fig. 40). For **R-ql-X** (X = Cl, Br, I), the Au atom contribution to holes and electrons was very small (< 1%); while in **R-py-I** the Au atom contribution to holes was larger than 27%. The results demonstrated that the functional module of pyridine/quinoline with different π* orbitals in ligands modulated the electronic structures and further affected the contribution of Au to SOC."

Supplementary Table 26. Au atom contribution to holes and electrons of S₁, T₁, T₂ states of *R-ql-Cl*, *R-ql-Br*, *R-ql-I* and *R-py-I*.

geometries	Au atom	Hole	Electron	Overlap	Diff.
R-ql-Cl	S ₁	0.27 %	0.27 %	0.27 %	0.39 %
	T ₁	0.03 %	0.29 %	0.10 %	0.26 %
	T ₂	0.05 %	0.26 %	0.26 %	0.21%
R-ql-Br	S ₁	0.13 %	0.60 %	0.28 %	0.46 %
	T ₁	0.07 %	0.27 %	0.14 %	0.20 %
	T ₂	0.08 %	0.18 %	0.12 %	0.10 %
R-ql-I	S ₁	0.05 %	0.52 %	0.15 %	0.47 %
	T ₁	0.16 %	0.11 %	0.13 %	-0.05 %
R-py-I	S ₁	0.64 %	28.38 %	4.28 %	27.74 %
	T ₁	0.37 %	27.42 %	3.17 %	27.05 %

Supplementary Table 27. Cu atom contribution to holes and electron of S₁, T₁, T₂ states of *R-ql-Cl*, *R-ql-Br*, *R-ql-I*, and *R-py-I*.

geometries	Cu atom	Hole	Electron	Overlap	Diff.
R-ql-Cl	S ₁	71.02%	2.42%	11.45%	-68.61%
	T ₁	47.03%	3.98%	13.66%	-43.05%
	T ₂	52.85%	2.44%	11.39%	-50.40%
R-ql-Br	S ₁	65.01%	2.19%	10.34%	-62.82%
	T ₁	37.42%	3.06%	10.58%	-34.35%
	T ₂	62.98%	1.82%	10.64%	-61.20%
R-ql-I	S ₁	54.78%	1.90%	8.75%	-52.87%
	T ₁	14.04%	1.65%	4.81%	-12.38%
R-py-I	S ₁	57.11%	18.66%	27.48%	-38.45%
	T ₁	36.19%	20.48%	21.87%	-15.71%

Supplementary Table 28. Halid X (Cl, Br, I) ions contribution to holes and electron of S₁, T₁, T₂ states of *R-ql-Cl*, *R-ql-Br*, *R-ql-I*, and *R-py-I*.

geometries	X (Cl, Br, I)	Hole	Electron	Overlap	Diff.
R-ql-Cl	S ₁	19.89%	0.53%	3.17%	-19.37%
	T ₁	12.40%	0.92%	3.33%	-11.49%
	T ₂	15.37%	0.49%	2.70%	-14.86%
R-ql-Br	S ₁	25.89%	0.61%	3.83%	-25.28%
	T ₁	12.49%	0.84%	3.20%	-11.65%
	T ₂	24.58%	0.38%	2.94%	-24.22%
R-ql-I	S ₁	36.28%	0.73%	4.92%	-35.55%
	T ₁	5.81%	0.59%	1.82%	-5.22%
R-py-I	S ₁	34.82%	17.71%	19.70%	-17.12%
	T ₁	60.10%	18.04%	24.71%	-42.06%

Chirality and circularly polarized luminescence

17. On page 18, lines 368-369. To understand the chiroptical signals contributed by the asymmetrically arranged Cu(I)-halogen moieties in these novel AuCu₄ clusters, the authors may compare the enantiopure pairs of *R/S*-[Au(NHC^{py})₂]PF₆ and *R/S*-[Au(NHC^{ql})₂]PF₆ with the AuCu₄ clusters by CD measurements. What is the possible reason for "transferring the chirality at the two C atoms of NHC ligands to the whole cluster molecule (page 19, lined 378-379)"?

Response: As suggested by **Reviewer 2**, we explained the transfer of the chirality from the chiroptical properties of chiral clusters from chiral electronic transitions and chiral structure of inorganic Au(I)Cu(I)-halogen moieties. These descriptions have been included and highlighted in the revised manuscript (Pages 20-21).

In CH₂Cl₂ solution (1×10^{-5} mol/L) at room temperature, these enantiomeric pairs of clusters all gave rise to a new CD band in the range of 340–400 nm, related to those of the precursor *R/S*-[Au(NHC^{py/ql})₂]PF₆ with CD signal peaks below 350 nm (Supplementary Figs. 42-43). CD responses were basically consistent with the UV–vis absorption spectra, and those above 340 nm generally involved the electronic transitions between orbitals of cluster molecules (Supplementary Figs. 33-34, 42-47 and Tables 13-17). For example, **R-ql-I** cluster had CD band with peaks at 377 nm, which mainly originated from transitions from HOMO-4 (Cu₄-dominated) to LUMO+3 (quinoline-dominated) and LUMO+4 (Cu₄-dominated). For those CD signals below 340 nm, the chiral electronic transitions partially overlapped with ones in the chiral precursor (Supplementary Figs. 43, 45, 47). In addition, based on the single-crystal structure, we analyzed the asymmetrical conformation of inorganic Au(I)Cu(I)-halogen moieties in **R-ql-I** as a representative, which lacked the mirror plane (σ) and inversion (*i*) symmetry elements, presenting *C*₁ symmetry and chirality (Supplementary Fig. 5). The above chiroptical properties and structural analysis both suggested that the chirality of NHC ligands was successfully transferred to the whole cluster molecule.

Supplementary Figure 42. CD (left) and UV (purple line; right) spectra of *R/S*-[Au(NHC^{py})₂]PF₆ and *R/S*-py-*X* (*X* = Br and I) in CH₂Cl₂ (1×10^{-5} mol/L) under ambient conditions.

Supplementary Figure 43. CD (left) and UV (purple line; right) spectra of *R/S*-[Au(NHC^{ql})₂]PF₆ and *R/S*-ql-*X* (*X* = Cl, Br and I) in CH₂Cl₂ (1×10^{-5} mol/L) under ambient conditions.

18. On page 18. The conditions for CD spectra (Supplementary Figures 35-36) and CPL spectra (Figure 6) in solution should be described.

Response: As suggested by **Reviewer 2**, CD and CPL properties of the solutions tested in the manuscript were characterized by dichloromethane (1×10^{-5} mol/L) as a solvent under ambient conditions. We double-checked the whole supplementary thoroughly and provided detailed descriptions of the solvents and ambients used in the revised supporting information.

19. In the corresponding Supplementary information, pages 46-47, lines 548-554, Supplementary Figs. 35-36, the CD signals for the clusters in the solid state and in solution are different, for example, the ql-based cluster (**R/S-NHCql-AuCu4-I**) in the range of 300-200 nm. Why does this make a difference? Does cluster aggregation affect the chiroptical signals?

Response: As suggested by **Reviewer 2**, we provided the possible explanations of the CD difference between the solid state and in solution in the range of 300-200 nm.

Between 200 and 230 nm, the absorption effect of the solvent resulted in # peaks in CD spectra of a cluster solution, which had no meaning. Above 230, when compared with #2 peak, #1 peaks became more obvious, but with nearly consistent positions. While #2 peaks red-shifted by ~ 8 nm from solution state to solid state (Fig. R1). It was likely that the aggregation effect contributed to the slight shift of the chiroptical signals, due to the dipole interactions between the polar cluster molecules in the crystalline state.

Figure R1. CD spectra of **R/S-ql-I** in the solid and CH_2Cl_2 solution (1×10^{-5} mol/L) under ambient conditions.

Circularly polarized organic light-emitting diodes

20. With regard to the result that the AuICu_4 cluster in this study shows a high external quantum efficiency, it is also interesting to evaluate the comparison with a linear Au(I) complex with NHC ligands (external quantum efficiency = 26.3%, ref. 11 in this study).

Response: As suggested by **Reviewer 2**, we provided the comparison with the carbene-metal-amide complex (CMA1, 26.3% at 550nm) in ref. 11 in the revised manuscript (Page 23).

“The EQE of 20.8% at 586 nm of **S-ql-I** cluster was smaller than the reported linear Au(I) complex with NHC ligands (CMA1, 26.3% at 550nm),¹¹ probably because the longer lifetime of **S-ql-I** (~2 μs) than CMA1 (~350 ns) is not a beneficial factor in OLED performance.”

General issues:

21. Regarding cluster names: it is confusing that different cluster names are used in parts of the cluster. For example, *R/S*-AuCu₄I₄(NHC^{ql})₂PF₆ is denoted *R/S*-NHC^{ql}-AuCu₄-I in the text (lines 110-114) and Figure 6, and the same compound is abbreviated "V" in Table 1 and Figures 2, 4 and 5. In addition, it was sometimes referred to as a "ql-I cluster" (line 186), "ql set cluster" (line 221), "ql base set" (line 259), or "ql-I base cluster" (line 278). In one paper, systematically name all compounds and use only one abbreviation for each compound. For example, I-V or 1-5 would be acceptable.

Response: As suggested by **Reviewer 2**, we have unified the abbreviations for each cluster to facilitate understanding, such as *R/S*-py-Br, *R*-py-I, *R/S*-ql-Cl, *R/S*-ql-Br and *R/S*-ql-I. We double checked the whole manuscript and supporting information thoroughly and corrected these abbreviations in the revised manuscript and supporting information.

22. It is better to indicate the solvent used, especially in Table 1 and in the captions of the supplementary figures.

Response: As suggested by **Reviewer 2**, we have provided the used the solvent in the revised manuscript and supporting information.

23. Regarding significant digits: the authors sometimes use "93%" (e.g., line 28) and sometimes "93.0%" (e.g., line 91). All values of quantum yield, efficiency, etc. in the text should be consistent.

Response: As suggested by **Reviewer 2**, we have revised the number as "93.0%" based on the unified significant digits in the revised manuscript to ensure the unity of significant digits.

Reviewer #3

This review is of crystallographic work presented in the manuscript.

A total of 17 single crystal structures are presented in the manuscript along with figures of PXRD data. There are significant problems with the refinement and reporting of many of the structures which need to be addressed before the work can be recommended for publication. Additional points on the presentation of the SCXRD and PXRD work in the manuscript should be addressed.

Response: We are very grateful to **Reviewer 3** for their professional comments and constructive suggestions. We recollected single-crystal X-ray diffraction data of each crystal and refined them to give better results. PXRD work was also re-performed and refined. All crystal data have been updated and the crystallographic tables have been revised correspondingly.

Comments on main manuscript crystallographic content:

1. 125 Did you mean “chiral space groups” or Sohnke space groups?

Response: Following this suggestion, we have corrected the “chiral space groups” to “Sohnke space groups” in the revised manuscript.

2. 135 Excessive quoted precision on discussion of a dihedral angle (88.873°).

Response: As suggested by **Reviewer 3**, two significant digits were retained and corrected in the manuscript. We have corrected this improper significant digit in the revised Manuscript and **Supplementary Fig. 5a**.

Supplementary Figure 5a. Motif of AuCu_4 in *R-ql-I*. Color codes: Au, yellow; Cu, brown.

3. 150 Did the authors mean dissolved or desolvated?

Response: We originally intended to express desolvated, but due to our negligence in analyzing single crystal data before, the result was misjudged. The reanalysis of the data showed that solvent molecules were present in the crystals. So we have corrected this error in the revised manuscript.

4. 152 Inference of stacking change made from 0.02 shift in 2θ . A fitting refinement of the data (such as the Le Bail method) should be performed to confirm that the shift is attributable to a significant change in unit cell geometry. SI Fig 8 (PXRD) make plots bigger – cannot be seen in enough detail. Suggest each plot needs to be full page.

Response: As suggested by **Reviewer 3**, PXRD of all clusters was re-tested and the results showed that *R-ql-Br* had a slight shift at high temperatures due to the presence

of more solvent molecules. The PXRD data of **R-ql-Br** after heat treatment at 150 °C were determined through the refinement of the PXRD patterns using the reflex module combined in the Materials Studio program with ultra-fine convergence quality. The pseudo-Voigt function and Berar-Baldinozzi correction were applied to fit the peak profile and asymmetry, respectively. The refined parameters included FWHM (U, V, W), profile parameters (NA, NB), asymmetry (P1–P4), background coefficients, lattice (a , b , c , V), and lattice strain (A, B, C). As shown in **Supplementary Fig. 16**, the data fitting results showed that compared with the as-synthesized, the volume of the **R-ql-Br** after heating at 150 °C was significantly reduced and the a , b , c parameter also had a slight change, indicating that the slight shift of PXRD pattern may be due to the change of cluster stacking in the unit cell caused by the escape of solvent molecules.

In addition, the 2θ angle range of PXRD patterns of all clusters was set to 5-30° and reformatted for clarity (**Supplementary Fig. 9**). We have updated the PXRD profiles of all clusters in the revised supporting information.

Supplementary Fig. 16. Pawley refinement of the PXRD pattern for **R-ql-Br** cluster after heat treatment at 150 °C.

5. 159 reproduce fig. 2 at a larger size in SI. I agree that it shows PXRD structure generally retained, but many possibly relevant small details in patterns cannot be discerned with image so small.

Response: We thank **Reviewer 3** for the reminder. For clarity, the 2θ angle of the PXRD pattern in Fig. 2e was set to range 5-30° and reproduced in the revised supporting information (**Supplementary Fig. 9**).

Supplementary Figure 9. PXRD patterns of (a) *R/S*-py-Br, (b) *R/S*-py-I, (c) *R/S*-ql-Cl, (d) *R/S*-ql-Br, (e,f) *R/S*-ql-I.

General SCXRD:

6. For all level A and B alerts in the check CIF report a validation response form field should be added to the CIF giving details of the likely origin of the alert and what steps were taken to mitigate any potential problems.

Response: We thank **Reviewer 3** for the concern. All single crystal X-ray diffraction data were recollected using Mo-K α radiation SCXRD to recollect and refined to ensure that there were no A and B alerts. In addition, all CIF documents have been updated and uploaded to the CCDC database.

7. Use the refine_special_details field in each CIF to describe special measures taken

during refinements to overcome the issues noted for all structures (e.g. reason for use of restraints, reason for omitting data, check on suggested twinning, estimates of solvent screened content). Contents of these sections should be reproduced in SI.

Response: We thank **Reviewer 3** for pointing out this issue. The special measures taken to overcome the problems noted for all structures during the refinement process are described in detail and written in the revised Supporting Information.

R/S-py-X (X = Br and I) and **R/S-ql-X** (X = Cl, Br and I) were measured by single-crystal X-ray diffraction (SCXRD) with a Bruker diffractometer at 200 K, using Mo-K α radiation ($\lambda = 0.71073$ Å). **R-ql-I** was also measured at different temperatures (100, 150, 180, 200, 250, and 300 K). Data collection and reduction were performed using the program CrysAlisPro⁴. The intensities were corrected for absorption using the empirical method implemented in SCALE3 ABSPACK scaling algorithm. The structures were solved with intrinsic phasing methods (SHELXT-2015), and refined by full-matrix least-squares on F^2 using OLEX2⁵, which utilizes the SHELXL-2015 module. The least-squares refinement of the structural model was performed under hard geometry restraints and displacement parameter restraints due to the weak diffraction and serious disorder of PF₆⁻, Et₂O and CH₂Cl₂ molecules in the lattice, such as ISOR, SADI, SIMU, and DFIX. Solvent molecules of all clusters have been identified and further confirmed by thermogravimetric and elemental analysis. All host molecular atoms were refined anisotropically, and the hydrogen atoms were included in idealized positions.

For **R-py-Br**, due to the weak diffraction and serious disorder of PF₆⁻ and CH₂Cl₂ molecules in the lattice, we used SADI restraints C-Cl bond of CH₂Cl₂ molecular; used ISOR restraints F and C atoms of PF₆⁻ and CH₂Cl₂ molecular. We examined the twinning by BASF and confirmed the presence of Twin Law (-1, 0, 0, 0, -1, 0, 0, 0, -1), BASF [0.008(7)]. No solvent squeeze was used.

For **S-py-Br**, due to the weak diffraction and disorder of PF₆⁻ and CH₂Cl₂ molecules in the lattice, we used ISOR restraints F and C atoms of PF₆⁻ and CH₂Cl₂ molecular. We examined the twinning by BASF and confirmed the presence of Twin Law (-1, 0, 0, 0, -1, 0, 0, 0, -1), BASF [0.015(8)]. No solvent squeeze was used.

For **R-py-I**, due to the weak diffraction and serious disorder of PF₆⁻ and Et₂O molecules in the lattice, we used SADI and DFIX restraints P-F bond of PF₆⁻ molecular; we used DFIX restraints C-C and C-O bonds of Et₂O molecular; we used ISOR restraints F atoms of PF₆⁻. There are no possible Twin Laws. No solvent squeeze was used.

For **S-py-I**, due to the weak diffraction and serious disorder of Et₂O molecules in the lattice, we used DFIX restraints C-C and C-O bonds of Et₂O molecular; we used SIMU restraints O and C atoms of Et₂O molecular. There are no possible Twin Laws. No solvent squeeze was used.

For **R-ql-Cl**, due to the weak diffraction and serious disorder of Et₂O molecules in the lattice, we used DFIX restraints C-C and C-O bond of Et₂O molecular; we used SIMU and ISOR restraints F and C atoms of PF₆⁻ and Et₂O. We examined the twinning by BASF and confirmed the presence of Twin Law (0.5, 1.5, 0, -0.5, 0.5, 0, 0, 0, 1), BASF [0.0004(2)]. No solvent squeeze was used.

For **S-ql-Cl**, due to the weak diffraction and serious disorder of PF₆⁻ and Et₂O molecules in the lattice, we used DFIX restraints C-C and C-O bond of Et₂O molecular; we used SIMU and ISOR restraints F and C atoms of PF₆⁻ and Et₂O; we used ISOR restraints disorder C atoms of NHC^{ql} ligand. We examined the twinning by BASF and confirmed the presence of Twin Law (0.5, 1.5, 0, -0.5, 0.5, 0, 0, 0, 1), BASF [0.0004(2)]. No solvent squeeze was used.

For **R-ql-Br**, due to the weak diffraction and serious disorder of Et₂O molecules in the lattice, we used DFIX restraints C-C and C-O bonds of Et₂O molecular; used ISOR and SIMU restraints O and C atoms of Et₂O molecular; used ISOR restraints disorder C atoms of NHC^{ql} ligand. We examined the twinning by BASF and confirmed the presence of Twin Law (-1, 0, 0, 0, -1, 0, 0, 0, -1), BASF [0.009(9)]. No solvent squeeze was used.

For **S-ql-Br**, due to the weak diffraction and serious disorder of Et₂O molecules in the lattice, we used DFIX restraints C-C and C-O bonds of Et₂O molecular; used ISOR restraints disorder C atoms of NHC^{ql} ligand; used EDAP and EXYZ restraints C atom of Et₂O and CH₂Cl₂ to share one atomic coordinate. There are no possible Twin Laws. No solvent squeeze was used.

For **R-ql-I-100 K**, due to the weak diffraction and serious disorder of Et₂O molecules in the lattice, we used DFIX restraints C-C and C-O bonds of Et₂O molecular; we used SIMU restraints O and C atoms of Et₂O molecular. We examined the twinning by BASF and confirmed the presence of Twin Law (-1, 0, 0, 0, -1, 0, 0, -1, 2), BASF [0.010(4)]. No solvent squeeze was used.

For **R-ql-I-150 K**, due to the weak diffraction and serious disorder of Et₂O molecules in the lattice, we used DFIX restraints C-C and C-O bonds of Et₂O molecular. There are no possible Twin Laws. No solvent squeeze was used.

For **R-ql-I-180 K**, due to the weak diffraction and serious disorder of Et₂O molecules in the lattice, we used DFIX restraints C-C and C-O bonds of Et₂O molecular; we used SIMU restraints O and C atoms of Et₂O molecular. There are no possible Twin Laws. No solvent squeeze was used.

For **R-ql-I-200 K**, due to the weak diffraction and serious disorder of Et₂O molecules in the lattice, we used DFIX restraints C-C and C-O bonds of Et₂O molecular; we used SIMU restraints O and C atoms of Et₂O molecular; we used ISOR restraints disorder C atoms of NHC^{ql} ligand. There are no possible Twin Laws. No solvent squeeze was used.

For **S-ql-I-200 K**, due to the weak diffraction and serious disorder of Et₂O molecules in the lattice, we used DFIX restraints C-C and C-O bonds of Et₂O molecular; we used SIMU restraints O and C atoms of Et₂O molecular; we used ISOR restraints disorder C atoms of NHC^{ql} ligand. We examined the twinning by BASF and confirmed the presence of Twin Law (-1, 0, 0, 0, -1, 0, 0, 0, -1), BASF [0.004(4)]. No solvent squeeze was used.

For **R-ql-I-250 K**, due to the weak diffraction and serious disorder of Et₂O molecules in the lattice, we used DFIX restraints C-C and C-O bonds of Et₂O molecular; we used SIMU restraints O and C atoms of Et₂O molecular. We used ISOR restraints

disorder C atoms of NHC^{q1} ligand. There are no possible Twin Laws. No solvent squeeze was used.

For **R-ql-I-300 K**, due to the weak diffraction and serious disorder of Et₂O molecules in the lattice, we used DFIX restraints C-C and C-O bonds of Et₂O molecular; used SIMU restraints O and C atoms of Et₂O molecular. We used ISOR restraints disorder C atoms of NHC^{q1} ligand. We examined the twinning by BASF and confirmed the presence of Twin Law (-1, 0, 0, 0, -1, 0, 0, 0, -1), BASF [0.007(6)]. No solvent squeeze was used.

8. Where the reported Flack parameters of structure deviate from zero by >3*esd there should be a comment or explanation about what this means for the assignment of absolute structure.

Response: We thank **Reviewer 3** for the reminder. As suggested, we double-checked the flack parameters of all structures. The flack parameters of all structures are <3*esd, indicating no obvious twinning to eliminate the above problems.

9. Wherever the DELU restraint has been used please explain why it has been selected rather than the updated version of this restraint: RIGU (or re-refine with RIGU instead). Use of RIGU might mitigate the need for widespread unexplained used of ISOR in many structures.

Response: As suggested by **Reviewer 3**. No DELU restraint was used due to all crystal structure retests and refinements.

10. All structures should employ Gaussian absorption correction (Numerical absorption correction based on Gaussian integration over a multifaceted crystal model) (current is multiscan) as this is essential for heavy atom structures and will alleviate bad reflection horror show. If not possible – recollect.

Response: We thank **Reviewer 3** for the valuable comments and suggestions. We initially collected all structures using Cu-K α radiation single crystal X-ray diffraction (SCXRD) with strong diffraction, which resulted in a large μ value due to heavy atomic absorption. As suggested, all structures should employ Gaussian absorption correction (Numerical absorption correction based on Gaussian integration over a multifaceted crystal model). However, Gaussian absorption correction has strict requirements for crystal photography. Unfortunately, we have not collected high-quality crystal photos, so we cannot carry out Gaussian absorption correction. Therefore, to solve this problem, we chose to use Mo-K α radiation SCXRD to recollect all structures, reducing the μ value to near 5 (**Supplementary Tables 1-5, 11**) and updating all CIF documents.

Supplementary Table 1. Crystal data and structure refinement for *R/S-ql-I*

Compound	R-ql-I	S-ql-I
CCDC number	2225247	2225253
Empirical formula	C ₇₄ H ₆₆ AuCu ₄ F ₆ I ₄ N ₈ OP	C ₇₄ H ₆₆ AuCu ₄ F ₆ I ₄ N ₈ OP
Formula weight	2187.04	2187.04
Temperature/K	200	200
Crystal system	orthorhombic	orthorhombic
Space group	P 2 ₁ 2 ₁ 2	P 2 ₁ 2 ₁ 2
a /Å	26.7253(11)	26.701(2)
b /Å	19.5313(7)	19.506(2)
c /Å	14.9188(7)	14.9367(14)
α /°	90	90
β /°	90	90
γ /°	90	90
Volume/Å ³	7787.3(6)	7779.6(13)
Z	4	4
ρ_{calc} g/cm ³	1.865	1.867
μ /mm ⁻¹	4.62	4.624
F(000)	4200	4200
Crystal size/mm ³	0.12 × 0.04 × 0.04	0.15 × 0.05 × 0.05
Radiation	MoK α (λ = 0.71073)	MoK α (λ = 0.71073)
2 θ range for data collection/°	3.758 to 54.986	3.758 to 55.03
Index ranges	-34 ≤ h ≤ 34, -25 ≤ k ≤ 25, -19 ≤ l ≤ 19	-34 ≤ h ≤ 34, -25 ≤ k ≤ 25, -19 ≤ l ≤ 19
Reflections collected	487450	506974
Independent reflections	17859 [R _{int} = 0.0935, R _{sigma} = 0.0255]	17875 [R _{int} = 0.1018, R _{sigma} = 0.0279]
Data/restraints/parameters	17859/25/889	17875/23/889
Goodness-of-fit on F ²	1.036	1.078
Final R indexes [I >= 2 σ (I)]	R ₁ = 0.0250, wR ₂ = 0.0596	R ₁ = 0.0265, wR ₂ = 0.0699
Final R indexes [all data]	R ₁ = 0.0317, wR ₂ = 0.064	R ₁ = 0.0317, wR ₂ = 0.0751
Largest diff. peak/hole / e Å ⁻³	0.75/-1.06	1.67/-1.87
Flack parameters	0.0017(18)	0.004(4)

$$R_1 = \frac{\sum ||F_o| - |F_c||}{\sum |F_o|}, wR_2 = \left[\frac{\sum [w(F_o^2 - F_c^2)^2]}{\sum w(F_o^2)^2} \right]^{1/2}.$$

Supplementary Table 2. Crystal data and structure refinement for *R/S*-py-Br

Compound	R -py-Br	S -py-Br
CCDC number	2225239	2225240
Empirical formula	C _{58.5} H ₅₉ AuBr ₄ ClCu ₄ F ₆ N ₈ OP	C _{58.5} H ₅₉ AuBr ₄ ClCu ₄ F ₆ N ₈ OP
Formula weight	1841.32	1841.32
Temperature/K	200	200
Crystal system	orthorhombic	orthorhombic
Space group	P 2 ₁ 2 ₁ 2 ₁	P 2 ₁ 2 ₁ 2 ₁
a /Å	15.2743(7)	15.2964(9)
b /Å	15.7324(8)	15.7426(10)
c /Å	27.9349(14)	27.9235(18)
α /°	90	90
β /°	90	90
γ /°	90	90
Volume/Å ³	6712.8(6)	6724.1(7)
Z	4	4
ρ_{calc} g/cm ³	1.822	1.819
μ /mm ⁻¹	5.93	5.92
F(000)	3580	3580
Crystal size/mm ³	0.12 × 0.12 × 0.03	0.12 × 0.11 × 0.03
Radiation	MoK α (λ = 0.71073)	MoK α (λ = 0.71073)
2 θ range for data collection/°	3.9 to 55.03	3.712 to 55.098
Index ranges	-19 ≤ h ≤ 19, -20 ≤ k ≤ 20, -36 ≤ l ≤ 36	-19 ≤ h ≤ 19, -20 ≤ k ≤ 20, -36 ≤ l ≤ 36
Reflections collected	381690	375047
Independent reflections	15439 [R _{int} = 0.0738, R _{sigma} = 0.0250]	15484 [R _{int} = 0.1076, R _{sigma} = 0.0333]
Data/restraints/parameters	15439/55/828	15484/72/835
Goodness-of-fit on F ²	1.163	1.161
Final R indexes [I ≥ 2 σ (I)]	R ₁ = 0.0309, wR ₂ = 0.0810	R ₁ = 0.0386, wR ₂ = 0.0911
Final R indexes [all data]	R ₁ = 0.0357, wR ₂ = 0.0865	R ₁ = 0.0509, wR ₂ = 0.1006
Largest diff. peak/hole / e Å ⁻³	1.22/-1.11	1.43/-2.05
Flack parameters	0.008(7)	0.015(8)

$$R_1 = \frac{\sum ||F_o| - |F_c||}{\sum |F_o|}, wR_2 = \left[\frac{\sum [w(F_o^2 - F_c^2)^2]}{\sum w(F_o^2)^2} \right]^{1/2}.$$

Supplementary Table 3. Crystal data and structure refinement for *R/S-py-I*

Compound	R-py-I	S-py-I
CCDC number	2225252	2225251
Empirical formula	C ₅₈ H ₅₈ AuCu ₄ F ₆ I ₄ N ₈ P	C ₅₈ H ₅₈ AuCu ₄ F ₆ I ₄ N ₈ P
Formula weight	1986.82	1986.82
Temperature/K	200	200
Crystal system	tetragonal	tetragonal
Space group	I 4 ₁	I 4 ₁
a /Å	17.6372(5)	17.6980(10)
b /Å	17.6372(5)	17.6980(10)
c /Å	22.5259(7)	22.5113(15)
α /°	90	90
β /°	90	90
γ /°	90	90
Volume/Å ³	7007.2(5)	7051.0(9)
Z	4	4
ρ_{calc} g/cm ³	1.883	1.872
μ /mm ⁻¹	5.123	5.092
F(000)	3784	3784
Crystal size/mm ³	0.14 × 0.12 × 0.1	0.12 × 0.1 × 0.1
Radiation	MoK α (λ = 0.71073)	MoK α (λ = 0.71073)
2 θ range for data collection/°	4.618 to 55.006	4.604 to 54.968
Index ranges	-22 ≤ h ≤ 22, -22 ≤ k ≤ 22, -29 ≤ l ≤ 29	-22 ≤ h ≤ 22, -22 ≤ k ≤ 22, -29 ≤ l ≤ 29
Reflections collected	199395	209824
Independent reflections	8039 [R _{int} = 0.0518, R _{sigma} = 0.0184]	8072 [R _{int} = 0.0504, R _{sigma} = 0.0169]
Data/restraints/parameters	8039/24/394	8072/9/374
Goodness-of-fit on F ²	1.138	1.150
Final R indexes [I ≥ 2 σ (I)]	R ₁ = 0.0228, wR ₂ = 0.0672	R ₁ = 0.0174, wR ₂ = 0.0484
Final R indexes [all data]	R ₁ = 0.0274, wR ₂ = 0.0740	R ₁ = 0.0211, wR ₂ = 0.0557
Largest diff. peak/hole / e Å ⁻³	0.81/-1.26	0.83/-0.85
Flack parameters	0.0012(18)	0.0039(16)

$$R_1 = \frac{\sum ||F_o| - |F_c||}{\sum |F_o|}, wR_2 = \left[\frac{\sum [w(F_o^2 - F_c^2)^2]}{\sum w(F_o^2)^2} \right]^{1/2}.$$

Supplementary Table 4. Crystal data and structure refinement for *R/S-ql-Cl*

Compound	R-ql-Cl	S-ql-Cl
CCDC number	2225238	2225237
Empirical formula	C ₇₂ H ₆₁ AuCl ₆ Cu ₄ F ₆ N ₈ O _{0.5} P	C ₇₀ H ₅₆ AuCl ₆ Cu ₄ F ₆ N ₈ P
Formula weight	1784.18	1832.05
Temperature/K	200	200.00(10)
Crystal system	orthorhombic	orthorhombic
Space group	C 222 ₁	C 222 ₁
a /Å	15.2368(11)	15.2263(15)
b /Å	25.7783(11)	25.7708(15)
c /Å	18.9604(11)	18.9483(15)
α /°	90	90
β /°	90	90
γ /°	90	90
Volume/Å ³	7447.2(8)	7435.2(10)
Z	4	4
ρ_{calc} g/cm ³	1.591	1.594
μ /mm ⁻¹	3.313	3.319
F(000)	3540	3540
Crystal size/mm ³	0.12 × 0.03 × 0.03	0.13 × 0.05 × 0.03
Radiation	MoK α (λ = 0.71073)	MoK α (λ = 0.71073)
2 θ range for data collection/°	3.776 to 55.108	3.778 to 55.832
Index ranges	-19 ≤ h ≤ 19, -33 ≤ k ≤ 33, -33 ≤ l ≤ 24	-19 ≤ h ≤ 19, -33 ≤ k ≤ 33, -24 ≤ l ≤ 24
Reflections collected	248898	211371
Independent reflections	8593 [R _{int} = 0.0999, R _{sigma} = 0.0286]	8748 [R _{int} = 0.1158, R _{sigma} = 0.0384]
Data/restraints/parameters	8593/50/455	8748/40/442
Goodness-of-fit on F ²	1.034	1.030
Final R indexes [I ≥ 2 σ (I)]	R ₁ = 0.0285, wR ₂ = 0.0718	R ₁ = 0.0307, wR ₂ = 0.0735
Final R indexes [all data]	R ₁ = 0.0359, wR ₂ = 0.0755	R ₁ = 0.0461, wR ₂ = 0.0792
Largest diff. peak/hole / e Å ⁻³	0.66/-0.68	0.64/-0.77
Flack parameters	0.009(3)	0.008(3)

$$R_1 = \frac{\sum ||F_o| - |F_c||}{\sum |F_o|}, wR_2 = \left[\frac{\sum [w(F_o^2 - F_c^2)^2]}{\sum w(F_o^2)^2} \right]^{1/2}.$$

Supplementary Table 5. Crystal data and structure refinement for *R/S-ql-Br*

Compound	R-ql-Br	S-ql-Br
CCDC number	2225250	2225243
Empirical formula	$C_{77}H_{73}AuBr_4Cl_2Cu_4F_6$ $N_8O_{1.5}P$	$C_{77}H_{73}AuBr_4Cl_2Cu_4F_6N_8$ $O_{1.5}P$
Formula weight	2121.07	2121.07
Temperature/K	200	200
Crystal system	orthorhombic	orthorhombic
Space group	$P2_12_12_1$	$P2_12_12_1$
a /Å	18.0354(11)	18.1049(10)
b /Å	19.6631(10)	19.6370(10)
c /Å	25.8766(15)	25.8205(14)
α /°	90	90
β /°	90	90
γ /°	90	90
Volume/Å ³	9176.7(9)	9179.9(9)
Z	4	4
ρ_{calc} g/cm ³	1.535	1.535
μ /mm ⁻¹	4.378	4.376
F(000)	4164	4164
Crystal size/mm ³	0.12 × 0.04 × 0.04	0.12 × 0.04 × 0.03
Radiation	MoK α (λ = 0.71073)	MoK α (λ = 0.71073)
2 θ range for data collection/°	3.768 to 55.124	3.776 to 55.004
Index ranges	-23 ≤ h ≤ 23, -25 ≤ k ≤ 25, -33 ≤ l ≤ 33	-23 ≤ h ≤ 23, -25 ≤ k ≤ 25, -33 ≤ l ≤ 33
Reflections collected	606321	642823
Independent reflections	21127 [R_{int} = 0.1304, R_{sigma} = 0.0361]	21052 [R_{int} = 0.0999, R_{sigma} = 0.0300]
Data/restraints/parameters	21127/116/1030	21052/61/962
Goodness-of-fit on F ²	1.107	1.099
Final R indexes [$I \geq 2\sigma(I)$]	$R_1 = 0.0480$, $wR_2 = 0.1305$	$R_1 = 0.0433$, $wR_2 = 0.1196$
Final R indexes [all data]	$R_1 = 0.0587$, $wR_2 = 0.1381$	$R_1 = 0.0520$, $wR_2 = 0.1266$
Largest diff. peak/hole / e Å ⁻³	1.43/-1.37	1.74/-1.11
Flack parameters	0.009(9)	0.004(2)

$$R_1 = \sum ||F_o| - |F_c|| / \sum |F_o|, wR_2 = [\sum [w(F_o^2 - F_c^2)^2] / \sum w(F_o^2)^2]^{1/2}.$$

Supplementary Table 11. Crystal data and structure refinement for **R-ql-I** at different temperatures

	100 K	150 K
CCDC number	2225244	2225246
Empirical formula	C ₇₄ H ₆₆ AuCu ₄ F ₆ I ₄ N ₈ OP	C ₇₄ H ₆₆ AuCu ₄ F ₆ I ₄ N ₈ OP
Formula weight	2187.04	2187.04
Temperature/K	100	150
Crystal system	orthorhombic	orthorhombic
Space group	P 2 ₁ 2 ₁ 2	P 2 ₁ 2 ₁ 2
a /Å	19.4472(9)	26.6715(11)
b /Å	26.6351(14)	19.4832(8)
c /Å	14.7879(8)	14.8453(7)
α /°	90	90
β /°	90	90
γ /°	90	90
Volume/Å ³	7659.8(7)	7714.3(6)
Z	4	4
ρ_{calc} g/cm ³	1.896	1.883
μ /mm ⁻¹	4.697	4.664
F(000)	4200	4200
Crystal size/mm ³	0.12 × 0.04 × 0.04	0.12 × 0.04 × 0.04
Radiation	MoK α (λ = 0.71073)	MoK α (λ = 0.71073)
2 θ range for data collection/°	3.782 to 55.024	3.772 to 54.966
Index ranges	-25 ≤ h ≤ 25, -34 ≤ k ≤ 34, -19 ≤ l ≤ 19	-34 ≤ h ≤ 34, -25 ≤ k ≤ 25, -19 ≤ l ≤ 19
Reflections collected	356146	405554
Independent reflections	17579 [R _{int} = 0.0938, R _{sigma} = 0.0306]	17700 [R _{int} = 0.0901, R _{sigma} = 0.0270]
Data/restraints/parameters	17579/13/889	17700/9/888
Goodness-of-fit on F ²	1.149	1.080
Final R indexes [I ≥ 2 σ (I)]	R ₁ = 0.0229, wR ₂ = 0.0558	R ₁ = 0.0240, wR ₂ = 0.0616
Final R indexes [all data]	R ₁ = 0.0266, wR ₂ = 0.0628	R ₁ = 0.0285, wR ₂ = 0.0636
Largest diff. peak/hole / e Å ⁻³	1.80/-1.74	1.46/-1.62
Flack parameters	0.010(4)	0.0013(17)

Supplementary Table 11 (continued)

	180 K	250 K
CCDC number	2225245	2225248
Empirical formula	C ₇₄ H ₆₆ AuCu ₄ F ₆ I ₄ N ₈ OP	C ₇₄ H ₆₆ AuCu ₄ F ₆ I ₄ N ₈ OP
Formula weight	2187.04	2187.04
Temperature/K	180	250.0
Crystal system	orthorhombic	orthorhombic
Space group	P 2 ₁ 2 ₁ 2	P 2 ₁ 2 ₁ 2
a /Å	26.6989(12)	26.7394(12)
b /Å	19.5129(8)	19.5607(8)
c /Å	14.8876(7)	14.9925(7)
α /°	90	90
β /°	90	90
γ /°	90	90
Volume/Å ³	7756.0(6)	7841.7(6)
Z	4	4
ρ_{calc} g/cm ³	1.873	1.852
μ /mm ⁻¹	4.638	4.588
F(000)	4200	4200
Crystal size/mm ³	0.12 × 0.04 × 0.04	0.12 × 0.04 × 0.04
Radiation	MoK α (λ = 0.71073)	MoK α (λ = 0.71073)
2 θ range for data collection/°	3.764 to 55.208	3.746 to 55.014
Index ranges	-34 ≤ h ≤ 34, -25 ≤ k ≤ 25, -19 ≤ l ≤ 19	-34 ≤ h ≤ 34, -25 ≤ k ≤ 25, -19 ≤ l ≤ 19
Reflections collected	580981	417573
Independent reflections	17885 [R _{int} = 0.0900, R _{sigma} = 0.0238]	18007 [R _{int} = 0.0970, R _{sigma} = 0.0299]
Data/restraints/parameters	17885/19/890	18007/32/887
Goodness-of-fit on F ²	1.104	1.095
Final R indexes [I ≥ 2 σ (I)]	R ₁ = 0.0231, wR ₂ = 0.0592	R ₁ = 0.0287, wR ₂ = 0.0630
Final R indexes [all data]	R ₁ = 0.0298, wR ₂ = 0.0643	R ₁ = 0.0418, wR ₂ = 0.0728
Largest diff. peak/hole / e Å ⁻³	1.21/-1.59	1.00/-1.64
Flack parameters	0.0031(15)	-0.002(2)

Supplementary Table 11 (continued)

	300 K
CCDC number	2225249
Empirical formula	C ₇₀ H ₅₆ AuI ₄ Cu ₄ F ₆ N ₈ P
Formula weight	2112.92
Temperature/K	300.0
Crystal system	orthorhombic
Space group	P 2 ₁ 2 ₁ 2
a /Å	19.5906(6)
b /Å	26.7630(8)
c /Å	15.0553(5)
α /°	90
β /°	90
γ /°	90
Volume/Å ³	7893.5(4)
Z	4
ρ_{calc} g/cm ³	1.778
μ /mm ⁻¹	4.553
F(000)	4032
Crystal size/mm ³	0.12 × 0.04 × 0.04
Radiation	MoK α (λ = 0.71073)
2 θ range for data collection/°	3.736 to 55.042
Index ranges	-25 ≤ h ≤ 25, -34 ≤ k ≤ 34, -19 ≤ l ≤ 19
Reflections collected	466984
Independent reflections	18167 [R _{int} = 0.1224, R _{sigma} = 0.0337]
Data/restraints/parameters	18167/71/889
Goodness-of-fit on F ²	1.071
Final R indexes [I ≥ 2 σ (I)]	R ₁ = 0.0375, wR ₂ = 0.0906
Final R indexes [all data]	R ₁ = 0.0606, wR ₂ = 0.1042
Largest diff. peak/hole / e Å ⁻³	1.13/-1.78
Flack parameters	0.007(6)

$$R_1 = \frac{\sum ||F_o| - |F_c||}{\sum |F_o|}, \quad wR_2 = \left[\frac{\sum [w(F_o^2 - F_c^2)^2]}{\sum w(F_o^2)^2} \right]^{1/2}.$$

11. Where a solvent mask has been applied to the data (*R*_py_I, *S*_py_I, *R*_ql_Br) and estimation of the omitted solvent content should be made and added to the unit cell contents and moiety entries. Solvent contents are used to calculate several reported parameters in the CIF and should reflect the best estimate of the complete contents of the crystal.

*R*_py_Br

Explain the strategy for use of DFIX on PF6

Why DELU, not RIGU?

Explain why ISOR employed for DCM and PF6 (obsv prob disordered)
Solvent mask used but no estimation of contents made or included in unit cell contents
(suggest diethyl ether and a DCM)

S_py_Br

Excessive (55) omits

R_py_I

Solvent mask used but no estimation of contents made or included in unit cell contents
(suggest around two DCM)

Excessive omit list of 282 reflections omitted. This suggests an underlying problem with the data, such as untreated twinning, which should be addressed or explained. The assignment of absolute structure, as indicated by the Flack parameter, cannot be taken at face value with such a large portion of data omitted. What is the outcome of refinement and absolute structure determination without omits?

S_py_I

The checkCIF report indicates that the Flack parameter is inconclusive and that a BASF/TWIN refinement should be attempted. This refinement should be attempted and the outcome and conclusions recorded in a validation response form (VRF) in the cif.

R_ql_Br

Used DELU not RIGU

Wide spread use of ISOR – why?

269 Reflections omitted – see R_py_I comments.

BASF/TWIN refinement suggested – see comment for S_py_I

S_ql_Br

Used DELU not RIGU

Wide spread use of ISOR – why?

Suggested BASF/TWIN refinement.

R_ql_Cl

Use of DELU instead of RIGU

Include solvent mask contents

S_ql_Cl

Use of DELU instead of RIGU

R_ql_I_CH2Cl2

Use of DELU instead of RIGU

Include solvent mask contents

S_ql_I_CH2Cl2

Use of DELU instead of RIGU

Include solvent mask contents

R_ql_i_100K

Include solvent mask contents

R_ql_i_150K

Suggested BASF/TWIN refinement.

Include solvent mask contents

R_ql_i_180K

Include solvent mask contents

R_q1_i_200K

Include solvent mask contents

R_q1_i_250K

Include solvent mask contents

R_q1_i_300K

Include solvent mask contents

Use of DELU instead of RIGU

R_q1_l_300K

Include solvent mask contents

Response: We thank **Reviewer 3** for the professional suggestion. All single crystal X-ray diffraction data were recollected and refined to ensure that there were no A and B alerts. The special measures taken to overcome the problems noted for all structures during the refinement process were described in detail (see details above comments 7) and included in the revised supporting information.

Reviewer #4

The work performed by Zang and coworkers demonstrated the synthesis of asymmetric Au-Cu₄ clusters with multidentate chiral NHC ligands exhibiting both the chemical stability and bright photoluminescence with CPL activity. The chiral heterometallic clusters were further applied to a CP-OLED device with a high external quantum efficiency. Some of the obtained clusters are considered to possess TADF property contributing to the extremely high PLQY over 93% in the solid state. *The significance of this study may include such the remarkable records (PLOY and EQE), TADF, thermal stability, and CPL and CP-OLED activity. With all those performances comprehensively considered, I think the present results may attract the readership of Nat. Commun.* The following questions maybe needed to be addressed in the revision cycles.

Response: We appreciate **Reviewer 4**'s highly positive comments and recognition of the significance of the work presented in our manuscript.

1. In the introduction section, the ligand design was a little introduced. How is the present ligand design rational from the viewpoint of OLED application?

As suggested by **Reviewer 4**, we provided ligand design from the viewpoint of an OLED application.

“Considering TADF emitters through harvesting singlet excitons showed superiority in OLED applications, for high-efficiency TADF, metal clusters having the appropriate lowest unoccupied molecular orbital (LUMO) to avoid cluster-center emission is necessary.^{29, 33} Concieving above strategy, we embedded pyridine/quinoline ring with adjustable π^* orbitals in ligand to achieve spatial separation between donor and acceptor moieties.”

2. The chiral centers of ligand are a little bit apart from the actual coordination sites. Could the authors discuss the introduction of distortion in the AuCu₄(X₄) core motifs with the result of SCXRD?

Response: As suggested by **Reviewer 4**, we discussed the AuCu₄(X₄) core based on SCXRD structure (Page 7).

AuCu₄I₄ structure was discussed with **R-ql-I** as a representative. The motif of AuCu₄ can be represented as two triangles of Cu-Au-Cu sharing a gold atom, which is nearly orthometric with a dihedral angle of 88.27°. In addition, The two Cu-Au-Cu triangular planes and the four I-Cu-Cu triangular planes have different bond lengths and bond angles (Supplementary Fig. 5), resulting in the absence of a mirror plane and a C₁ point group, and suggesting the structural distortion of AuCu₄I₄ skeleton which was induced by chiral ligands. We have added a detailed discussion about this issue in our revised manuscript.

Supplementary Figure 5. (a) Motif of AuCu₄ in *R-ql-I*. (b) Distorted structure of the AuCu₄I₄ skeleton in *R-ql-I*, including bond length. (c) The bond angle of the AuCu₄I₄ skeleton in *R-ql-I*. Color codes: Au, yellow; I, purple; Cu, brown.

3. Page 8, from line 161: Is it possible to discuss the perspective of electronic structure of whole cluster with individual consideration of each given part in this stage? After all, the electronic structure of clusters is discussed based on the DFT calculation later.

Response: As suggested by **Reviewer 4**, we have added the discussion of the electronic structures of the whole cluster with individual consideration in the last paragraph of the theoretical calculations section (Pages 18-19).

4. Page 14, line 287: “from 5.4% and 62.8% to 93.0%.” Maybe “of PLQY” is missing. This sentence says that the more efficient ISC leads to the higher PLQY value. Is this right? If my understanding is correct, the final emitting state is the S₁ for TADF. For the cluster-III, I guess the S₁ to T₂ ISC followed by the rapid IC to T₁ occurred but the RISC to S₁ was rather prohibited, resulting in the large contribution of phosphorescence with low efficiency. The similar explanation appears later (in the end of Page 16).

Response: As suggested by **Reviewer 4**, we have corrected “from 5.4% and 62.8% to 93.0%.” to “...the faster k_{ISC} and k_{RISC} supported the higher PLQY in the order of 5.4% (*R-ql-Cl*) < 62.8% (*R-ql-Br*) < 93.0% (*R-ql-I*)” in this sentence. And revised the explanations in the revised manuscript (Pages 17-18).

“Therefore, *R-ql-Cl*, we gave a reasonable explanation: the S₁ to T₂ ISC followed by the rapid IC to T₁ occurred but the RISC to S₁ was rather prohibited, resulting in the large contribution of phosphorescence with low efficiency (5.4%).”

5. Could the effect of Au atom on the SOC be discussed quantitatively with theoretical calculations?

Response: Thank you for your advice. As suggested by **Reviewer 4**, we calculated the contributions of each moiety involving the transitions to S₁ and T₁/T₂. The results are the following:

The effect of the Au atom on the SOC can be indirectly calculated by statistics of the Au atom's contribution to holes and electrons. As shown in **Supplementary Table 26**, we have calculated the Au atom contribution to holes and electrons of S₁, T₁, T₂ states of **R-ql-X** (X=Cl, Br, I) and **R-py-I**. In **R-ql-X** (X=Cl, Br, I), the Au atom contribution to holes and electrons was very small (< 1%), while in **R-py-I** the Au atom contribution to holes was larger than 27% (**Pages 15-16**).

Combining with the molecular orbital (**Supplementary Fig. 32**), atom orbital (**Supplementary Table 26**) and hole and electron pairs (**Fig. 4 and Supplementary Fig. 40**) analysis, we can find that the lowest unoccupied molecular orbital (LUMO) mainly contributed by p atomic orbitals of the Au, which led to the excited electron of S₁/T₁ in **R-py-I** came from the d atomic orbitals of Cu and p atomic orbitals of I to p atomic orbitals of the Au (d_(Cu)→p_{Au}, p_(I)→p_{Au}). While in **R-ql-X** (X = Cl, Br, I), the Au orbital contribution stayed in high-lying energies molecular orbital of LUMO+4 (p atomic orbitals of the Au), LUMO, LUMO+1, LUMO+2 is three energetically degenerate molecular orbitals, the electron cloud was mainly concentrated on the NHC^{ql} ligands, the Au orbital contribution almost 0%. The excited electron of S₁/T₁/T₂ in **R-ql-X** (X = Cl, Br, I) came from the d atomic orbitals of Cu and p atomic orbitals of X to p atomic orbitals of the NHC^{ql} ligands (d_(Cu)→p_{NHC^{ql}}, p_(Cl, Br, I)→p_{NHC^{ql}}). So NHC^{ql} and NHC^{py} ligands modulating the electronic structure further affected the contribution of Au to SOC.

Supplementary Table 26. the Au atom contribution to holes and electron of S₁, T₁, states of **R-ql-X** (X=Cl, Br, I) and **R-py-I**.

geometries	Au atom	Hole	Electron	Overlap	Diff.
R-ql-Cl	S ₁	0.27 %	0.27 %	0.27 %	0.39 %
	T ₁	0.03 %	0.29 %	0.10 %	0.26 %
	T ₂	0.05 %	0.26 %	0.26 %	0.21%
R-ql-Br	S ₁	0.13 %	0.60 %	0.28 %	0.46 %
	T ₁	0.07 %	0.27 %	0.14 %	0.20 %
	T ₂	0.08 %	0.18 %	0.12 %	0.10 %
R-ql-I	S ₁	0.05 %	0.52 %	0.15 %	0.47 %
	T ₁	0.16 %	0.11 %	0.13 %	-0.05 %
R-py-I	S ₁	0.64 %	28.38 %	4.28 %	27.74 %
	T ₁	0.37 %	27.42 %	3.17 %	27.05 %

6. Fig. 5d-f: More detailed explanation should be provided for the result of transient absorption measurement. How is the shape of TA spectrum of III different from those of IV and V.

Response: As suggested by **Reviewer 4**, we detailed explanation of transient absorption.

We performed transient absorption (TA) spectroscopy to further examine the differences in the ultrafast electron dynamics **R-ql-X** (X = Cl, Br and I) clusters (**Figs.**

5d-f). Under pumped laser excitation at 360 nm, **R-ql-X** (X = Cl, Br and I) clusters displayed net ground-state bleaching (GSB) at about 380 to 385 nm and two photoinduced absorptions (PA) signals (centered at ~450 nm and ~570 nm, respectively). Note that spectra below 380 nm were not detected because of the photo leak of the pump laser, but we could still speculate that it was also GSB signal according to the results of UV-visible absorption of **R-ql-X** (X = Cl, Br and I) clusters. The broad PA signal peaks were attributed to the excited state absorption. However, the two absorbed signal intensities of **R-ql-Cl** were slightly different from those of **R-ql-Br** and **R-ql-I**, which may be due to the larger absorption cross sections of **R-ql-Cl** at ~570 nm, while **R-ql-X** (X = Br and I) had larger absorption cross sections at ~450 nm. Following the initial signal decay, the PA signal featured a signal build-up process, suggesting that an excited triplet state appeared. So we could safely attribute the signal build-up process to the ISC process from the singlet to the triplet. According to the global fit, we got the rising ISC rate of **R-ql-Cl** (75 ps), **R-ql-Br** (34 ps), to **R-ql-I** (12 ps) (Figs. 5d-f), which were consistent with the trend of the DFT calculations (Supplementary Table 29).

7. Supplementary Figure 39b: Spectra in the top of (b) should be those for *R/S*-NHC^{ql}-AuCu₄-“Cl”.

Response: As suggested by **Reviewer 4**, we have corrected this mistake in **Supplementary Fig. 50** in the revised Supporting Information.

Supplementary Figure 50. g_{lum} values **R/S-py-X** (X = Br and I) (a) of **R/S-ql-X** (X = Cl, Br and I) (b) in CH₂Cl₂ under ambient conditions.

8. A comment on the comparison of g_{EL} values with g_{PL} in the solid state could be given.

Response: As suggested by **Reviewer 4**, we provide the comparison of g_{EL} values with g_{PL} in the solid state in the revised manuscript (Page 23).

“Reasonable dissymmetry factors (g_{EL}) of 5.0×10^{-4} for the *R*-OLED and -7.7×10^{-4} for the *S*-OLED (Fig. 8d and Supplementary Fig. 52) were basically consistent with the photoluminescence asymmetry factor ($|g_{PL}| = 9.0 \times 10^{-4}$), which showed that the chirality of cluster in OLED devices did not change a lot.”

Reviewers' Comments:

Reviewer #1:

Remarks to the Author:

The authors considered my previous comments and revised the manuscript suitably. I appreciate the further calculations which strengthen the contents and discussion. Also, the theoretical part of the manuscript is now much improved to understand the contents. I have noticed the following minor points that the authors may consider. I would like to leave these issues to the authors and the further review is not necessary for my review. I recommend the publication of this paper.

(1) I think it is better to slightly change the discussion from p. 16 to 17 with less emphasizing the case of R-ql-Cl, because the kISC from S1 to T1 for R-ql-Cl is nearly zero and the expression of "faster kISC and kRISC" is not valid for R-ql-Cl cluster and the explanation of the origin for PL of R-ql-Cl is given later and in p. 18. So, I suggest to focus on the large PLQE for R-ql-Br and R-ql-I cases here. For example,

The calculated ... (kISC) from S1 to T1 of R-ql-Br (...) and R-ql-I (...) clusters is much larger than that of R-ql-Cl ($\sim 0 \text{ s}^{-1}$). (p. 16)

Resultantly, the faster kISC and kRISC lead to the high PLQY of R-ql-Br (62.8%) and R-ql-I (93.0%) compared to that of R-ql-Cl (5.4%). (p. 17)

(2) It is recommended to revise some sentences and notations as follows:

heavy-atom effect from Cl to Br to I, => heavy-atom effect from Cl to I, (p. 14)

$\langle S_1 | HSOC | T_2 \rangle \Rightarrow \langle S_1 | H_SOC | T_2 \rangle$ (p. 15)

SOCME (p. 15) should be defined when it is firstly introduced.

inter-system => intersystem (p. 17)

Resultantly. => Resultantly, (p. 17)

Reviewer #2:

Remarks to the Author:

[See attached file]

I appreciate the authors' efforts and time in responding to my comments. Many of them have been answered appropriately. I support the acceptance of this manuscript by Nature Communications, but would like to point out typographical errors and minor details in the revised manuscript.

For the main text part:

On page 4, line 70, the expression of "... enantiomer chiral NHC ligands" should be "...using the enantiopure chiral NHC ligands".

On page 5, lines 79-80, "Conceiving above strategy, we embedded pyridine/quinoline ring with adjustable π^* orbitals in ligand to achieve spatial separation between donor and acceptor moieties." In the case of the cited reference 12 (Di et al, Science 356, 159-163 (2017).), the donor and acceptor are indeed separated for about 3.7 Å. However, it is difficult to understand in this study.

On page 5, lines 88-93, "Two pairs of enantiomers of tridentate NHC ligands were designed and synthesized (Fig. 1a)..." Fig.1a shows the precursors of these NHC ligands, not the direct ligands yet. Please also confirm the caption in Figure 1a. Also, the names using "(4*R*/4*S*, 5*R*/5*S*)-..." are not appropriate, since it may be confused with the case of (4*R*, 5*S*)-. It may be better to use (4*R*, 5*R*)-/ (4*S*, 5*S*)- instead. Similarly, on page 6, line 103, "(1*R*/1*S*, 2*R*/2*S*)-..." may be confused with the case of (1*R*, 2*S*)-.

On page 6, line 125, "Sohnke space group" should be "Sohncke space group".

On page 13, line 264, "(X = Br, and I)" should be "(X = Br and I)".

On page 14, lines 266-267, (X = Cl, Br and I) should be (X = Cl, Br, and I); a similar problem in lines 273-274, and etc. (please carefully check the main text and SI).

For the Supporting Information:

On page 15, line 361, **Supplementary Figure 1.** "(a) ^1H NMR spectrum of *R*-NHC^{ql}-H·PF₆." should be "(a) ^1H NMR spectrum of *R*-NHC^{py}-H·PF₆ (CDCl₃ solvent?)." In the same caption, the "*R*-NHC^{py}-H PF₆" should be "*R*-NHC^{py}-H·PF₆", the "*R*-NHC^{ql}-H PF₆" should be "*R*-NHC^{ql}-H·PF₆".

Reviewer #3:

Remarks to the Author:

All suggested changes to the manuscript and the requested SCXRD refinement details are fine. Since the original submission the authors have recollected the SCXRD data using Mo radiation and repeated the refinements. The new data is a great improvement and has eliminated the major crystallographic refinement problems which affected the original data sets.

There are some minor problems with the new SCXRD refinements which need to be fixed before publication can be recommended. Below is a list of requests for each structure and some general details on the problems which are found across many of the structures.

Twin Refinements:

The authors have correctly checked for possible twinning in many of the structures by using a Twin law and BASF parameter. However, in all cases for these structures the BASF parameters have refined to values of zero (within 3 esd) indicating that twinning is not present. In these cases the twin instruction and BASF should be removed from the instruction file and the structures re-refined without them. A note should be included in the special details that the twin law was tested but refined to BASF of zero, and hence, removed.

Weighting Scheme and Outlier Reflections:

Several structures have Alert A and Alert B level warnings relating to the weighting scheme and outlier reflections. In all cases the outlier reflections should be omitted from the refinements. This will typically be around 2-7 reflections in each structure (not the large number of reflections omitted from the original data as a result of adsorption corrections issues).

Solvent residues geometries:

Several structures contain diethyl ether solvent residues with chemically implausible geometries. These models should either be restrained to give sensible geometries or omitted from model by using a solvent screen or SQUEEZE routine. Where DFIX instructions are used for 1,2 and 1,3 bond distances allow double the ESD for the 1,3 bond distances e.g.

DFIX 1.48 0.01 O73 C74 O73 C72 O77 C76 O77 C79

DFIX 1.53 0.01 C75 C74 C72 C71 C79 C78 C76 C73

DFIX 2.4 0.02 C74 C72 C79 C76 C75 O73 O73 C71 C73 O77 O77 C78

DFIX 3.6 0.02 C74 C71

LIST 6

Some structures have been refined with a LIST 6 instruction rather than the usual LIST 4 instruction. Unless there is a specific reason for using LIST 6 it should be changed to LIST 4 to ensure that the reflections reported in the fcf file are correct for determination of the Flack parameter.

Version of SHELXL:

The structures presented have been refined with SHELXL version 2013/2 rather than the latest release of software 2018/3. It would be beneficial to update your SHELXL installation with the latest version of the software available free of charge for academic use from <https://shelx.uni-goettingen.de/>

S q| cl

BASF is zero – remove

Ether linear – remedy or squeeze

R q| cl

BASF is zero – remove

Ether linear – remedy or squeeze

LIST 6 to LIST 4 for Flack/Hoof

R py br

BASF is zero

Remove outlier reflections to eliminate alert A and Bs

R q| I 150

Remove outlier reflections to eliminate alert A and Bs

R q| I 250K

Remove outlier reflections to eliminate alert A and Bs

R q| I 200K

LIST 6 to LIST 4 for Flack/Hooft
Remove 7 outlier reflections
Ether solvent O2 – poor geometry – fix or solvent screen
R q| br
BASF is zero – remove
R q| I 100K
Remove outlier reflections to eliminate alert A and Bs
R py i
Linear Et2O - resolve or solvent screen
LIST 6 to LIST 4 for Flack/Hooft
Remove outlier reflections to eliminate alert A and Bs
S q| i
Remove outlier reflections to eliminate alert A and Bs
Ether solvent O2 – poor geometry – fix or solvent screen
S py i
Remove outlier reflections to eliminate alert A and Bs
LIST 6 to LIST 4 for Flack/Hooft
R q| I 300K
Remove outlier reflections to eliminate alert A and Bs
R q| Br
Remove outlier reflections to eliminate alert Bs
S py Br
Fine
R q| I 180K
Remove outlier reflections to eliminate alert A and Bs
S_q|_I_CH2Cl2
LIST 6 to LIST 4 for Flack/Hooft

Reviewer #4:

Remarks to the Author:

I acknowledged the authors responded to all the reviewers' comments in a sincere manner and all the answers seem appropriate. I have no additional concerns or questions.

Point-by-point response to the reviewers.

Reviewer #1

The authors considered my previous comments and revised the manuscript suitably. I appreciate the further calculations which strengthen the contents and discussion. Also, the theoretical part of the manuscript is now much improved to understand the contents. I have noticed the following minor points that the authors may consider. I would like to leave these issues to the authors and the further review is not necessary for my review. I recommend the publication of this paper.

Response: We appreciate **Reviewer 1's** positive comments very much.

(1) I think it is better to slightly change the discussion from p. 16 to 17 with less emphasizing the case of *R*-*ql*-Cl, because the k_{ISC} from S_1 to T_1 for *R*-*ql*-Cl is nearly zero and the expression of “faster k_{ISC} and k_{RISC} ” is not valid for *R*-*ql*-Cl cluster and the explanation of the origin for PL of *R*-*ql*-Cl is given later and in p. 18. So, I suggest to focus on the large PLQE for *R*-*ql*-Br and *R*-*ql*-I cases here. For example, The calculated ...(k_{ISC}) from S_1 to T_1 of *R*-*ql*-Br (...) and *R*-*ql*-I (...) clusters is much larger than that of *R*-*ql*-Cl (~ 0 s⁻¹). (p. 16) Resultantly, the faster k_{ISC} and k_{RISC} lead to the high PLQY of *R*-*ql*-Br (62.8%) and *R*-*ql*-I (93.0%) compared to that of *R*-*ql*-Cl (5.4%). (p. 17)

Response: As suggested by **Reviewer 1**, we have made a slight change discussion from p. 16 to 17. “The calculated intersystem crossing rate (k_{ISC}) from S_1 to T_1 of ***R*-*ql*-Br** (1.04×10^{11} s⁻¹) and ***R*-*ql*-I** (2.60×10^{12} s⁻¹) was much larger than that of ***R*-*ql*-Cl** (0 s⁻¹).”

(2) It is recommended to revise some sentences and notations as follows: heavy-atom effect from Cl to Br to I, => heavy-atom effect from Cl to I, (p. 14) $\langle S_1 | H_{SOC} | T_2 \rangle$ => $\langle S_1 | H_{soc} | T_2 \rangle$ (p. 15) SOCME (p. 15) should be defined when it is firstly introduced. inter-system => intersystem (p. 17) Resultantly. => Resultantly, (p. 17).

Response: As suggested by **Reviewer 1**, we have corrected the above problem in the revised manuscript.

Reviewer #2

I appreciate the authors' efforts and time in responding to my comments. Many of them have been answered appropriately. I support the acceptance of this manuscript by Nature Communications, but would like to point out typographical errors and minor details in the revised manuscript.

Response: We appreciate **Reviewer 2** for the recommendation very much.

For the main text part:

1. On page 4, line 70, the expression of “... enantiomer chiral NHC ligands” should be “...using the enantiopure chiral NHC ligands”.

Response: As suggested by **Reviewer 2**, we have corrected “... enantiomer chiral NHC ligands” to “...using the enantiopure chiral NHC ligands”.

2. On page 5, lines 79-80, “Conceiving above strategy, we embedded pyridine/quinolone ring with adjustable π^* orbitals in ligand to achieve spatial separation between donor and acceptor moieties.” In the case of the cited reference 12 (Di et al, Science 356, 159-163 (2017).), the donor and acceptor are indeed separated for about 3.7 Å. However, it is difficult to understand in this study.

Response: As suggested by **Reviewer 2**, we have provided the explanation for this sentence.

“Conceiving above strategy, we embedded pyridine/quinolone ring with adjustable π^* orbitals in ligand, so that we achieve spatial separation between donor (Cu with d electrons) and acceptor (pyridine/quinolone with π^*) moieties in metal cluster.”

In addition, the theoretical calculations gave the description of partial separation between donor and acceptor moieties in *R/S*-**ql-X** (X = Cl, Br, and I) (**Figure 4**).

3. On page 5, lines 88-93, “Two pairs of enantiomers of tridentate NHC ligands were designed and synthesized (Fig. 1a)...” Fig.1a shows the precursors of these NHC ligands, not the direct ligands yet. Please also confirm the caption in Figure 1a. Also, the names using “(4*R*/4*S*, 5*R*/5*S*)-...” are not appropriate, since it may be confused with the case of (4*R*, 5*S*)-. It may be better to use (4*R*, 5*R*)-/ (4*S*, 5*S*)- instead. Similarly, on page 6, line 103, “(1*R*/1*S*, 2*R*/2*S*)-...” may be confused with the case of (1*R*, 2*S*)-.

Response: As suggested by **Reviewer 2**, we have corrected “(4*R*/4*S*, 5*R*/5*S*)-...” and “(1*R*/1*S*, 2*R*/2*S*)-...” to “(4*R*, 5*R*)-/(4*S*, 5*S*)-” and “(1*R*, 2*R*)-/(1*S*, 2*S*)-...”, respectively.

4. On page 6, line 125, “Sohnke space group” should be “Sohncke space group”.

Response: As suggested by **Reviewer 2**, we have corrected “Sohnke space group” to “Sohncke space group”.

5. On page 13, line 264, “(X = Br, and I)” should be “(X = Br and I)”.

On page 14, lines 266-267, (X = Cl, Br and I) should be (X = Cl, Br, and I); a similar problem in lines 273-274, and etc. (please carefully check the main text and SI).

Response: As suggested by **Reviewer 2**, we double-checked the whole manuscript and supporting information thoroughly and corrected the above problem in the revised manuscript and supporting information.

For the Supporting Information:

On page 15, line 361, **Supplementary Figure 1**. “(a) ¹H NMR spectrum of *R*-NHC^{ql}-H·PF₆.” should be “(a) ¹H NMR spectrum of *R*-NHC^{py}-H·PF₆ (CDCl₃ solvent?).” In the same caption, the “*R*-NHC^{py}-H PF₆” should be “*R*-NHC^{py}-H·PF₆”, the “*R*-NHC^{ql}-H PF₆” should be “*R*-NHC^{ql}-H·PF₆”.

Response: As suggested by **Reviewer 2**, we double-checked the whole supporting information thoroughly and corrected the above problem in the revised supporting information.

Reviewer #3

All suggested changes to the manuscript and the requested SCXRD refinement details are fine. Since the original submission the authors have recollected the SCXRD data using Mo radiation and repeated the refinements. The new data is a great improvement and has eliminated the major crystallographic refinement problems which affected the original data sets. There are some minor problems with the new SCXRD refinements which need to be fixed before publication can be recommended. Below is a list of requests for each structure and some general details on the problems which are found across many of the structures.

Response: We sincerely thank **Reviewer 3** for the positive comments.

1. Twin Refinements:

The authors have correctly checked for possible twinning in many of the structures by using a Twin law and BASF parameter. However, in all cases for these structures the BASF parameters have refined to values of zero (within 3 esd) indicating that twinning is not present. In these cases the twin instruction and BASF should be removed from the instruction file and the structures re-refined without them. A note should be included in the special details that the twin law was tested but refined to BASF of zero, and hence, removed.

Response: We thank **Reviewer 3** for the concern. As suggested, the twin law has been tested to show that BASF is zero. We refined all the crystal data and removed BASF and twin instructions, and included them in the special details of cif.

2. Weighting Scheme and Outlier Reflections:

Several structures have Alert A and Alert B level warnings relating to the weighting scheme and outlier reflections. In all cases the outlier reflections should be omitted from the refinements. This will typically be around 2-7 reflections in each structure (not the large number of reflections omitted from the original data as a result of adsorption corrections issues).

Response: We thank **Reviewer 3** for the reminder. As suggested, we rechecked all the crystal data and omitted some outlier reflections to ensure that there were no A and B alerts. In addition, all CIF documents have been updated and uploaded to the CCDC database.

3. Solvent residues geometries:

Several structures contain diethyl ether solvent residues with chemically implausible geometries. These models should either be restrained to give sensible geometries or omitted from model by using a solvent screen or SQUEEZE routine. Where DFIX instructions are used for 1,2 and 1,3 bond distances allow double the ESD for the 1,3 bond distances e.g.

DFIX 1.48 0.01 O73 C74 O73 C72 O77 C76 O77 C79

DFIX 1.53 0.01 C75 C74 C72 C71 C79 C78 C76 C73

DFIX 2.4 0.02 C74 C72 C79 C76 C75 O73 O73 C71 C73 O77 O77 C78

DFIX 3.6 0.02 C74 C71

Response: We thank **Reviewer 3** for the reminder. As suggested, we used the DFIX instruction to re-limit and refine the ether molecule to reach the scientific geometry.

LIST 6

4. Some structures have been refined with a LIST 6 instruction rather than the usual LIST 4 instruction. Unless there is a specific reason for using LIST 6 it should be changed to LIST 4 to ensure that the reflections reported in the fcf file are correct for determination of the Flack parameter.

Response: We thank **Reviewer 3** for pointing out this issue. We double-checked the all crystal data thoroughly and corrected the above problem in the revised crystal data.

5. Version of SHELXL:

The structures presented have been refined with SHELXL version 2013/2 rather than the latest release of software 2018/3. It would be beneficial to update your SHELXL installation with the latest version of the software available free of charge for academic use from <https://shelx.uni-goettingen.de/>

Response: Thanks to the generous sharing of **Reviewer 3**, we have updated the software to make it more convenient to refine the crystal data.

S ql cl

BASF is zero – remove

Ether linear – remedy or squeeze

R ql cl

BASF is zero – remove

Ether linear – remedy or squeeze

LIST 6 to LIST 4 for Flack/Hoof

R py br

BASF is zero

Remove outlier reflections to eliminate alert A and Bs

due to the weak diffraction and serious disorder of PF₆⁻ and CH₂Cl₂ molecules in the lattice,

R ql I 150

Remove outlier reflections to eliminate alert A and Bs

R ql I 250K

Remove outlier reflections to eliminate alert A and Bs

R ql I 200K

LIST 6 to LIST 4 for Flack/Hoof

Remove 7 outlier reflections

Ether solvent O₂ – poor geometry – fix or solvent screen

R ql br

BASF is zero – remove

R ql I 100K

Remove outlier reflections to eliminate alert A and Bs

R py i

Linear Et₂O - resolve or solvent screen
LIST 6 to LIST 4 for Flack/Hooft
Remove outlier reflections to eliminate alert A and Bs
S ql i
Remove outlier reflections to eliminate alert A and Bs
Ether solvent O2 – poor geometry – fix or solvent screen
S py i
Remove outlier reflections to eliminate alert A and Bs
LIST 6 to LIST 4 for Flack/Hooft
R ql I 300K
Remove outlier reflections to eliminate alert A and Bs
S ql Br
Remove outlier reflections to eliminate alert Bs
S py Br
Fine
R ql I 180K
Remove outlier reflections to eliminate alert A and Bs
S ql I CH₂Cl₂
LIST 6 to LIST 4 for Flack/Hooft

Response: We thank **Reviewer 3** for pointing out these issues. We refined all the crystal data to ensure that all of the above issues were addressed. In addition, all CIF documents have been updated and uploaded to the CCDC database. The special measures taken to overcome the problems noted for all structures during the refinement process were described in detail and written in the revised Supporting Information.

Reviewer 4

I acknowledged the authors responded to all the reviewers' comments in a sincere manner and all the answers seem appropriate. I have no additional concerns or questions.

Response: We appreciate **Reviewer 4**'s positive comments very much.